# ConvBench: A Multi-Turn Conversation Evaluation Benchmark with Hierarchical Ablation Capability for Large Vision-Language Models

**Shuo Liu[1], Kaining Ying[1], Hao Zhang[1], Yue Yang[1], Yuqi Lin[1], Tianle Zhang[1], Chuanhao Li[1], Yu Qiao[1], Ping Luo[2,1], Wenqi Shao[1✉], Kaipeng Zhang[1✉]**

[1]OpenGVLab, Shanghai AI Laboratory,
[2]The University of Hong Kong

## Abstract

Multi-turn visual conversation is an important ability of real-world AI assistants. However, the related evaluation benchmark is missed. This paper presents ConvBench, a multi-turn conversation benchmark with hierarchical capabilities ablation evaluation for Large Vision-Language Models (LVLMs). ConvBench comprises 577 curated multi-turn conversations, encompassing 215 tasks. These tasks are broad and open-ended, which resemble real-world user behaviors. ConvBench progressively examines the LVLMs' perception, reasoning, and creativity capabilities in each conversation and can decouple these capabilities in evaluations and thus perform reliable error attribution. Besides, considering the diversity of open-ended questions, we introduce an efficient and reliable automatic evaluation framework. Experimental results reveal that ConvBench is a significant challenge for current LVLMs, even for GPT4V, which achieves only a 39.51% score. Besides, we have some insightful findings, such as the weak perception of LVLMs inhibits authentic strengths in reasoning and creation. We believe our design of hierarchical capabilities, decoupling capabilities evaluation, and multi-turn conversation can blaze a new trail in LVLMs evaluation. Code and benchmark are released at https://github.com/shirlyliu64/ConvBench.

## 1 Introduction

Open-ended multi-turn visual conversations are usual in our daily lives and should be one of the features of artificial general intelligence (AGI). Recent large vision language models (LVLMs, e.g., GPT4V [1], Claude [2], and InternVL-Chat [3]) have achieved impressive performance in such conversations, which is also the common usage of LVLMs. However, existing benchmarks [4, 5, 6, 7, 8, 9, 10, 11, 12, 13] for LVLMs conduct single-turn conversations, and most use close-ended multi-choice questions for evaluation. To this end, this paper presents a multi-turn conversation benchmark named ConvBench to measure the advancement of LVLMs. Beyond open-ended and multi-turn, our questions are hierarchical and real-world, and we introduce an automatic evaluation method. In particular, in each conversation, ConvBench can decouple different capabilities evaluations and perform reliable error attribution, blazing a new trail in LVLMs evaluation.

We first introduce the hierarchical structure of multi-turn questions. Previous benchmarks treat different multimodal capabilities independently using independent questions while ignoring the

---

✉ Corresponding Authors: zhangkaipeng@pjlab.org.cn; shaowenqi@pjlab.org.cn

38th Conference on Neural Information Processing Systems (NeurIPS 2024) Track on Datasets and Benchmarks.

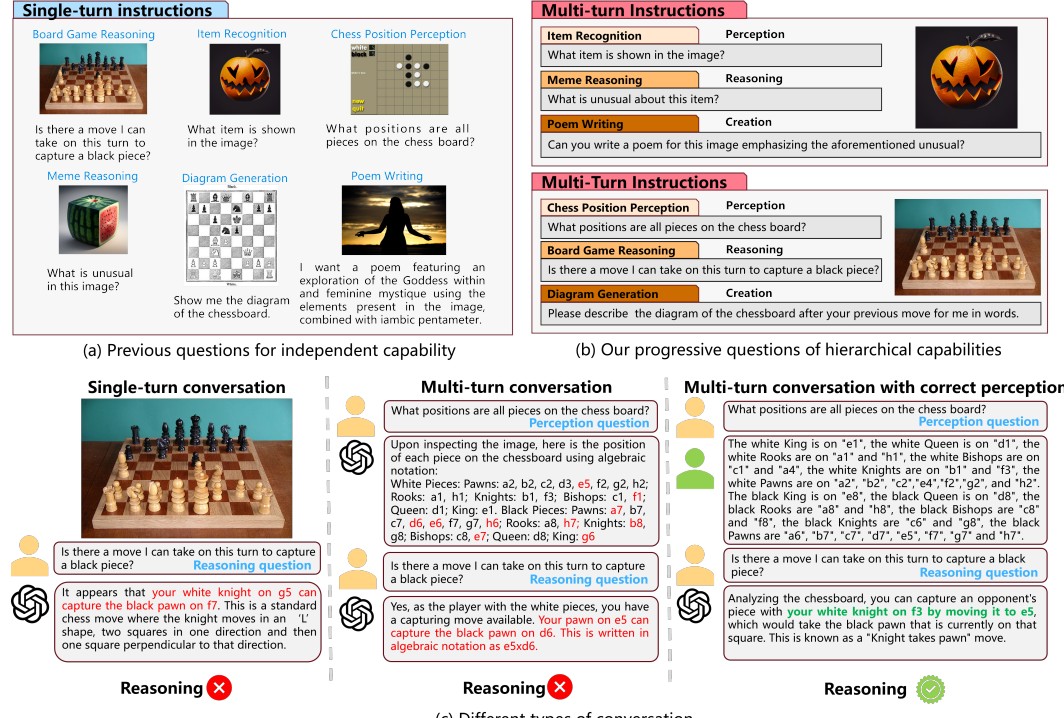

Figure 1: The comparison between previous evaluation benchmarks (a) and our ConvBench (b). Previous benchmarks assess capabilities independently in a single-turn conversation, while our ConvBench evaluates multi-turn conversation by hierarchically assessing perception, reasoning, and creativity. (c) shows that the error should be attributed to perception rather than reasoning, while ConvBench employs annotated reference answers that can do such accurate error attribution.

fact that multimodal capabilities are highly dependent on each other. It makes it hard to conduct error attribution. For example, when the model responds incorrectly to a reasoning question, it is unclear whether it is attributed to the perception or reasoning error of LVLMs (see Figure 1 (c)). Besides, humans reason based on their perceptions and generate new ideas through perceptual and reasoning skills. Therefore, ConvBench progressively examines the perception, reasoning, and creativity capabilities in a multi-turn conversation, seeing Figure 1 (b) for examples. In this way, we can conduct more accurate error attribution by giving reference answers to previous questions.

We then introduce the source of the multi-turn questions. We collect images and questions from the perspective of real-world user behavior. Specifically, VisIT-Bench [7] is a single-turn visual question answering (VQA) dataset collected from users' "wish list". ConvBench uses its images and extends its questions in our multi-turn hierarchical structure. The new questions align with real-world user behavior and involve many new tasks. We illustrate the dataset construction in Figure 3 and details in Section 3. Overall, ConvBench comprises 577 meticulously curated multi-turn VQA samples, spanning 71, 65, and 79 distinct types of perception, reasoning, and creation tasks, respectively.

Unlike most existing benchmarks employing multi-choice questions, we allow LVLMs to output anything to resemble real-world user behavior. However, its evaluation is much more challenging since the correct answer is not only. Therefore, we introduce a reliable automated evaluation named ConvBenchEval for open-ended visual questions. Specifically, we annotate each question's reference answers and each image's question-aware caption. Additionally, we annotate assessment focus points for creation questions for more accurate evaluation. Based on the above annotations (see Figure 3 for examples) and LVLMs' responses, we then employ ChatGPT [14] to judge if the LVLM's response is better than the reference answer. We illustrate the evaluation pipeline in Figure 4. We also conduct a human evaluation on a subset of ConvBench while ChatGPT achieves 81.83% judgment agreement with human evaluation results, demonstrating the reliability of ChatGPT.

We assess 20 publicly available LVLMs. The evaluation results reveal several innovative findings: i) Our ConvBench provides a significant challenge for evaluating the follow-up capability of LVLMs' multi-turn conversation, notably GPT4V [1] achieves only 39.51% overall score. ii) The novel hierarchical ablation evaluations of ConvBench conclude that the weakness of "OCR", "Fine-grained", and "Spatial" perception of current LVLMs may inhibit the performance of the next reasoning and creation tasks. The weakness of LVLMs' reasoning capability demanding "Professional Knowledge", "Emotional Intelligence", "Imagination", and "Sense of Space" may hinder the performance of the next creation. iii) The performances across different tasks of different LVLMs show a similar distribution, which suggests the development of current LVLMs are synchronous. iv) Performance improves as the language model size of LVLM increases. v) A declined performance between the first turn and subsequent turns shows that LVLMs tend to generate comprehension biases as the multi-turn conversation progresses or forget the information of previous turns. vi) The high-quality dialogue history provides important guidance to the LVLMs' responses and plays an important role in in-context learning examples.

The contributions of our work are summarized as follows. (1) We present the first open-ended multi-turn visual conversations benchmark ConvBench. It is challenging, real-world, and aligns with user behavior. (2) In each conversation, ConvBench progressively examines a three-level hierarchy of multimodal capabilities, including perception, reasoning, and creativity. It can decouple these capabilities in evaluations and conduct reliable error attribution. (3) We present an automatic evaluation to handle the challenging open-ended multi-turn conversations and achieve high agreement with the human evaluation. (4) Our experimental results reveal that ConvBench is challenging for current LVLMs, even for GPT4V. Besides, we have some insightful findings from experiments.

## 2 Related Work

### 2.1 Large Vision-Language Models.

Building upon the achievements of Large Language Models (LLMs) [14, 15, 16], Large Vision-Language Models (LVLMs) [1, 17, 18, 19, 20, 21, 22, 23, 24, 25] have recently showcased remarkable proficiency across various tasks, demonstrating advanced perception, reasoning, and creative capabilities. A favored approach to enhancing LVLMs involves integrating visual knowledge into the semantic framework of LLMs, thereby leveraging the LLMs' strong performance in interpreting and responding to prompts. For instance, BLIP-2 [26] introduces the Q-Former to synchronize vision foundation models with LLMs without modifying the underlying models. MiniGPT4 [17] utilizes a straightforward fully connected layer, requiring only a minimal set of caption data. LLaVA [18] enhances the LLM with high-quality instructional data generated by GPT4. QWen-VL [19] undergoes fine-tuning with high-resolution images, employing multi-task training strategies. mPLUG-DocOwl [20] expands the capabilities of LVLMs to include document understanding.

### 2.2 Large Vision-Language Models Benchmarks.

With the advancement of Vision-Language Models (LVLMs), existing standard evaluation benchmarks like MSCOCO [13], GQA [27], VQA [28, 29], etc., are no longer sufficient to assess the comprehensive multimodal abilities of LVLMs. In response, a variety of benchmarks have been developed specifically for LVLM evaluation, including OwlEval [20], LAMM [12], LVLM-eHub [11], SEED [10], MMBench [9], and MM-Vet [8]. These benchmarks primarily focus on assessing basic perceptual abilities. In addition, VisIT-Bench [7] covers a broad spectrum of tasks, ranging from simple recognition to complex reasoning. Recent research has also introduced LVLM benchmarks requiring expert-level domain knowledge and intricate reasoning, such as MathVista [6] and MMMU [5] and MMT-Bench [4]. However, these benchmarks tend to address perception, reasoning, and creation tasks in isolation without establishing connections among these tasks. Furthermore, the current benchmarks predominantly focus on single-turn interactions. The ConvBench addresses these gaps by not only offering a hierarchical ablation evaluation that moves from perception through reasoning to creation but also by evaluating LVLMs' capabilities in multi-turn conversational contexts.

## 2.3 Benchmarks for Multi-turn Large Models.

For LLMs, there have been some classic work. MT-Bench [30] is the first multi-turn conversation benchmark, which focuses on two-turn follow-up dialogues across eight topics ("Writing", "Knowledge", "Math" and so on). It only focuses on the most basic and important multi-turn abilities of Context Memory and Anaphora Resolution. MT-Bench-101 [31] is the first dataset to specifically focus on 13 fine-grained multi-turn dialogue abilities, such as, "Separate Input", "Topic Shift", "Content Confusion", "Content Rephrasing" and so on. Otherwise, there are other benchmarks: MT Bench++ [32] is an eight-turn multi-turn conversation benchmark to qualitatively evaluate multi-turn instruction following ability. It is built on expanding MT-Bench by manually annotating six additional follow-up questions. MINT [33] is a benchmark that evaluates LLMs' ability to solve challenging tasks with multi-turn interactions by using tools and leveraging natural language feedback. For LVLMs, to the best of our knowledge, ConvBench is the first multi-turn visual conversation benchmark, which can also be considered as a multi-modal version of MT-Bench, but with the innovative addition of hierarchical ablation testing. ConvBench depends on manual annotation for obtaining the three-turn interactive instructions to allow evaluating multi-turn instruction-following ability for visual dialogue. ConvBench also mostly focuses on the most basic and important multi-turn visual abilities of Context Memory and Anaphora Resolution.

# 3 ConvBench

## 3.1 Overview of ConvBench

The ConvBench includes 577 image-instruction pairs tailored for multi-turn dialogues. Each pair is structured with three sequential instructions, each targeting a distinct cognitive skill—beginning with perception, followed by reasoning, and culminating in creation. This structure underscores the cognitive evolution from basic perceptual comprehension to logical reasoning and finally to sophisticated creative expression. The detailed definition and division for perception, reasoning, and creation can be seen in the appendix. As shown in Figure 2, our benchmark, encompassing 215 tasks, is divided into 71 tasks focused on perception, 65 on reasoning, and 79 on creation. These practical tasks are creative, useful, and real-world demands, which are downstream users of language technologies are likely to need. Existed benchmarks for LVLMs almost focus on computer-vision deep learning tasks, which cannot assess the ability of LVLMs to solve the diverse and never-before-seen real-world tasks. VisIT-Bench [7] has first proposed a path to evaluate LVLMs for practical tasks. We inherit and extend it to evaluate multi-turn visual conversation capabilities and explore the hierarchical ablation evaluations for LVLMs.

## 3.2 Data Curation Process

**Data Collection.** Our benchmark collection is structured into five distinct stages, as depicted in Figure 3. Annotators play a very important role in the process. These stages are as follows:

**i) Multi-turn Instruction Formation.** We extend each single-turn sample in VisIT-Bench to a sample with three-turn instructions. We ensure the first, second, and third turns are represented as perception, reasoning, and creation, respectively. The instructions are designed based on the last ones.

**ii) Task Category Induction.** We derive the task categories by inducing them from instructions. The bottom-up approach to task collection guarantees that the tasks under investigation are tailored to meet real-world requirements. ConvBench consists of 215 tasks, and we have included the details in the supplementary materials. Task category induction is done with human annotation to ensure reasonableness and accuracy. For example, if the question is "How to write the recipe for the food shown in the image?", the task is inducted as "Recipe Writing".

**iii) Instruction-Conditioned Caption Annotation.** VisIT-Bench [7] has first proposed the generation of the instruction-conditioned caption. The two important purposes for generating instruction-conditioned captions have been evaluated in the VisIT-Bench. One is to provide a comprehensive description of the image for building a robust automated assessment framework. The other is generating raw reference answers for humans to verify in the next "Reference Generation" step. Therefore, in this work, the image and the instructions are also provided for the annotators to generate

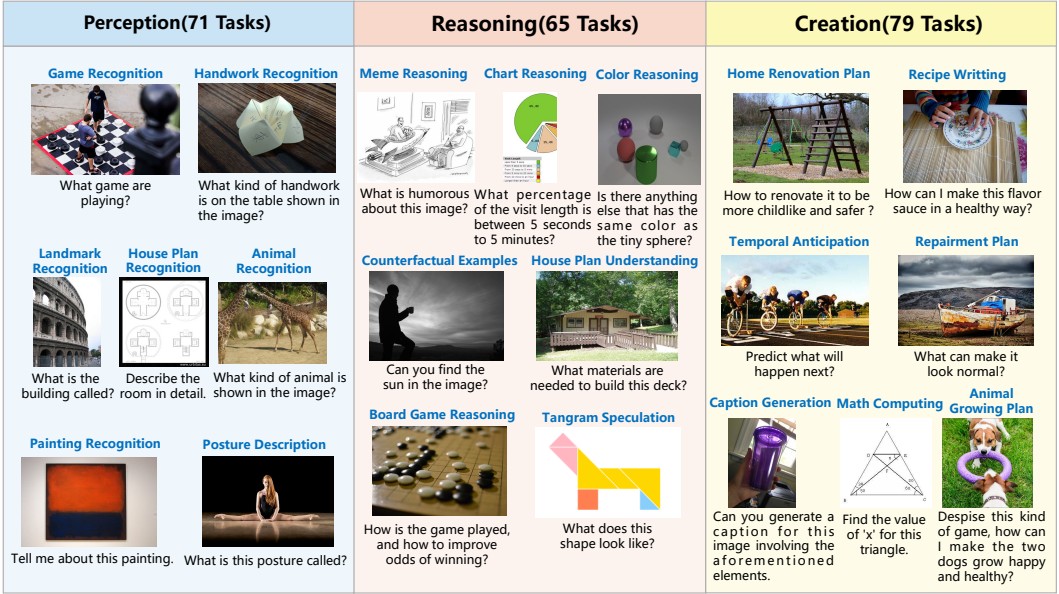

Figure 2: Visualization of example tasks in ConvBench. It consists of 215 tasks constructed in perception, reasoning, and creation hierarchy.

a caption. We first prompt GPT4V with "Describe this image in detail." We then polish the responses by humans according to the instructions to obtain the final instruction-conditioned caption.

**iv) High-Quality Reference Generation.** Similar to VisIT-Bench [7], they verify each response with human annotators. For each sample, we feed GPT4V with the instruction-conditioned caption, the image, multi-turn instructions, and our well-designed prompt in a multi-turn conversation fashion to generate each instruction's response. We meticulously refine these responses as reference answers, removing their biases and enhancing their quality and relevance.

**v) Focus Point Annotation.** The creativity instruction is an open-ended question without a standard answer. Therefore, we annotate specific focus points related to each creation instruction. These annotations are used as criteria to assess whether the model produces instructive answers to the instruction, seeing Step 5 in Figure 3.

## 4  ConvBenchEval and Hierarchical Ablation Evaluation

Human evaluation is really meaningful for judging human preference responses. However, it is costly to obtain human judgments for new model submissions. Similar to VisIT-Bench [7] and MT-Bench [30], to support faster model development, we introduce ConvBenchEval($\cdot$), an automated evaluation pipeline designed for multi-turn visual conversation assessment and hierarchical ablation evaluation. It aligns best with human preferences, whose agreement with human evaluation reaches $81.83\%$. More agreement experiment results can be found in the appendix, which can validate the effectiveness of our evaluation methodology.

ConvBenchEval($\cdot$) comprises four key components: perception, reasoning, creation, and overall conversation evaluation modules, as shown in Figure 4. We recursively use ConvBenchEval($\cdot$) in three settings as follows to conduct the hierarchical ablation evaluation for error attribution. For clarity, we denote the instruction, model response, reference, and focus points as $I_i, M_i, R_i$, and $F_i$ at each turn, respectively, where $i$ indicates the turn index. Note that $F_1$ and $F_2$ are null focus points.

**i) ConvBenchEval for Evaluating the Performances of Each Turn and Overall Conversation.** The model response of each turn is obtained in the principle of multi-turn conversation. The response of each turn is generated based on the front instructions and responses in this setting. The formula for generating each turn's response can be expressed as $M_i = f(\{I_i\}_{i=0}^{i-1}, \{M_i\}_{i=0}^{i-1}, I_i), i = 1, 2, 3$, where $f$ denotes the model inference function, $I_i$ denotes the instruction at the turn $i$, $M_i$ denotes

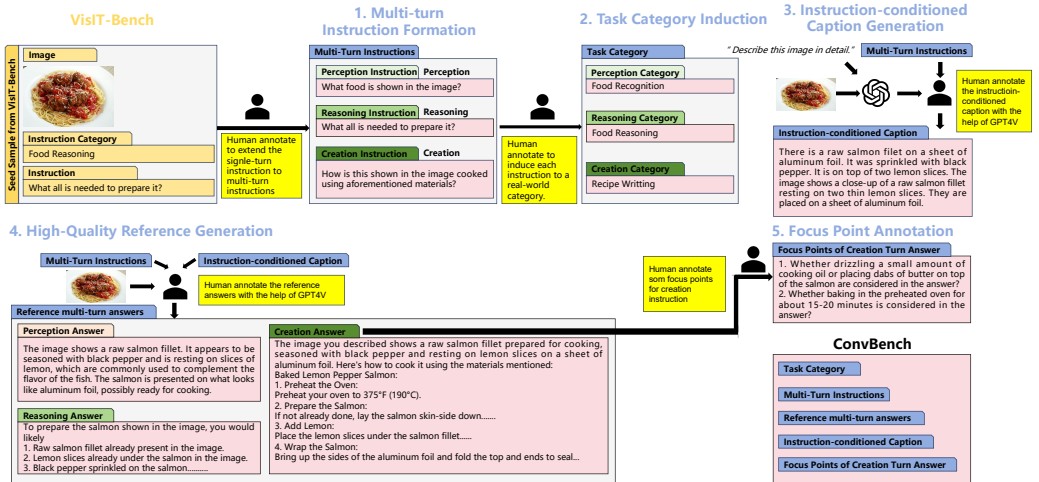

Figure 3: The pipeline of data curation. We develop three multi-turn instructions for each image to assess the perception, reasoning, and creation capabilities. We also annotate the referenced answers to facilitate automatic evaluation and error analysis.

the response at the turn $i$. In the first turn, there is no front instruction ($I_0$) or response ($M_0$). The evaluation process can be expressed as $S_i, J_i = \text{ConvBenchEval}(\{M_i\}_{i=1}^3, \{R_i\}_{i=1}^3, F_3; P_i), i = 1, 2, 3$ where $S_i$ is the capability score to the turn $i$, $J_i$ is the judgment from the ChatGPT to the turn $i$, and $P_i$ is the prompt specific to the turn $i$. Finally, we feed all instructions, responses, focus points, and judgments into ChatGPT to obtain the overall conversation score as given by $S_O, J_O = \text{ConvBenchEval}(\{M_i\}_{i=1}^3, \{R_i\}_{i=1}^3, \{J_i\}_{i=1}^3; P_O)$ where $S_O$ is the capability score for overall multi-turn conversation, $J_O$ is the judgment from the ChatGPT to the overall multi-turn conversation, $P_O$ is the prompt for evaluating the overall conversation capability.

**ii) Hierarchical Ablation Evaluation with Perfect Perception Condition.** In this setting, the influence of inaccurate perception on reasoning, creation, and overall conversation can be derived. We directly use the human-annotated perception answers as the responses of the perception turn, replacing the outputs from LVLMs at the perception turns, which can be represented as $\hat{M}_1 = R_1$. The model inference at the reasoning and creation turns can be written as $\hat{M}_i = f(\{I_i\}_{i=1}^{i-1}, \{\hat{M}_i\}_{i=1}^{i-1}, I_i), i = 2, 3$, where $f$ denotes the model inference function, $I_i$ denotes the instruction at the turn $i$, $\hat{M}_i$ denotes the response at the turn $i$ with the perfect perception conditions. The evaluation process without considering perception error can be expressed as $\hat{S}_i, \hat{J}_i = \text{ConvBenchEval}(\{\hat{M}_i\}_{i=1}^3, \{R_i\}_{i=1}^3, F_3; P_i), i = 2, 3$. Finally, the overall conversation score without considering perception score can be given by $\hat{S}_O, \hat{J}_O = \text{ConvBenchEval}(\{\hat{M}_i\}_{i=1}^3, \{R_i\}_{i=1}^3, \{\hat{J}_i\}_{i=2}^3; P_O)$, where $\hat{S}_O$ is the capability score for overall multi-turn conversation with perfect perception conditions, $\hat{J}_O$ is the judgment from the ChatGPT for the overall multi-turn conversation with perfect perception conditions, $P_O$ is the prompt for evaluating the overall conversation capability. By comparing $S_i$ and $\hat{S}_i$ ($i = 2, 3, O$), we can see how perception error affects the performance of reasoning, creativity, and overall conversation.

**iii) Hierarchical Ablation Evaluation with Perfect Perception and Reasoning Conditions.** In this setting, we further explore how reasoning errors affect creativity and overall conversation. The human-annotated perception answer and reasoning answer are directly used as the responses of the perception turn and reasoning turn, respectively, replacing the original responses from LVLMs, which can be represented as $\tilde{M}_i = R_i, i = 1, 2$. Then, the model inference at the creation turn can be written as $\tilde{M}_i = f(\{I_i\}_{i=0}^{i-1}, \{\tilde{M}_i\}_{i=1}^{i-1}, I_i), i = 3$, where $f$ denotes the model inference function, $I_i$ denotes the instruction at the turn $i$, $\tilde{M}_i$ denotes the response at the turn $i$ with perfect perception and reasoning conditions. The evaluation process without considering perception and reasoning error can be expressed as $\tilde{S}_i, \tilde{J}_i = \text{ConvBenchEval}(\{\tilde{M}_i\}_{i=1}^3, \{R_i\}_{i=1}^3, F_3; P_i), i = 3$. Finally, the overall conversation score without considering perception and reasoning score can be given

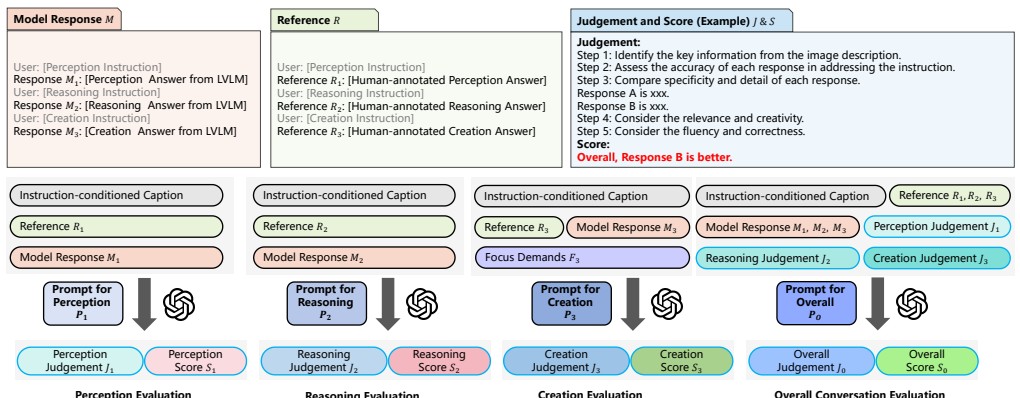

Figure 4: An illustration of the evaluation pipeline of ConvBench. 🅢 indicates the ChatGPT.

by $\tilde{S}_O, \tilde{J}_O = \text{ConvBenchEval}(\{\tilde{M}_i\}_{i=1}^3, \{R_i\}_{i=1}^3, \tilde{J}_3; P_O)$, where $\tilde{S}_O$ is the capability score for overall multi-turn conversation with perfect perception and reasoning conditions, $\tilde{J}_O$ is the judgment from the ChatGPT to the overall multi-turn conversation with perfect perception and reasoning conditions, $P_O$ is the prompt for evaluating the overall conversation capability. By comparing $\hat{S}_i$ and $\tilde{S}_i$ $(i = 3, O)$, we can check how perception and reasoning errors affect the performance of creativity and overall conversation.

**Pairwise Grading.** Note that the function in i) - iii) $\text{ConvBenchEval}(\cdot)$ is a pairwise scheme as described in the following. Similar to VisIT-Bench [7], $\text{ConvBenchEval}(\cdot)$ also employs a pairwise grading scheme and an absolute metric. Specifically, all the LVLM responses and the references shown in Figure 4, are anonymously presented to the ChatGPT in a random order. The ChatGPT is tasked with a pairwise comparison to decide the responses from which is superior. The percentages of cases where the ChatGPT prefers the output from LVLM rather than the human-verified reference output are defined as the absolute metric, *i.e.* win rate. This metric can directly judge whether the LVLMs' performance reaches human preference. The system prompts designed for the pairwise grading scheme for obtaining the win rate, detailed in the Appendix, encourage the ChatGPT to engage in a step-by-step thought process, making its reasoning explicit. In our forced-choice setup, ties are not permitted; thus, if the ChatGPT deems the responses of equal quality, it is instructed to select one arbitrarily.

## 5 Experiment and Analysis

### 5.1 Hierarchical Ablation Evaluation Comparisons and Analysis

We employed our proposed $\text{ConvBenchEval}(\cdot)$ methodology to perform a quantitative analysis. The outcomes of the evaluation results are detailed in Table 1. $S_1$, $S_2$, and $S_3$ denote the scores for perception, reasoning, and creation, respectively. Meanwhile, $\hat{S}_2$ and $\hat{S}_3$ correspond to the scores for reasoning and creation, respectively, and under conditions of perfect perception. $\tilde{S}_3$ is the score for creation, assuming perfect conditions for both perception and reasoning. We present our principal insights from the evaluation results as follows:

(1) **ConvBench Provides Challenge for LVLMs.** This benchmark sets formidable challenges for modern models. GPT4V, despite being a sophisticated model, shows only modest achievements in perception, reasoning, and creation. Moreover, there is still a gap in performance between open-source models and closed-source models in various real-world use cases. ConvBench exposes a stark discrepancy between the performance of these models and that of humans.

(2) **Weak Perception Undermines LVLMs' Reasoning and Creation Performance.** Under conditions of perfect perception, we see significant improvements in reasoning and creation abilities, as indicated by the data in the $\hat{S}_2$ and $\hat{S}_3$ columns of Table 1. The figure reflects the enhancement in reasoning and creation attributed to impeccable perception. Across 20 LVLMs, the average increase

Table 1: Comparison of Performance for LVLMs on ConvBench. Quantitative ConvBench Evaluation Results for 20 LVLMs with Pairwise Grading method. The results in the table are win-rate vs human. $R_2$ is defined as $(S_1 + S_2 + S_3)/3$, indicative of the mean performance over three turns. Meanwhile, $R_1$ is computed as $(R_2 + S_O)/2$, representing the model's overall score.

| Model | $R_1$ | $R_2$ | $S_1$ | $S_2$ | $S_3$ | $S_0$ | $\hat{S}_2(\hat{S}_2 - S_2)$ | $\hat{S}_3(\hat{S}_3 - S_3)$ | $\hat{S}_O(\hat{S}_O - S_0)$ | $\tilde{S}_3(\tilde{S}_3 - \hat{S}_3)$ | $\tilde{S}_O(\tilde{S}_O - \hat{S}_O)$ |
|---|---|---|---|---|---|---|---|---|---|---|---|
| GPT4V [1] | **39.51** | **38.47** | 38.47 | **39.34** | **37.61** | **40.55** | 47.31(+16.97) | 37.78(+0.61) | 37.61(-2.94) | 38.99(+1.21) | 38.30(+0.69) |
| Claude [2] | 36.60 | 37.49 | **38.99** | 39.17 | 34.32 | 35.70 | 45.93(+6.76) | **38.99**(+4.67) | 43.15(+7.45) | **39.16**(+0.17) | **40.21**(-2.94) |
| Reka Flash [34] | 25.60 | 24.67 | 25.13 | 27.56 | 21.32 | 26.52 | 32.93(+5.37) | 22.88(+1.56) | 25.82(-0.70) | 24.78(+1.90) | 26.00(+0.18) |
| InternVL-Chat-V1-2 [3] | 21.17 | 22.41 | 24.96 | 21.31 | 20.97 | 19.93 | 32.06(+10.75) | 28.25(+7.28) | 29.64(+9.71) | 33.62(+5.37) | 35.18(+5.54) |
| InternVL-Chat-V1-5 [3] | 17.65 | 20.22 | 26.00 | 17.33 | 17.33 | 15.08 | 27.73(+10.40) | 23.40(+6.07) | 25.13(+10.05) | 32.24(+8.84) | 33.80(+8.67) |
| ShareGPT4V-13B [35] | 17.56 | 17.45 | 17.85 | 18.72 | 15.77 | 17.68 | 32.58(+13.86) | 30.33(+14.56) | 28.94(+11.26) | 32.41(+2.08) | 31.54(+2.60) |
| LLaVA-V1.5-13B [36] | 16.93 | 18.08 | 20.45 | 18.02 | 15.77 | 15.77 | 32.76(+14.74) | 25.65(+9.88) | 28.94(+13.17) | 32.06(+6.41) | 28.94(+0.00) |
| ShareGPT4V-7B [35] | 16.32 | 16.87 | 16.81 | 19.24 | 14.56 | 15.77 | 32.76(+13.52) | 23.05(+8.49) | 25.13(+9.36) | 29.46(+6.41) | 30.33(+5.20) |
| LLaVA-V1.5-7B [36] | 16.15 | 17.56 | 19.06 | 19.24 | 14.38 | 14.73 | 33.80(+14.56) | 23.22(+8.84) | 26.52(+11.79) | 30.68(+7.46) | 32.58(+6.06) |
| XComposer2 [37] | 15.83 | 16.41 | 17.16 | 19.06 | 13.00 | 15.25 | 30.50(+11.44) | 20.97(+7.97) | 22.36(+7.11) | 28.60(+7.63) | 29.81(+7.45) |
| mPLUG-Owl2 [38] | 14.93 | 15.83 | 17.50 | 17.16 | 12.82 | 14.04 | 27.90(+10.74) | 17.50(+4.68) | 20.80(+6.76) | 24.26(+6.76) | 24.44(+3.64) |
| Qwen-VL-Chat [19] | 14.33 | 14.62 | 16.29 | 18.37 | 9.19 | 14.04 | 28.25(+9.88) | 16.12(+6.93) | 22.70(+8.66) | 25.30(+9.18) | 26.52(+3.82) |
| MiniGPT4 [17] | 10.95 | 10.80 | 11.61 | 11.27 | 9.53 | 11.09 | 27.56(+16.29) | 18.20(+8.67) | 22.53(+11.44) | 22.88(+4.68) | 23.74(+1.21) |
| LLaMA-Adapter-v2 [39] | 9.04 | 9.59 | 8.84 | 10.92 | 9.01 | 8.49 | 27.38(+16.46) | 15.60(+6.59) | 19.41(+10.92) | 18.37(+2.77) | 19.24(-0.17) |
| MMAlaya [40] | 5.55 | 5.89 | 7.28 | 6.41 | 3.99 | 5.20 | 22.53(+16.12) | 9.88(+5.99) | 15.25(+10.05) | 14.21(+4.33) | 16.81(+1.56) |
| Monkey [41] | 3.70 | 4.10 | 3.64 | 5.20 | 3.47 | 3.29 | 16.64(+11.44) | 7.28(+3.81) | 10.75(+7.46) | 13.86(+6.58) | 15.94(+5.19) |
| Otter [22] | 2.78 | 2.60 | 3.12 | 3.12 | 1.56 | 2.95 | 14.21(+11.09) | 5.37(+3.81) | 9.01(+6.06) | 8.49(+3.12) | 13.00(+3.99) |
| XComposer [37] | 1.21 | 1.73 | 1.73 | 1.91 | 1.56 | 0.69 | 12.13(+10.22) | 2.77(+1.21) | 8.49(+7.80) | 10.40(+7.63) | 12.48(+3.99) |
| BLIP2-FLAN-T5-XXL [42] | 0.32 | 0.29 | 0.35 | 0.52 | 0.00 | 0.35 | 3.47(+2.95) | 1.91(+1.91) | 2.95(+2.60) | 5.72(+3.81) | 8.49(+5.54) |
| BLIP2-FLAN-T5-XL [42] | 0.06 | 0.11 | 0.00 | 0.17 | 0.17 | 0.00 | 3.12(+2.95) | 0.17(+0.00) | 2.25(+2.25) | 3.97(+3.80) | 8.67(+6.42) |

in reasoning and creation scores are 11.38 and 5.65, respectively. By analyzing these enhanced reasoning or creation cases with perfect perception information, we found that the **"OCR" Perception Task** may influence the performances of "In-context Visual Scene Understanding", "Meme Reasoning", and "Chart Reasoning". The **Fine-Grained Perception Tasks** like "Location Recognition" perception task may influence the performances of "Location Understanding" and "Travel Plan Writing" tasks, "Celebrity Recognition" perception task may influence the performances of "Celebrity Understanding", "PowerPoint Production" and "Cultural Knowledge Reasoning" tasks. The **Spatial Perception Tasks** like "Board Chess Position Description" may influence the performances of "Board Game Reasoning" and "Diagram Generation" tasks. Our benchmark clarifies the origins of these errors in reasoning and creativity, determining whether they stem from visual perception issues or language reasoning shortcomings. With the aid of human-verified visual comprehension, the authentic strengths of the language module in reasoning and creation will be more precisely evaluated.

(3) **Limited Reasoning Impacts LVLM's Creation Abilities.** Under ideal conditions for perception and reasoning, shifts in creation capabilities are documented in the $\tilde{S}_3$ column of Table 1. The numbers in brackets indicate adjustments in creation scores due to human-verified reasoning accuracy. Among the 20 LVLMs evaluated, there is an average increase of 5.32 in creation scores, which indicates that reasoning inaccuracies can adversely affect LVLMs' performance in creative tasks. By analyzing these enhanced creation cases with perfect reasoning information, we found that some reasoning tasks involving **Professional Knowledge** influence the next creation tasks. For example, "Physical Knowledge Reasoning" may influence the accuracy of "Physical Problem Computing". Some reasoning tasks needing **Emotional Intelligence** influence the relative creation tasks. For example, "Human Emotion Reasoning" may influence the performance of "Blog Writing", "Humanity Discussion" and "How Visual Content Arouses Emotions" tasks. Some reasoning tasks containing **Imagination** like "Tangram Speculation" may influence the corresponding creative task ("Tangram Segmentation"). Some reasoning tasks requiring **Sense of Space** like "Location Relative Position" may influence the "Navigation" task.

(4) **LVLM's Performance across Various Real-world Tasks.** As depicted in Figure 2 in the appendix, LVLMs demonstrate weak performance in fine-grained tasks, such as "Movie recognition", "Position Description", "Outfit recognition", "Make-up Description" and so on. Lack of real-world application data for pretraining and finetuning may lead to the weakness. The datasets used for pretraining and finetuning are more high-quality, the better performance will be. The performances of open source LVLMs, which are above 14.00, are all using various high-quality datasets for training. The superiority of model architecture may be constructed on high-quality datasets. Also as shown in

Figure 2 in the appendix, the performance of open-source models with better performance on different tasks shows a similar distribution as that of closed-source models. It means that the challenges of real-world cases faced by open-source and closed-source models are similar, which suggests the development of current LVLMs is synchronous. ConvBench aids in highlighting the strengths and weaknesses of the instruction-following LVLMs along various real-world use cases.

## 5.2 Multi-Turn Conversation Comparisons and Analysis

The results of the multi-turn conversation evaluation are meticulously outlined in Table 1. $S_O$ represents the scores for multi-turn conversation performance. Concurrently, $\hat{S}_O$ signifies the scores for multi-turn conversation performance under the assumption of flawless perception. Moreover, $\tilde{S}_O$ reflects the score for multi-turn conversation performance, premised on ideal conditions for both perception and reasoning. We delineate our key findings from the experimental results as follows:

(1) **The Challenge of Follow-up Multi-Turn Conversation for LVLMs.** ConvBench is manually tailored by designing the questions based on the last responses, always referencing specific contents mentioned in the last responses. Like the real-world application of a general-purpose AI assistant, a user always asks for additional information based on the assistant's prior responses. ConvBench provides a benchmark for evaluating the LVLM's follow-up ability to engage in coherent conversations. Table 1 shows that closed-source LVLMs, including GPT4V, Claude, and Reka, generally outperform open-source ones in the follow-up capability of multi-turn dialogues. ConvBench presents substantial challenges in multi-turn visual conversations with LVLMs and plays a role in the LVLM field, such as MT-Bench [30] in the LLM field.

(2) **Per-Turn Performance.** According to Table 1, we compute the mean scores of 20 models for each turn, to explore the impact of turn count on LVLMs' performance. The average performances of LVLMs are 15.44, 15.40, and 12.55 in the first, second, and third turn, respectively, which shows a decline between the first turn and subsequent turns, which suggests that LVLMs tend to generate comprehension biases as the multi-turn dialogue progresses or forget the content of previous turns.

(3) **The Impact of Dialogue History in Multi-Turn Conversation for LVLMs.** When comparing the $S_O$ column against the $\hat{S}_O$ and $\tilde{S}_O$ columns in Table 1, the numbers in parentheses illustrate the improvement in overall multi-turn conversation performance resulting from flawless previous responses. The enhancement indicates that the LVLMs can leverage previous responses to improve the responses in the current turn. We also find that the high-quality dialogue history, which plays an important role in in-context learning examples, provides effective guidance for the LVLM's responses.

(4) **Effect of Language Model Size.** We find that the trend of increasing language model size is related to an improvement in LVLM's performance on multi-turn conversation by comparing the InterVL-Chat-v1-2, ShareGPT4-13B, and LLaVA-V1.5-13B with the InterVL-Chat-v1-5, ShareGPT4-7B, and LLaVA-V1.5-7B. The detailed model information is shown in the supplementary.

## 6 Conclusion and Limitation

We introduce a multi-turn conversation benchmark named ConvBench for LVLMs. It comprises 577 multi-turn conversations across 215 tasks. Each conversation consists of three-level hierarchical questions: perception, reasoning, and creation. It can decouple capacities in evaluation for more accurate error attribution. Besides, an automatic evaluation pipeline is proposed. From the experimental results, ConvBench is challenging for current LVLMs, and we also have some insightful findings.

**Limitation** Each conversation in ConvBench is constructed through meticulous annotations and large labor to ensure quality. Thus, the data scale is not too large. We will continually expand ConvBench in terms of data scale and the number of tasks in our future work.

**Broader Impact** We hope this work can blaze a new trail in LVLMs evaluation. We do not foresee obvious undesirable ethical/social impacts at this moment.

## Acknowledgments and Disclosure of Funding

This paper is partially supported by the National Key R&D Program of China (No.2022ZD0161000, No.2022ZD0160101, No.2022ZD0160102) and the General Research Fund of Hong Kong No.17200622 and 17209324.

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

# A  LVLMs Configuration

Table 2 summarizes the LVLMs evaluated in this paper, including their model sizes, visual encoders, and LLMs.

Table 2: Model architecture of 20 LVLMs evaluated on ConvBench.

| Models | Parameters | Vision Encoder | LLM |
|---|---|---|---|
| GPT4V [1] | - | - | - |
| Claude [2] | - | - | - |
| Reka Flash [34] | - | - | - |
| ShareGPT4V-7B [35] | 7.2B | CLIP ViT-L/14 | Vicuna-v1.5-7B |
| ShareGPT4V-13B [35] | 13.2B | CLIP ViT-L/14 | Vicuna-v1.5-13B |
| LLaVA-v1.5-7B [36, 43] | 7.2B | CLIP ViT-L/14 | Vicuna-v1.5-7B |
| LLaVA-v1.5-13B [36, 43] | 13.4B | CLIP ViT-L/14 | Vicuna-v1.5-13B |
| XComposer [37] | 8B | EVA-CLIP-G | InternLM-7B |
| XComposer2 [44] | 7B | CLIP ViT-L/14 | InternLM2-7B |
| mPLUG-Owl2 [38] | 8.2B | CLIP ViT-L/14 | LLaMA2-7B |
| QWenVL [19] | 9.6B | CLIP ViT-G/16 | QWen-7B |
| LLaMA-Adapter-v2 [39] | 7B | CLIP-ViT-L/14 | LLaMA-7B |
| BLIP2-Flan-T5-XL [42] | 12.1B | EVA-CLIP ViT-G/14 | Flan-T5-XL |
| BLIP2-Flan-T5-XXL [42] | 12.1B | EVA-CLIP ViT-G/14 | Flan-T5-XXL |
| InternVL-Chat-V1.2 [3] | 40B | InternViT-6B | Nous-Hermes-2-Yi-34B |
| InternVL-Chat-V1.5 [45] | 26B | InternViT-6B | InternLM2-20B |
| Monkey [41] | 9.8B | CLIP-ViT-BigHuge | Qwen-7B |
| MiniGPT4 [17] | 8.0B | EVA-G | Vicuna-7B |
| MMAlaya [40] | 7.8B | BLIP2-opt-2.7b | Alaya-7B-Chat |
| Otter [22] | 1.3B | CLIP ViT-L/14 | LLaMA-7B |

## B Task Category

| | Task Category |
|---|---|
| **Perception** | "Car Recognition","OCR","Role Identification","Celebrity Recognition","OCR Math","Material Recognition","Sign Description","Furniture Recognition","Location Recognition","Board Chess Position Detection","Painting Recognition","Device Recognition","Movie Recognition","Structure Recognition","Tangram Description","House Plan Recognition","Human Description","Flower Recognition","Outfit Recogntion","Graph Description","Statue Recogntion","Product Recognition","Image Description","Item Recognition","Handwork Recognition","Profession Identification","Food Recognition","Behavior Recognition","Color Recognition","Scene Description","Expression Recognition","Chemical Identification","Object Counting","Medicine Recognition","Shape Recognition","Plant Identification","Item Recogntion","Animal Recognition","Medical Recognition","Photo Recognition","Landmark Recognition","Length Estimate","Chart Description","Posture Description","Gestures Recognition","Aircraft Recognition","Traffic Sign identification","Event Recognition","Injury Description","Hazard Identification","Emotion Conditioned Output","Exercise Recognition","Geometry Problems Description","Astronomy Identification","Weather Recogntion","Game Recognition","Meter Reading","Recipe Description","Food Chain Description","Food Description","Position Description","Historical Relic Identification","Art Work Description","Sculpture Description","Logo Recognition","Make-up Description","Spatial Relationship Understanding","Weather Recognition","Attire Recognition","Differently Abled Recognition","Capacity Estimate" |
| **Reasoning** | "In-context Visual Scenace Understanding","Meme Reasoning","Math Reasoning","Structure Understanding","Traffic Sign Reasoning","Furniture Understanding","Location Understanding","Board Game Reasoning","Art Knowledge Reasoning","Device Reasoning","Dressing Reasoning","Tangram Speculation","Anagrams Reasoning","House Plan Understanding","Question Generation","Visual Commonsense Reasoning","Flower Understanding","Pop Culture Reasoning","Exercise Reasoning","Celebrity Understanding","Product Instruction","Figurative Speech Explanation","Paper Folding Reasoning","History Knowledge Reasoning","Physical Knowledge Reasoning","Cultural Knowledge Reasoning","Climate and Weather Understanding","Food Reasoning","Human Emotion Reasoning","Chemical Knowledge Reasoning","Biology Knowledge Reasoning","Medical Reasoning","Count Reasoning","Rhetoric Reasoning","Plant Identification Reasoning", "Chart Reasoning","Counterfactual Examples","In-context Visual Scene Understanding","Flavor Reasoning","Capacity Estimate Reasoning","Color Reasoning","Gestures Understanding","Location Relative Position","Contextual Knowledge of Events","Word Translation","Rational Action Identification","Geography Reasoning","Hazard Reasoning","Physical Knowledge Reasoning ","Position Reasoning","Graph Reasoning","Sign Understanding","Material Reasoning","Make-up Reasoning","Object Counting Reasoning","Sport Level Reasoning","Astronomy Reasoning","OCR Math Reasoning","Age Reasoning","Damage Evaluation Reasoning","Abstract Reasoning","Music Reasoning","Environment Reasoning","Role Identification Reasoning" |
| **Creation** | "Slogan Generation","Caption Generation","Math Computing","Building Materials Plan","Home Renovation Plan","Blog Writing","Algorithm Design","Artistic Appreciation","Device Principle Explanation","Movie Synopsis Writing","Travel Plan Writing","Tangram Segmentation","Computer Programming","Physical Computing","Advertisement Writing","Legalization Discussion","Chemistry Discussion","Roleplay","Plant Growing Plan","Exercise Plan","Place Recommendation","Dialogue Generation","Computer Knowledge","Essay Writing","How Visual Content Arouses Emotions","Diagram Generation","Poem Writing","Painting Drawing Teaching","Prompt Generation for Image Generation","News Report Writing","Temporal Anticipation","Multilingual Multicultural Understanding","Appliance Evaluation Report","Career Plan Generation","Treatment Plan","Catchy Titles Generation","Prompt Generation for Image Edition","Device Instructions Teaching","Calorie Estimate","Recipe Writting","Computer Program Description","Navigation","Science Fiction Scene Writing","Medical Suggestion","Humanity Discussion","Math Funtion Graphing","Story Writing","Metaphor Writing","Constrained Prompting","Photography Plan","Customized Captioner","Makeup Design","Chemical Computing","Powerpoint Production","Self-driving Design","Exercise Promotional Article Writing","Animal Growing Plan","Event Infulence and Meaning Discussion","Lyric Writing","Movie Review Writing","Packing Plan","Rubik's Cube Solution","Repairment Plan","Insurance Report Generation","Chemical Knowledge Computing","Mathematical Proof","Activity Recommendation","Biology Discussion","Geography Discussion","Meme Writing","Clothing Recommendation","Hairstyle Design","Planing","Nail Art Design","Damage Evaluation","Food Chain Computing","Architectural Plan","Metaphor Writting","Legal Rights Protection" |

Figure 5: List of existing task categories in ConvBench.

As shown in Figure 6, the performance of open-source models with better performance on different tasks shows a similar distribution as that of closed-source models. It means that the challenges of real-world cases faced by open-source and closed-source models are similar, which suggests the development of current LVLMs is synchronous. ConvBench aids in highlighting the strengths and weaknesses of the instruction-following LVLMs along various real-world use cases.

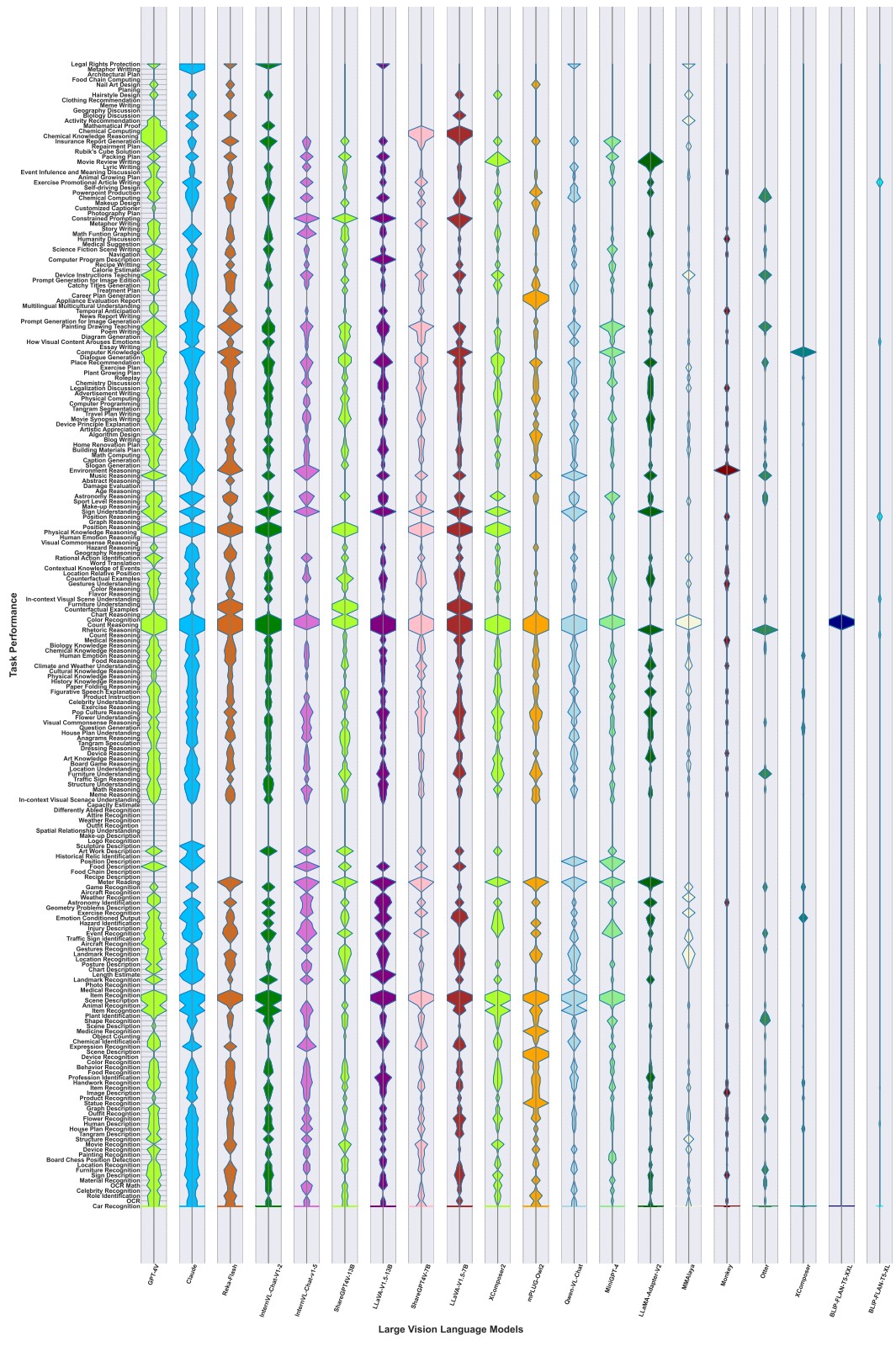

Figure 6: The Visualizations of model performance across various categories.

# C  Human Evaluation

We randomly sample 15 three-turn conversations for each model, a total of 300 dialogues, and recruit 20 human annotators to evaluate the performance of each turn and overall multi-turn conversation quality by judging if the responses of LVLMs outperformed reference answers (anonymous to annotators). We calculate the agreement of human judgment and our automatic evaluation (i.e., ConvBenchEval(·)) and find it reaches 81.83% (seeing Table 4 - 7 for detailed agreement of each turn of overall). It demonstrates the effectiveness of ConvBenchEval(·), which uses ChatGPT.

As shown in Table 3, we also try GPT4 in ConvBenchEval(·), and the agreement reaches 85.56%, similar to ChatGPT. The agreement between ChatGPT and GPT4 is very high at 87.38%. It demonstrates that using different LLMs as judges slightly influences the evaluation results. ConvBenchEval(·) armed with ChatGPT can is reliable and low-cost.

From the above tables, we also observe that though GPT4V is expensive and can capture images, its judgment performs worse than GPT4's judgment. It suggests that the annotated instruction-conditioned caption can provide accurate information from images, which is essential for judgment, while LVLM judges may fail in some cases. Also, annotating instruction-conditioned captions actually can help annotators to design high-quality questions.

Table 3: Agreement among different judgements. "Human" denotes the human expert. "ChatGPT", "GPT4", and "GPT4V" denotes ConvBenchEval(·) using ChatGPT, GPT4, and GPT4V, respectively. In GPT4V, we do not use annotated instruction-conditioned captions but the images.

| Set up Judge | Human | ChatGPT | GPT4 | GPT4V |
|---|---|---|---|---|
| Human | - | 81.83% | 85.56% | 82.14% |
| ChatGPT | - | - | 87.38% | 83.17% |
| GPT4 | - | - | - | 92.38% |

Table 4: Perception agreement among different judgments. "Human" denotes the human expert. "ChatGPT", "GPT4", and "GPT4V" denotes ConvBenchEval(·) using ChatGPT, GPT4, and GPT4V, respectively. In GPT4V, we do not use annotated instruction-conditioned captions but the images.

| Set up Judge | Human | ChatGPT | GPT4 | GPT4V |
|---|---|---|---|---|
| Human | - | 76.51% | 79.37% | 74.60% |
| ChatGPT | - | - | 83.49% | 76.83% |
| GPT4 | - | - | - | 87.94% |

Table 5: Reasoning agreement among different judgments. "Human" denotes the human expert. "ChatGPT", "GPT4", and "GPT4V" denotes ConvBenchEval(·) using ChatGPT, GPT4, and GPT4V, respectively. In GPT4V, we do not use annotated instruction-conditioned captions but the images.

| Set up Judge | Human | ChatGPT | GPT4 | GPT4V |
|---|---|---|---|---|
| Human | - | 80.32% | 87.62% | 85.34% |
| ChatGPT | - | - | 83.17% | 80.63% |
| GPT4 | - | - | - | 94.29% |

Table 6: Creation agreement among different judgments. "Human" denotes the human expert. "ChatGPT", "GPT4", and "GPT4V" denotes ConvBenchEval(·) using ChatGPT, GPT4, and GPT4V, respectively. In GPT4V, we do not use annotated instruction-conditioned captions but the images.

| Set up Judge | Human | ChatGPT | GPT4 | GPT4V |
|---|---|---|---|---|
| Human | - | 84.13% | 87.93% | 84.13% |
| ChatGPT | - | - | 90.79% | 87.30% |
| GPT4 | - | - | - | 93.33% |

Table 7: Overall conversation agreement among different judgments. "Human" denotes the human expert. "ChatGPT", "GPT4", and "GPT4V" denotes ConvBenchEval(·) using ChatGPT, GPT4, and GPT4V, respectively. In GPT4V, we do not use annotated instruction-conditioned captions but the images.

| Set up Judge | Human | ChatGPT | GPT4 | GPT4V |
|---|---|---|---|---|
| Human | - | 86.35% | 87.30% | 84.44% |
| ChatGPT | - | - | 92.06% | 87.94% |
| GPT4 | - | - | - | 93.97% |

## D    Cost Estimate for Evaluation

According to the the process of ConvBench. As shown in Table 1, a comprehensive evaluation for a model need 9 scores including $S_1$, $S_2$, $S_3$, $S_O$, $\hat{S}_2$, $\hat{S}_3$, $\hat{S}_O$, $\tilde{S}_3$, and $\tilde{S}_O$, The instruction-conditioned-caption is dense and the reference answers are annotated in detail. A comprehensive experiment need to consume about 42-52 billion tokens. In total, at current ChatGPT prices (0.5 dollars per 1M tokens), the multi-turn comparison evaluations required to assess a new model costs about 21-26 dollars. When we conduct our experiments, ChatGPT is the most cost-effective. The cost is satisfied, and thus, our evaluation is easily accessible to the community.

## E    Effectiveness for Automation Evaluation

We selected ChatGPT for evaluation in the early stages of our research to save on costs and make our evaluation framework more accessible to the community. We aim to ensure the effectiveness of the evaluation. We write high-quality instruciton-conditioned captions that include fine-grained descriptions of the image and information about the correct answers, which largely improves the performance of ChatGPT evaluation. In section C, we have assessed our automatic evaluation's agreement with human evaluation to have reached $81.83\%$, with the results being essentially indistinguishable from those of GPT-4V ($85.56\%$). Every sample's instruction-conditioned caption not only includes fine-grained descriptions of the image but also incorporates the information about the correct answers to the instructions into the caption to ensure ChatGPT has enough information to make a convincing judgment. Although ChatGPT may have difficulty detecting subtle differences between generated samples, ChatGPT combined with detailed captions acts as a more impartial and effective judge for scoring. According to the example in Figure 1 (c), with the help of the detailed caption, ChatGPT can also detect subtle differences between generated samples. We use the prompt via ChatGPT for evaluating reasoning turn in pairwise grading method.

If we only use the image caption which is not involved with the information of correct answers to instructions, the judge's result will be inaccurate. The judge will wrongly think Assistant A and Assistant B are both incorrect. The detailed judge results are as follows, where Assistant A is a LVLM, and Assistant B is human.

**Caption used in this setting**

There is a chessboard in the image. The chessboard is arranged in a total of 64 squares. In order to facilitate describing the chess game, the chessboard is marked horizontally with the letters "abcdefgh" and vertically with the numbers "12345678". For example, the lower left corner is "a1". The white King is on "e1", the white Queen is on "d1", the white Rooks are on "a1" and "h1", the white Bishops are on "c1" and "a4", the white Knights are on "b1" and "f3", the white Pawns are on "a2", "b2", "c2","e4","f2","g2", and "h2". The black King is on "e8", the black Queen is on "d8", the black Rooks are "a8" and "h8", the black Bishops are "c8" and "f8", the black Knights are "c6" and "g8", the black Pawns are "a6", "b7", "c7", "d7", "e5", "f7", "g7" and "h7".

**Evaluation Result via ConvBenchEval in this setting**

Let's break down the second-turn responses from Assistant A and Assistant B to determine which one is better.

User's Second Turn Question: "I am the player with the white chess pieces. Is there a move I can take on this turn to capture one of my opponent's pieces?"

Response from Assistant A: "Yes, as the player with the white pieces, you have a capturing move available. Your pawn on e5 can capture the black pawn on d6. This is written in algebraic notation as e5xd6."

Response from Assistant B: "Yes, you can capture your opponent's piece by moving your white knight from f3 to e5, capturing the black pawn."

Analysis: Accuracy: Assistant A: Incorrectly states that the white pawn on e5 can capture the black pawn on d6. This is incorrect based on the image context provided, as the white pawn on e5 is not in a position to capture a black pawn on d6. Assistant B: Incorrectly states that the white knight on f3 can move to e5 to capture a black pawn. However, based on the image context, there is no black pawn on e5; the black pawn is actually on e5, so this move is not possible either. Specificity: Assistant A: Provided a specific move in algebraic notation (e5xd6) but based on an inaccurate assessment of the board. Assistant B: Also provided a specific move, mentioning the knight move to e5, but again, this is not possible as described. Fluency: Both assistants provided fluent and understandable responses. However, fluency does not compensate for the lack of accuracy in both responses. Conclusion: Both Assistant A and Assistant B made errors in their responses by suggesting moves that are not possible based on the positions of the pieces described in the image context. However, Assistant B's mistake is somewhat less significant since it suggests a knight move that could be plausible with different board configurations, whereas Assistant A's move is completely off due to misidentifying the potential capture.

Overall, Response B is better despite both being incorrect, as it involves a lesser error in the interpretation of the board state.

If we use the image caption which is involved with the information of correct answers to instructions, the judge's result will be correct. The judge results are as follows, where Assistant A is a LVLM, and Assistant B is human.

**Caption used in this setting**

Caption used in this setting: There is a chessboard in the image. The chessboard is arranged in a total of 64 squares. In order to facilitate describing the chess game, the chessboard is marked horizontally with the letters "abcdefgh" and vertically with the numbers "12345678". For example, the lower left corner is "a1". The white King is on "e1", the white Queen is on "d1", the white Rooks are on "a1" and "h1", the white Bishops are on "c1" and "a4", the white Knights are on "b1" and "f3", the white Pawns are on "a2", "b2", "c2","e4","f2","g2", and "h2". The black King is on "e8", the black Queen is on "d8", the black Rooks are "a8" and "h8", the black Bishops are "c8" and "f8", the black Knights are "c6" and "g8", the black Pawns are "a6", "b7", "c7", "d7", "e5", "f7", "g7" and "h7". The player with white chess pieces can capture the opponent's piece by moving your white knight from f3 to e5, capturing the black pawn.

**Evaluation Result via ConvBenchEval in this setting**

To evaluate which second turn response better addresses the user's instruction, let's analyze both responses step by step.

The user's second instruction: "I am the player with the white chess pieces. Is there a move I can take on this turn to capture one of my opponent's pieces?"

Response from Assistant A: Content:

Assistant A suggests that the white pawn on e5 can capture the black pawn on d6, written as e5xd6. Accuracy:

This response is inaccurate based on the image context. The white pawn is not on e5; it is on e4, and there is no black pawn on d6 to capture. Relevance:

The response addresses the user's question directly but provides incorrect information. Specificity:

The move suggestion is clear but incorrect, which significantly reduces the response's value. Response from Assistant B: Content:

Assistant B correctly identifies that the white knight on f3 can capture the black pawn on e5 by moving to e5. Accuracy:

This response is accurate according to the image context. The knight on f3 can indeed capture the black pawn on e5. Relevance:

The response is directly relevant to the user's question and provides the correct move. Specificity:

The move suggestion is both clear and accurate, making it a strong response. Conclusion: Response A provides incorrect information and suggests an impossible move. Response B correctly identifies a valid capture move for the white pieces. Overall, Response B is better.

# F    Discussion on Automation for Annotation

## F.1    A Discussion on Automation for Multi-Turn Instruction Formation

It would surely be valuable to explore how to automate these processes to facilitate scaling. However, ensuring automatic annotation results of the same high quality as those of humans is undoubtedly very challenging. Our experiment results reveal LVLMs (like GPT4V) have weak performance for solving diverse and never-before-seen real-world tasks, and not to mention break down these kinds of tasks into sub-tasks (the first turn is perception, the second turn is reasoning and the third turn is creation.) Moreover, the pronouns (e.g it,they) are also needed for formatting the follow-up multi-turn instructions. Facing these challenges, there are some strategies for automating these processes to facilitate scaling: (1) GPT4V can be prompted to extend single-turn to multi-turn instructions, and human annotators rigorously filter these multi-turn instructions to form the final dataset. (2) A set of hierarchical instructions for a task (e.g board game playing) can be reused for multiple similar images. The samples in ConvBench can be used as seeds to facilitate scaling. (3) If ConvBench's method of decomposing instructions becomes a paradigm in the benchmarking field for multi-modal large models, with the power of the community, the hierarchical multi-turn instructions that are practical for different domains will be accumulated and scaled up. (4) Depending on the above collection methods of hierarchical multi-turn instructions, a high-quality pretrained dataset for extending single-turn to multi-turn instructions can also be organized. The dataset can be applied for Continue PreTraining and Supervised Finetuning on LLMs or LVLMs to obtain a large model, which can extend single-turn to multi-turn instructions automatically and effectively.

## F.2    A Discussion on Automation for Task Category Induction

Using a large language model for Task Category Induction can be a very feasible solution for automation for task category induction. However, there are still some hidden problems: (1) the results of induction via LLMs are not correct. (2) A task is summarized into two forms of expression (e.g. Travel Plan Writing and Formulating a Travel Plan), which will be represented as two kinds of tasks. Therefore, necessary manual annotations are still required to ensure the accuracy of Task Category Induction so far, and we will continue research in future work.

# G   Interfaces for Collecting Human Annotations

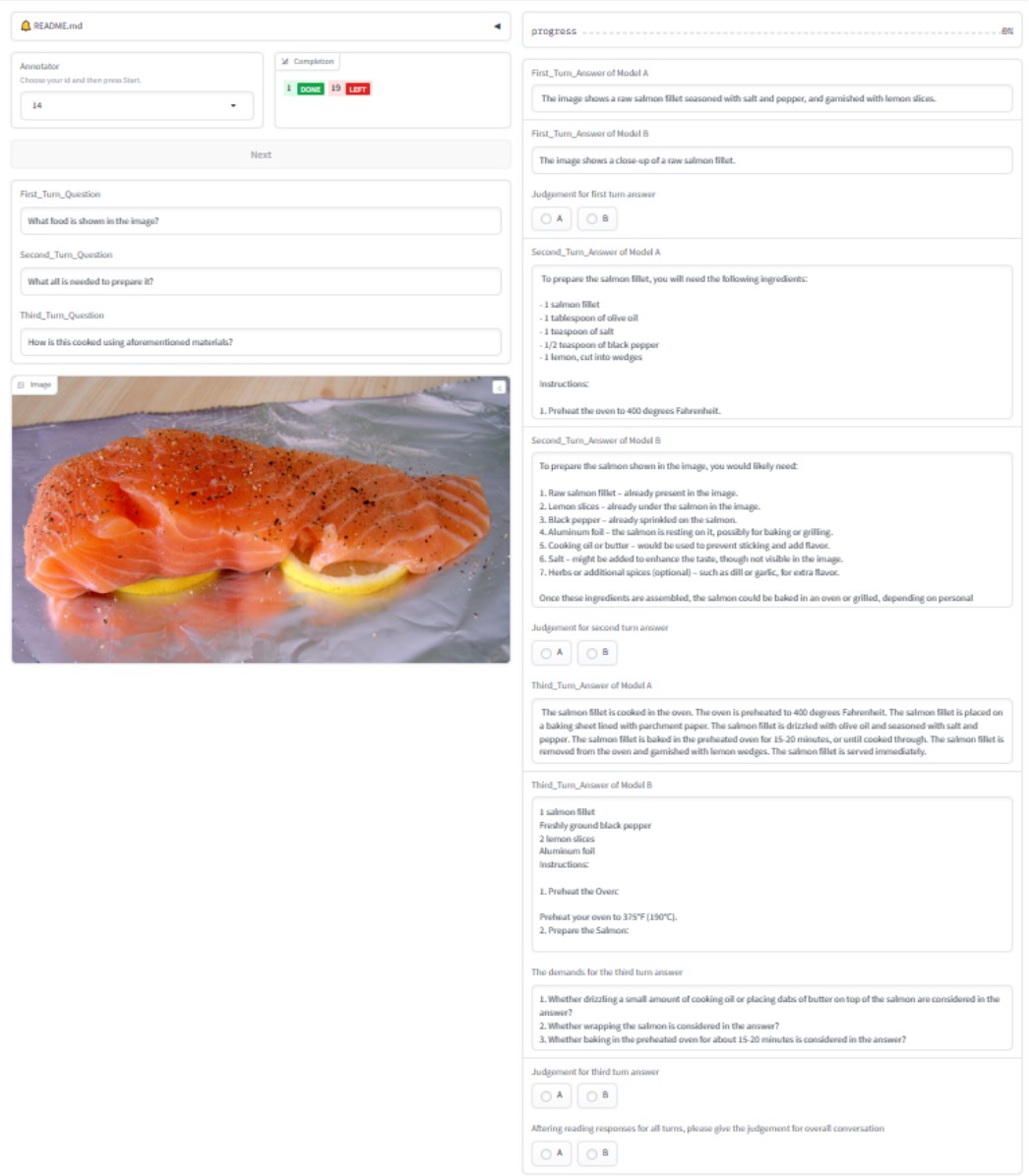

Figure 7: An interface that collects the human evaluation for evaluating the responses of LVLMs.

# H Direct Grading

We also explore the Direct Grading scheme in $\mathrm{ConvBenchEval}(\cdot)$. The direct grading will give a score of 0-10 for the model's responses using well-designed prompts.

When $\mathrm{ConvBenchEval}(\cdot)$ employs the direct grading scheme, a score of 0-10 is returned when comparing the response of the test model and reference. Unlike pairwise comparison, the response sets are now identifiable; specifically, the sets from the tested LVLM and a human participant are labeled as Assistant A's Conversation and Reference Answer, respectively. In this context, the ChatGPT judge is tasked with assigning scores directly to each turn answer and the overall conversation quality.

This prompt not only encourages the ChatGPT to engage in a chain-of-thought process but also includes the presentation of two full conversations within a single prompt. This approach is aimed at enhancing ChatGPT's ability to accurately evaluate and score the responses by maintaining a clear context and facilitating a comprehensive assessment.

As shown in Table 8, through the Direct Grading method, GPT4V achieves 7.30, 7.48, and 7.12, respectively, showing the best performance in perception, reasoning, and creation. Across 20 LVLMs under conditions of perfect perception, the average increase in reasoning and creation scores are both 1.25, via the Direct Grading approach, which shows there is still large potentials for these LVLMs to improve their perception for better reasoning and creation performance.

Comapring between Pairwise Grading and Direct Grading,the two grading strategies exhibit a high level of agreement. GPT4V ranks first and Reka second under both evaluation methodologies. Other models like InterVL-Chat, ShareGPT4V, LLaVA-V1.5, mPLUG-Owl2, Xcomposer2, and Qwen-VL-Chat also achieve high rankings. In contrast, BLIP2, XComposer, Otter, and MMAlaya are positioned lower in the rankings.

Table 8: Comparison of Performance for LVLMs on ConvBench. Quantitative ConvBench Evaluation Results for 20 LVLMs with Direct Grading method. The results in the table are the average scores of all the samples. $R_2$ is defined as $(S_1 + S_2 + S_3)/3$, indicative of the mean performance over three turns. Meanwhile, $R_1$ is computed as $(R_2 + S_O)/2$, representing the model's overall score.

| Model | $R_1$ | $R_2$ | $S_1$ | $S_2$ | $S_3$ | $S_0$ | $\hat{S}_2(\hat{S}_2 - S_2)$ | $\hat{S}_3(\hat{S}_3 - S_3)$ | $\hat{S}_O(\hat{S}_O - S_0)$ | $\tilde{S}_3(\tilde{S}_3 - \hat{S}_3)$ | $\tilde{S}_O(\tilde{S}_O - \hat{S}_O)$ |
|---|---|---|---|---|---|---|---|---|---|---|---|
| GPT4V | **7.09** | **7.30** | **7.30** | **7.48** | **7.12** | **6.88** | **8.23**(+0.75) | **8.00**(+0.88) | **8.25**(+1.37) | **7.34**(-0.66) | **8.18**(-0.07) |
| Claude | 6.54 | 6.75 | 6.53 | 7.04 | 6.68 | 6.32 | 7.48(+0.44) | 7.06(+0.38) | 7.55(+1.23) | 7.18(+0.12) | 8.13(+0.58) |
| Reka Flash | 6.78 | 6.86 | 6.93 | 7.25 | 6.41 | 6.70 | 7.10(-0.15) | 6.41(+0.00) | 7.32(+0.62) | 4.95(-1.46) | 6.95(-0.37) |
| ShareGPT4V-7B | 5.83 | 5.99 | 6.02 | 6.14 | 5.80 | 5.67 | 7.19(+1.05) | 6.77(+0.97) | 7.31(+1.64) | 6.93(+0.16) | 8.19(+0.88) |
| XComposer2 | 5.82 | 5.98 | 5.98 | 6.17 | 5.78 | 5.66 | 7.35(+1.18) | 7.04(+1.26) | 7.66(+2.00) | 7.00(-0.04) | 8.20(+0.54) |
| InternVL-Chat-V1-5 | 5.60 | 5.76 | 6.11 | 5.93 | 5.25 | 5.43 | 6.34(+0.41) | 5.60(+0.35) | 6.50(+1.07) | 6.32(+0.72) | 7.79(+1.29) |
| Qwen-VL-Chat | 5.54 | 5.65 | 5.96 | 5.78 | 5.22 | 5.43 | 7.04(+1.26) | 6.53(+1.31) | 7.26(+1.83) | 6.57(+0.04) | 8.00(+0.74) |
| InternVL-Chat-V1-2 | 5.49 | 5.69 | 5.80 | 5.88 | 5.39 | 5.29 | 6.66(+0.78) | 6.12(+0.73) | 6.75(+1.46) | 6.31(+0.19) | 7.70(+0.95) |
| LLaVA-V1.5-7B | 5.16 | 5.29 | 4.95 | 5.59 | 5.34 | 5.03 | 7.28(+1.69) | 6.68(+1.34) | 7.28(+2.25) | 6.72(+0.04) | 7.97(+0.69) |
| mPLUG-Owl2 | 5.04 | 5.17 | 4.98 | 5.38 | 5.14 | 4.91 | 6.77(+1.39) | 6.64(+1.50) | 7.22(+2.31) | 5.93(-0.71) | 7.62(+0.40) |
| LLaVA-V1.5-13B | 4.94 | 5.14 | 5.03 | 5.41 | 4.99 | 4.74 | 7.43(+2.02) | 7.13(+2.14) | 7.70(+2.95) | 6.14(-0.99) | 7.60(-0.10) |
| ShareGPT4V-13B | 4.85 | 5.03 | 5.16 | 5.06 | 4.86 | 4.67 | 7.42(+2.36) | 7.17(+2.31) | 7.65(+2.98) | 6.24(-0.93) | 7.65(+0.00) |
| LLaMA-Adapter-v2 | 4.77 | 4.91 | 4.77 | 5.47 | 4.48 | 4.64 | 6.68(+1.21) | 5.49(+1.01) | 6.68(+2.04) | 5.19(-0.30) | 7.36(+0.68) |
| Monkey | 4.49 | 4.60 | 5.11 | 4.68 | 4.01 | 4.37 | 6.28(+1.60) | 5.66(+1.65) | 6.76(+2.39) | 5.39(-0.27) | 7.30(+0.54) |
| MiniGPT4 | 3.85 | 4.04 | 3.99 | 4.40 | 3.73 | 3.66 | 6.66(+2.26) | 5.80(+2.07) | 6.75(+3.09) | 4.97(-0.83) | 7.01(+0.26) |
| MMAlaya | 3.60 | 3.75 | 4.07 | 3.91 | 3.28 | 3.44 | 5.64(+1.73) | 4.76(+1.48) | 5.91(+2.47) | 4.02(-0.74) | 6.47(+0.56) |
| Otter | 2.96 | 3.11 | 3.33 | 3.52 | 2.47 | 2.80 | 5.00(+1.48) | 4.11(+1.64) | 5.75(+2.95) | 3.25(-0.86) | 6.03(+0.28) |
| XComposer | 2.61 | 2.70 | 2.90 | 2.82 | 2.39 | 2.51 | 4.67(+1.85) | 3.84(+1.45) | 5.30(+2.79) | 3.90(+0.06) | 6.47(+1.17) |
| BLIP2-FLAN-T5-XXL | 2.37 | 2.45 | 2.81 | 2.59 | 1.95 | 2.28 | 3.18(+0.59) | 2.35(+0.40) | 4.03(+1.75) | 2.59(+0.24) | 5.75(+1.72) |
| BLIP2-FLAN-T5-XL | 2.14 | 2.21 | 2.55 | 2.33 | 1.74 | 2.07 | 2.74(+0.41) | 2.17(+0.43) | 3.81(+1.74) | 2.24(+0.07) | 5.44(+1.63) |

# I   Additional Experimental Results

## I.1   Chain of Thoughts

ConvBench can also be employed to examine studies on chain-of-thought reasoning. As illustrated in Figure 8, the single-turn approach, which involves directly requesting reasoning or creation instructions, yields inferior results compared to the multi-turn method. Figure 9 is an example in ConvBench which demonstrates how a multi-turn approach can bolster reasoning capabilities. Several studies have introduced techniques to enhance this performance. For instance, IdealGPT [46] generates sub-questions derived from the main question and responds with the corresponding sub-answers. It then analyzes the aggregate information from these sub-answers to deduce the most likely answer to the main question. Its experimental results also indicate that the chain-of-thought framework's performance surpasses that of Large Vocabulary Language Models (LVLMs) in a zero-shot context.

However, it is indeed worthwhile to investigate whether the decomposed sub-questions can contribute to solving the main question. ConvBench provides annotations to test the effectiveness of these decomposed sub-questions in answering the main question.

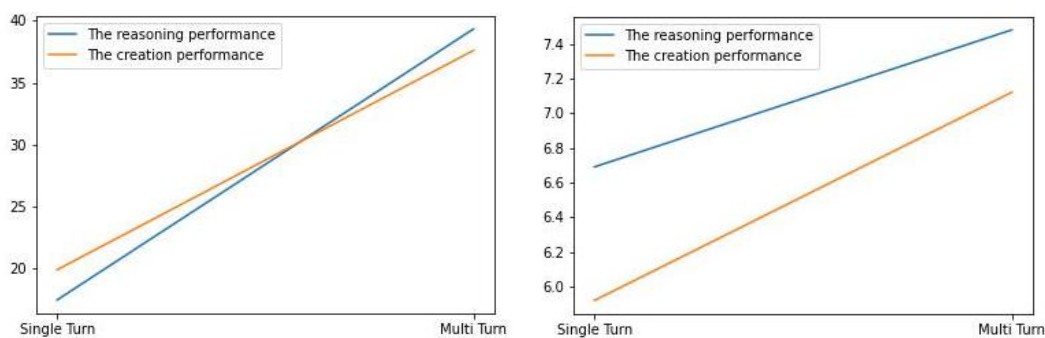

(a) Performance comparison between single-turn and multi-turn for GPT-4V with Pairwise Grading. (b) Performance comparison between single-turn and multi-turn for GPT-4V with Direct Grading.

Figure 8: The performance comparison between single-turn and multi-turn manner for GPT4V, using both Pairwise Grading method and Direct Grading method.

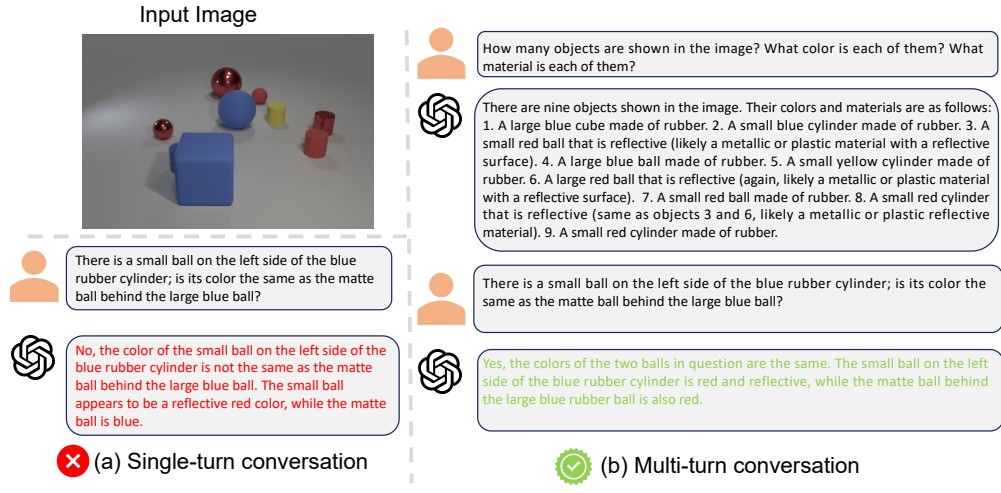

Figure 9: This example which demonstrates that a multi-turn approach can bolster reasoning capabilities.

## I.2  Ablation Studies

### I.2.1  Are Focus Demands necessary?

The creativity instruction presents an open-ended question that lacks a standard answer. We identify and annotate specific focal points related to each creative prompt to guide the evaluation process. To examine the effectiveness of these focal points, we conduct an experiment involving 50 creative instances. The experiments are divided into two groups: one that does not incorporate focal points in creation evaluations and the other that does. All other variables between the groups are kept constant. We manually assessed which one appeared more logical in each pair of evaluation outcomes from the two groups. The group using focal points achieves a 56% win rate, demonstrating the effectiveness of focal points. Figure 11 provides an example.

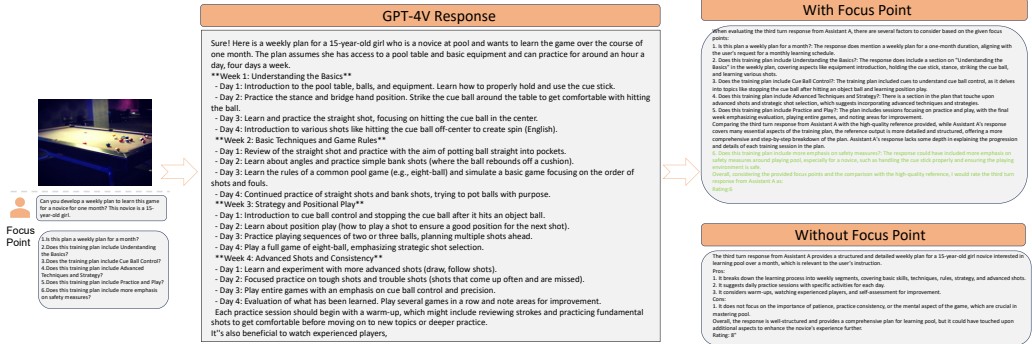

Figure 10: This example assesses the effectiveness of focus points. Provided with an image and instructions, GPT4V generates responses. The evaluation outcome incorporating focus points is deemed more appropriate compared to the result without focus points.

### I.2.2  Are the three turns' judgments helpful for the overall conversation evaluation?

We feed the perception, reasoning, and creation judgments into the final overall conversation evaluation. To examine their necessities, we conduct an experiment involving 50 instances. The experiments are divided into two groups: one that does not include three turns' judgments in overall conversation evaluations, and the other that does. All other variables between the groups are kept constant. We manually assessed which one appeared more logical in each pair of evaluation outcomes from the two groups. The group using the three turns' judgments achieves a 60% win rate. Moreover, 8% of instances fails to output a rating judgment while not using the three turns' judgments.

### I.2.3  Multi-Turn Conversation Score vs. Average Score of Three Turns

As indicated in Table 1 in the main paper, the average differences between the multi-turn conversation scores ($S_O$) and the corresponding average scores of the three turns ($R_2$) for the 20 LVLMs are 13.59 and 14.46, respectively. This suggests that the multi-turn conversation scores are generally lower than the corresponding average scores of the three turns. This discrepancy implies that the ConvBenchEval($\cdot$) evaluates more than just the performance of individual responses. The ConvBenchEval($\cdot$) can also consider the instruction-following ability in a multi-turn conversation, which is more reasonable for evaluating multi-turn visual conversation.

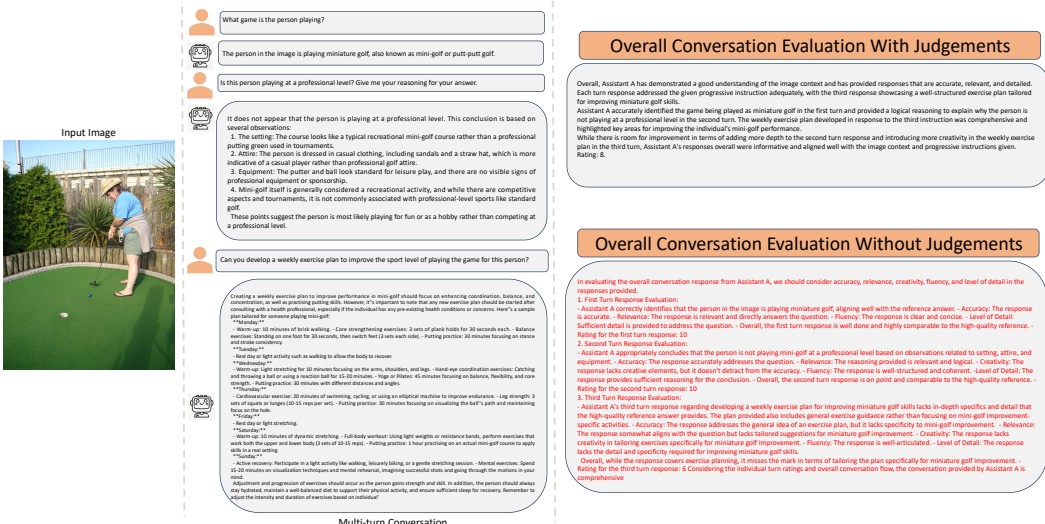

Figure 11: This example evaluates the influence of judgments made across three turns on the entire conversational process. In this case, evaluating the overall conversation without incorporating these judgments fails to yield a definitive overall rating for the overall conversation.

# J   Definition of Three Capabilities

**Perception:** This kind of task assesses an LVLM model's capability to replicate the human ability to process and extract meaningful information from visual inputs, necessitating an LVLM to have a comprehensive understanding of the image. These tasks typically involve generating text based on an image, such as celebrity recognition, food recognition, or car recognition.

**Reasoning:** This kind of task evaluates an LVLM model's ability to reason based on visual perception, demanding the application of commonsense knowledge using visual information to answer questions correctly. These tasks typically involve generating text based on an image, such as logic reasoning, meme reasoning, or visual commonsense reasoning.

**Creation:** This kind of task estimates an LVLM's ability to produce creative and diverse outputs without specific constraints or predefined templates based on the image. These tasks typically involve generating text based on an image, such as writing poems, stories, essays, or providing recommendations or suggestions.

# K   Limitation

ConvBench, while encompassing a broad array of potential applications, does not claim to include every conceivable vision-language task. Our aspiration is to progressively augment the benchmark with additional categories of tasks over time. The current iteration of our research is primarily centered on image-text modalities. Future iterations may broaden the scope to encompass other modalities such as audio and video, thereby facilitating a more holistic and comprehensive evaluation framework. Furthermore, although the dataset presents a diverse array of tasks, an increased number of examples per category could enhance the depth and richness of the evaluation. Lastly, while our GPT-based metric demonstrates a strong correlation with human judgment at both the instance level and the system level, we have observed indications that this metric may exhibit a greater predilection for outputs generated by GPT-based models compared to human preferences. Consequently, models that are trained, for instance, through the distillation of outputs from GPT4, might inadvertently gain an undue advantage within our current evaluation paradigm. This insight underscores the need for continuous refinement and calibration of our metrics to ensure equitable and unbiased assessment across a variety of model architectures and training methodologies.

## L  Ethical Discussion

ConvBench is constructed upon the robust framework of the VisIT-Bench benchmark [7], which has been meticulously vetted through extensive ethical reviews and content filtering processes to guarantee adherence to the highest ethical standards. The inception of VisIT-Bench [7] was anchored in a commitment to ethical principles, encompassing a profound respect for the consent choices of content creators and a rigorous dedication to the exclusion of inappropriate content, notably material of a pornographic nature. Leveraging this robust foundation, ConvBench introduces a suite of new annotations, each meticulously crafted and verified by human oversight. These include novel extended multi-turn instructions, refined instruction-conditioned captions, and human-verified reference answers. The benchmark is meticulously curated to eliminate any harmful content, ensuring alignment with ethical guidelines and fostering a benchmark that is not only effective but also ethically sound.

## M  Key Statistics of ConvBench

Table 9: Key statistics of ConvBench.

| Statistics | Resources |
|---|---|
| Avg. # Turns per Dialogue | 3.00 |
| Total # Dialogues | 577.00 |
| Total # Turns | 1731.00 |
| Avg. # Words in Prompt | 469.30 |
| Max. # Words in Prompt | 495.00 |
| Avg. # Words in Instruction | 83.88 |
| Max. # Words in Instruction | 930.00 |
| Avg. # Words in Instruction-Conditioned Caption | 1064.11 |
| Max. # Words in Instruction-Conditioned Caption | 4704.00 |
| Avg. # Words in Human-Verified Reference Answer | 1044.55 |
| Max. # Words in Human-Verified Reference Answer | 4704.00 |

## N  Computaional Resources

Table 10: Resource consumption of some models evaluated on ConvBench.

| Model | Resources | Times | Memory Utilization Per GPU |
|---|---|---|---|
| LLaVA-v1.5-7B | 1 x A100-80GB | 60min | 15890MiB |
| LLaVA-v1.5-13B | 1 x A100-80GB | 84min | 26717MiB |
| LLaVA-v1.5-7B | 8 x A100-80GB | 16min | 15890MiB |
| LLaVA-v1.5-13B | 8 x A100-80GB | 24min | 26708MiB |
| QWen-VL-Chat | 8 x A100-80GB | 25min | 21122MiB |
| InternVL-Chat-V1.2 | 8 x A100-80GB | 41min | 78990MiB |

The time for conducting ConvBenchEval($\cdot$) is about 4.5 hours. In Table 10, we present a detailed account of resource consumption for a selection of models. The inference times exhibit variability across different models. For instance, the compact LLaVA-v1.5-7B model, with its smaller architecture, completes the evaluation in a mere 16 minutes utilizing 8 GPUs. In contrast, the more substantial InternVL-Chat-V1.2 model demands a longer timespan of 41 minutes and approximately 80GB of memory to accomplish the same task. We use the open-source codebase (OpenCompass) which is equipped with support for multi-GPU distributed inference, which significantly expedites the inference process. This capability not only optimizes computational efficiency but also enhances the scalability of our framework, enabling researchers and practitioners to leverage the power of multiple GPUs to perform complex inferences more swiftly and effectively.

## O   Case Study

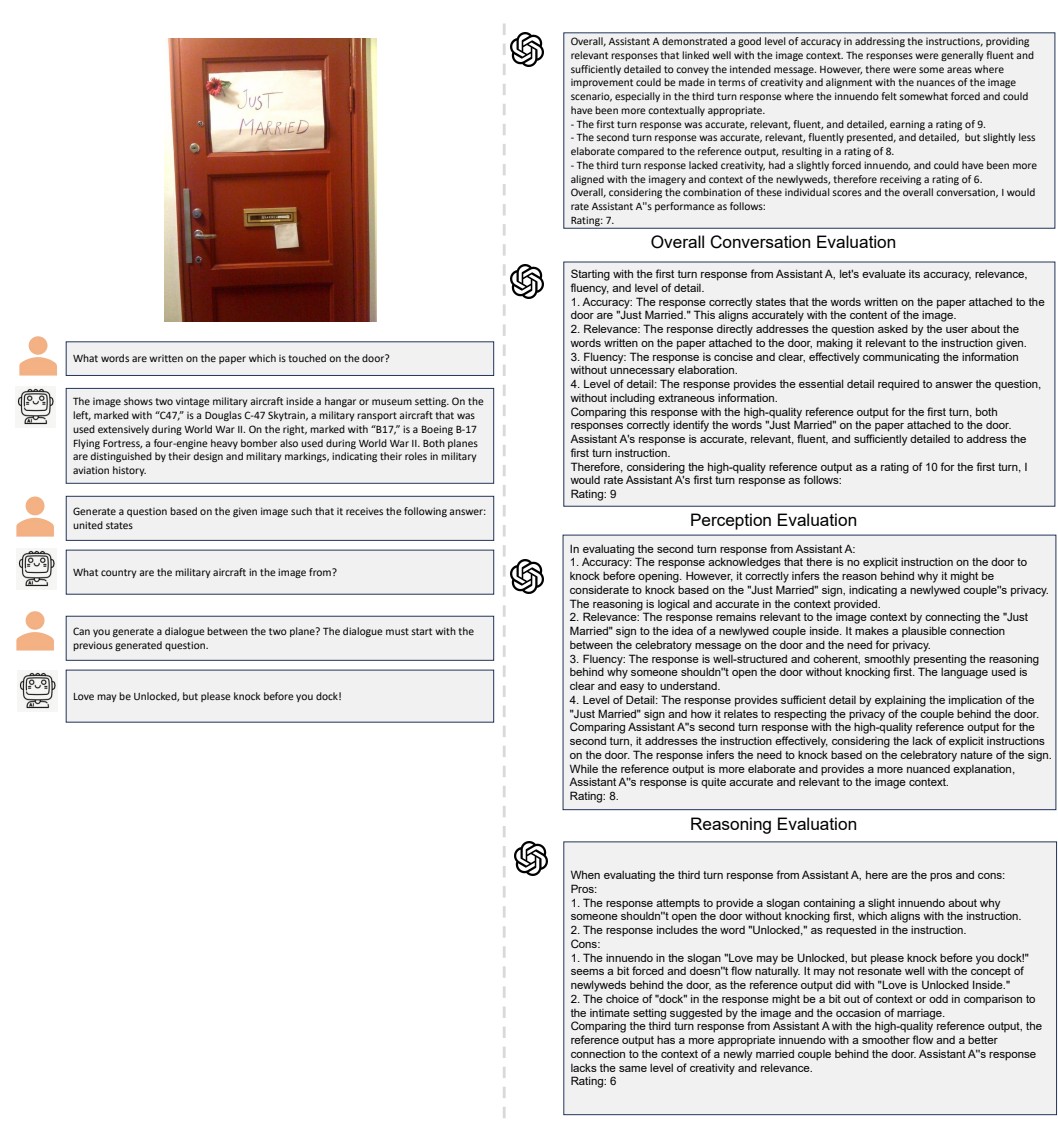

Figure 12: This example precisely interprets the text on the door without impairing further reasoning. In assessing creative performance, it is determined that the response is deficient in creativity. The performances across the three turns are considered to evaluate the multi-turn conversation's overall performance.

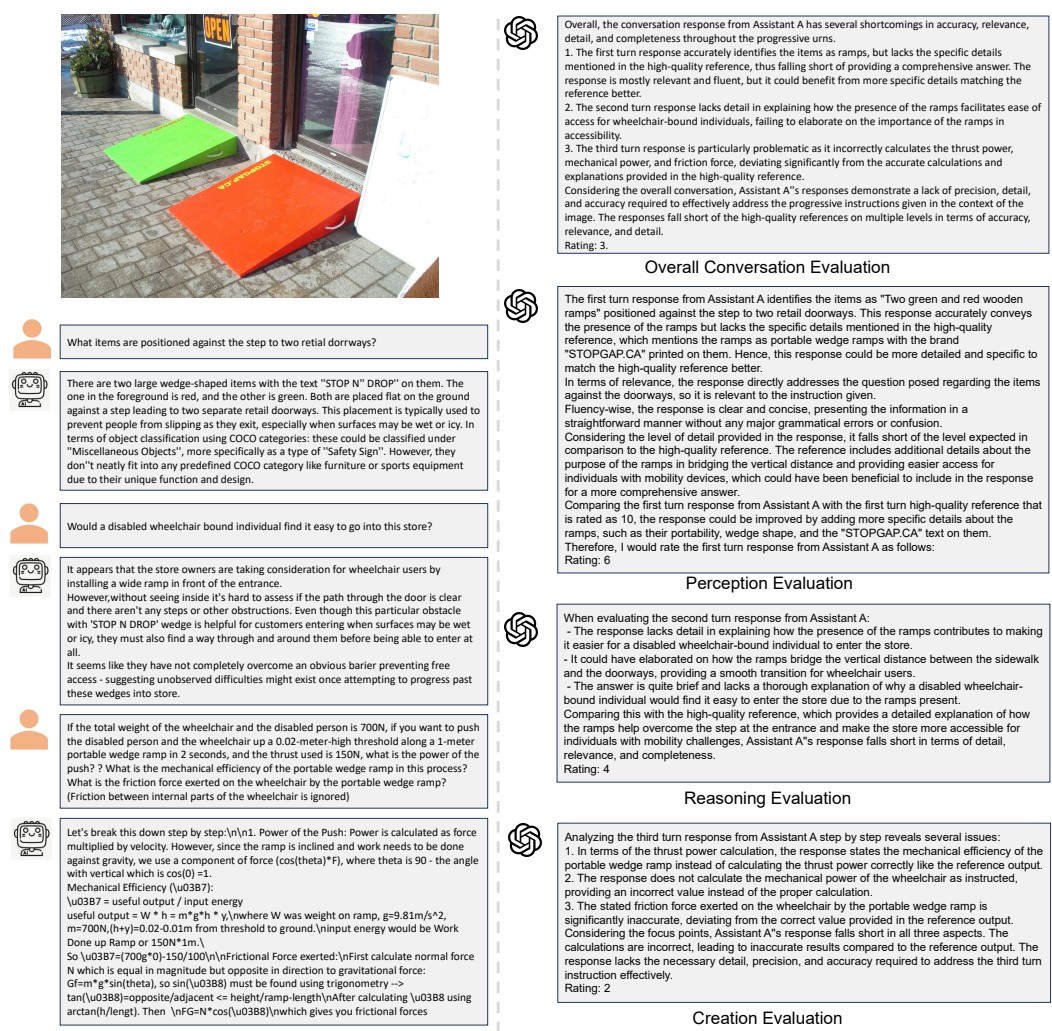

Figure 13: The accurate outcomes from the third turn include a push power of 75W, a mechanical power efficiency for the wheelchair of 9.33%, and a frictional force exerted by the wheelchair on the inclined surface of 136N. This example highlights the ongoing challenges faced by LVLMs in scientific domains.

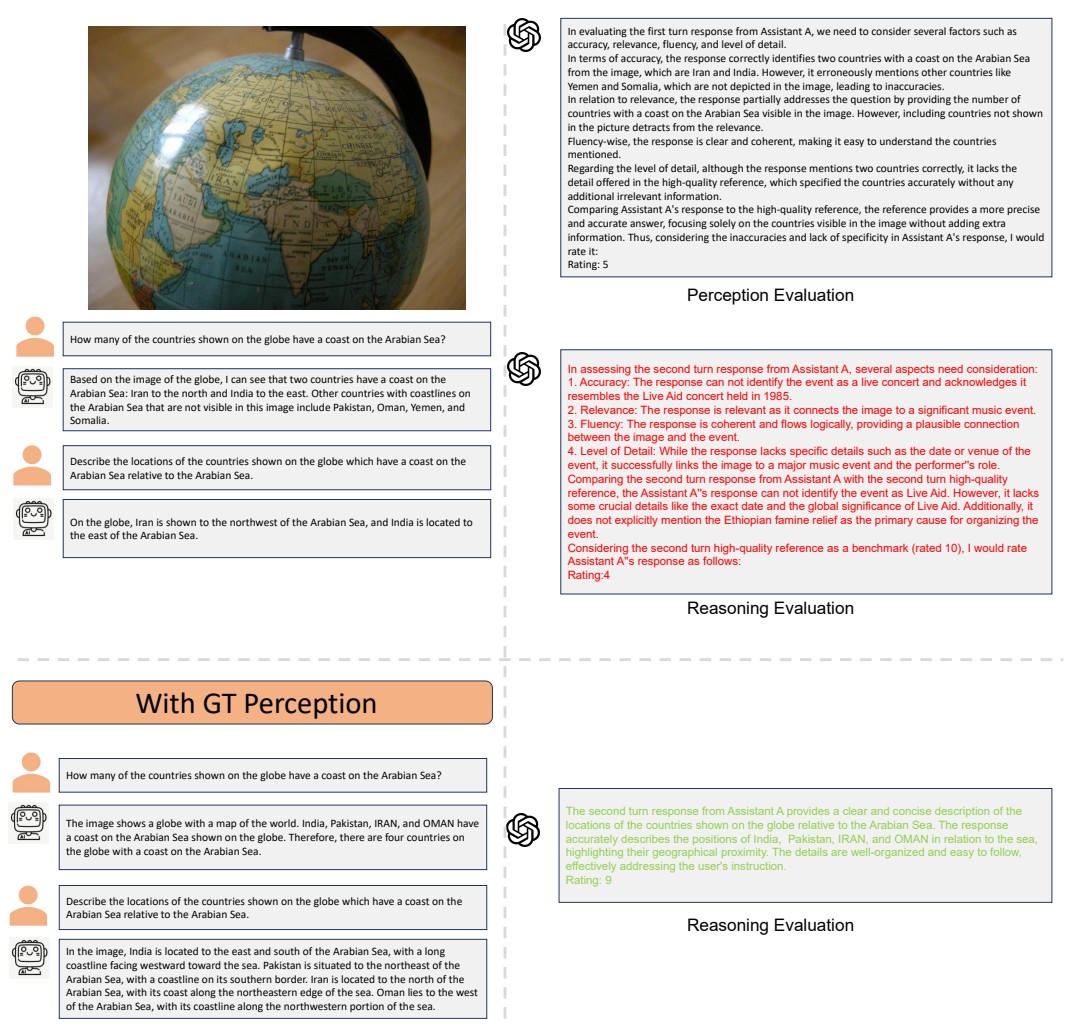

Figure 14: This example fails to offer a clear understanding of the countries surrounding the Arabian Sea, which impacts subsequent reasoning. However, when a clear perception is provided, the reasoning process is accurate. This indicates that inadequate perception can adversely affect reasoning.

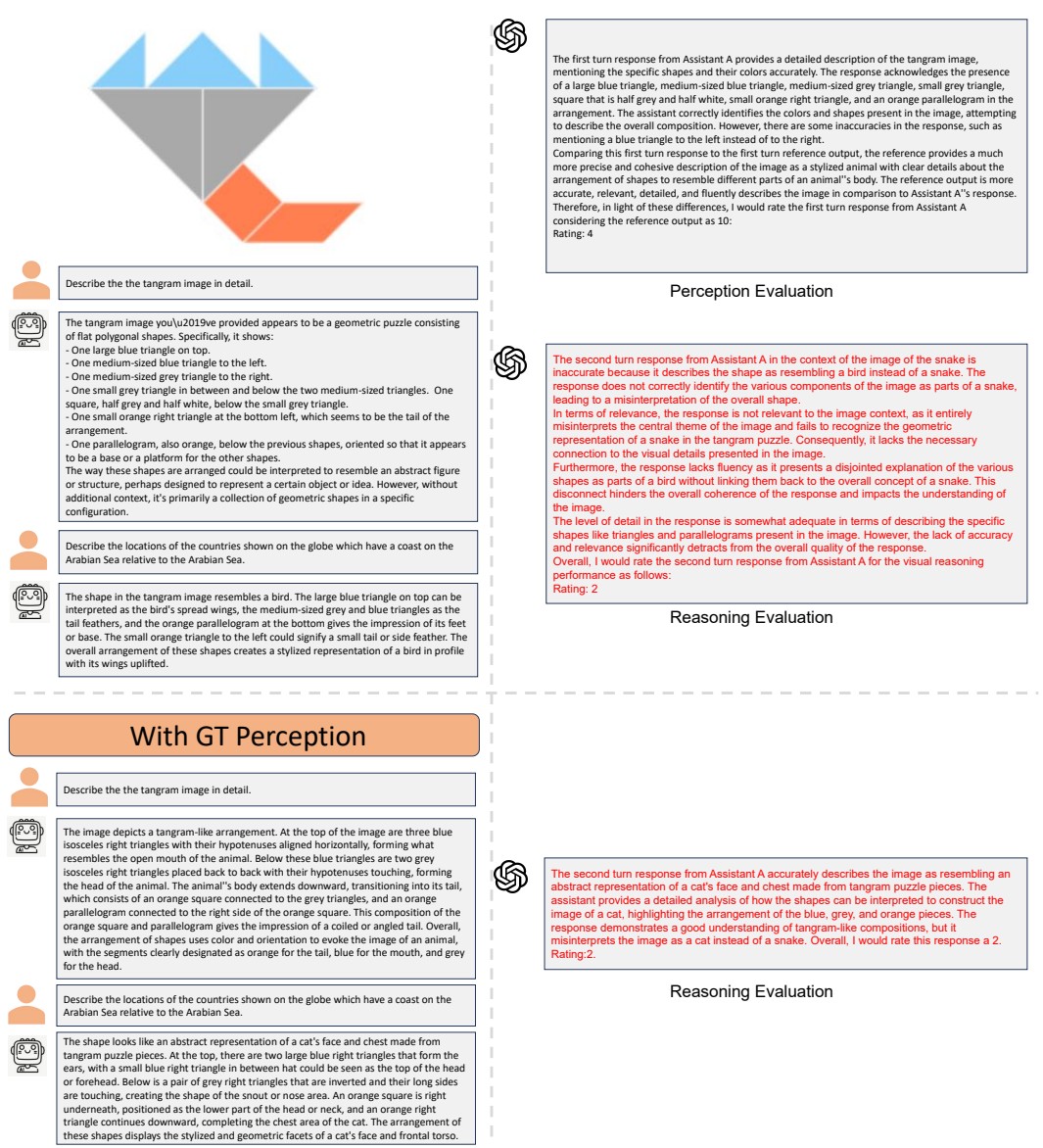

Figure 15: In this example, reasoning errors stem from the inability to accurately perceive each tangram and understand their positional relationships to one another. However, even with the provision of detailed image descriptions, the reasoning outcomes still contain errors.

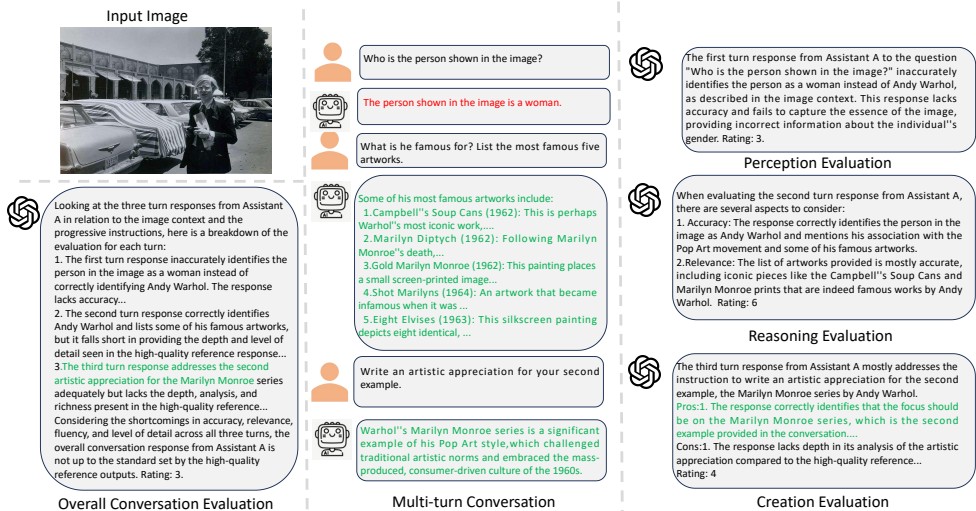

**Input Image**

**Who is the person shown in the image?**

The person shown in the image is a woman.

**What is he famous for? List the most famous five artworks.**

Some of his most famous artworks include:
1.Campbell''s Soup Cans (1962): This is perhaps Warhol''s most iconic work,....
2.Marilyn Diptych (1962): Following Marilyn Monroe''s death,...
3.Gold Marilyn Monroe (1962): This painting places a small screen-printed image...
4.Shot Marilyns (1964): An artwork that became infamous when it was ...
5.Eight Elvises (1963): This silkscreen painting depicts eight identical, ...

**Write an artistic appreciation for your second example.**

Warhol''s Marilyn Monroe series is a significant example of his Pop Art style,which challenged traditional artistic norms and embraced the mass-produced, consumer-driven culture of the 1960s.

**Multi-turn Conversation**

Looking at the three turn responses from Assistant A in relation to the image context and the progressive instructions, here is a breakdown of the evaluation for each turn:
1. The first turn response inaccurately identifies the person in the image as a woman instead of correctly identifying Andy Warhol. The response lacks accuracy...
2. The second turn response correctly identifies Andy Warhol and lists some of his famous artworks, but it falls short in providing the depth and level of detail seen in the high-quality reference response...
3.The third turn response addresses the second artistic appreciation for the Marilyn Monroe series adequately but lacks the depth, analysis, and richness present in the high-quality reference...
Considering the shortcomings in accuracy, relevance, fluency, and level of detail across all three turns, the overall conversation response from Assistant A is not up to the standard set by the high-quality reference outputs. Rating: 3.

**Overall Conversation Evaluation**

The first turn response from Assistant A to the question "Who is the person shown in the image?" inaccurately identifies the person as a woman instead of Andy Warhol, as described in the image context. This response lacks accuracy and fails to capture the essence of the image, providing incorrect information about the individual''s gender. Rating: 3.

**Perception Evaluation**

When evaluating the second turn response from Assistant A, there are several aspects to consider:
1. Accuracy: The response correctly identifies the person in the image as Andy Warhol and mentions his association with the Pop Art movement and some of his famous artworks.
2.Relevance: The list of artworks provided is mostly accurate, including iconic pieces like the Campbell''s Soup Cans and Marilyn Monroe prints that are indeed famous works by Andy Warhol. Rating: 6

**Reasoning Evaluation**

The third turn response from Assistant A mostly addresses the instruction to write an artistic appreciation for the second example, the Marilyn Monroe series by Andy Warhol.
Pros:1. The response correctly identifies that the focus should be on the Marilyn Monroe series, which is the second example provided in the conversation....
Cons:1. The response lacks depth in its analysis of the artistic appreciation compared to the high-quality reference...
Rating: 4

**Creation Evaluation**

Figure 16: This example demonstrates that the evaluation process encompasses not just the assessment of individual turns, but also the overall conversation. Specifically, it examines whether the LVLM precisely chooses the second example from the previous responses when addressing the instructions in the third turn.

# P    Prompt Templates

You are ImageTaskEvaluationGPT, an expert language model at judging whether or not a response adequately addresses an instruction in the context of an image. More specifically, you will be given the following:

1. An image context: This will describe the contents of an image with sufficient detail to address the instruction.
2. Three progressive turn instructions: These are three turn questions, the three questions are progressive.
3. Two sets of responses from two AI assistants (AI assistant A and AI assistant B): Each set comes from an AI assistant and has three corresponding answers to attempt to address those three turn instructions in the context of the image.

Your job is to judge whether the first turn response from Assistant A or the first turn response from Assitant B better. A and B are randomly ordered.

Some things to remember:

- Even though you are just a language model, the image description will be sufficiently detailed so that your judgements can be accurate.
- You should choose the assistant the follows the user's first instruction and answers the user's first question better.
- You are capable of judging response quality, accounting for important factors like correctness, relevance, fluency etc.
- Avoid any position biases and ensure that the order in which the responses were presented does not influence your evaluation.
- Do not allow the length of the responses to influence your evaluation.
- Do not favor certain names of the assistants. Be as objective as possible.
- You think step-by-step, but ultimately respond with "Response A" or  "Response B".

user (human authored)

I will describe an image to you, and provide three progressive instructions. Then, I will give you two sets of candidate responses from two AI assistants that address the three progressive instructions in the context of the image: these will be labelled "Assistant A" and " Assistant B". Your job is to first reason step-by-step about which response for the first turn instruction is better in terms of accuracy, relevance, fluency, etc. After reasoning step-by-step and comparing the pros/cons of corresponding responses for the first turn instruction, in the end, respond with "Overall, Response X is better." where X is either A or B.

assistant (human authored)

Sure, please provide the image context, the three progressive instructions, and the two sets of candidate responses. Then, I will think step-by-step and provide my ultimate judgement as to which response for the first turn instruction is better.

user (from human authored template)

OK. Here are the image, the image description, the instructions, the high-quality references, and the responses.
Image context: This is an image of a straight road in a desert-like landscape. The road is marked with double yellow lines, indicating a no-passing zone. The asphalt appears slightly worn but in generally good condition. On either side of the road, there is dry, sandy soil with sporadic, drought-resistant vegetation like small shrubs and grasses. ..

<|The Start of Assistant A's Conversation with User|>
### The first question from user:
What is the landscape and climate of this location like shown in the image?

### The first turn response from Assistant A:
The landscape in the image appears to be typical of a desert or arid region. You can infer the climate might be hot and dry based on the dry, sandy soil and presence of drought-resistant vegetation such as small shrubs and grasses that are adapted to such environments...

### The second turn question from user:
If you are driving through this area, what are vital supplies you will need? Why are these particularly important in this location?

### The second turn response from Assistant A:
When driving through a desert or arid landscape as show in the image, it is essential to carry the following supplies:
1. Water: This is the most critical supply given the high temperatures and dry conditions, which can lead to dehydration. Enough water should be carried to last the duration of the journey, plus some extra in case of emergencies...

### The third question from user:
According to previous responses, can you give me a three-day and two-night camping plan for three men and two women in this location from March 10th to March 12th, 2024.The budget is now set at $2,000.

### The third turn response from Assistant A:
Creating a camping plan for a group of five in a desert location with a budget of $2,000 over three days and two nights would involve careful planning to ensure that the group has a safe and enjoyable experience. Here\u2019s a suggested itinerary: Pre-Trip Preparations:\nGear: Rent high-quality camping gear suitable for desert conditions, including tents, sleeping bags, portable stove, cooking equipment, and a cooler...

<|The End of Assistant A's Conversation with User|>

<|The Start of Assistant B's Conversation with User|>
### The first question from user:
What is the landscape and climate of this location like shown in the image?

### The first turn response from Assistant B:
The image depicts a desert-like landscape with a straight road cutting through it. The terrain is relatively flat with sparse vegetation, consisting primarily of small shrubs and bushes typical of arid environments...

### The second turn question from user:
If you are driving through this area, what are vital supplies you will need? Why are these particularly important in this location?

### The second turn response from Assistant B:
When driving through such a desert area, some vital supplies you will need include:
1. Water: This is the most crucial supply as deserts are extremely dry and can have very high temperatures, leading to rapid dehydration...

###The third question from user:
According to previous responses, can you give me a three-day and two-night camping plan for three men and two women in this location from March 10th to March 12th, 2024.The budget is now set at $2,000.

### The third turn response from Assistant B:
With a budget of $2,000 for a three-day, two-night desert camping trip for five people, here's a suggested plan:
**Day 1: Arrival (March 10, 2024)**Morning:- Depart early morning to maximize daylight hours.- Pick up a rental 4WD vehicle if necessary (approx. $150/day). - On the way, buy supplies (food, water, fuel, etc.) - Allocate around $300 for groceries including water (approx. 1 gallon per person/day) and non-perishables. Afternoon:- Arrive at the designated camping site by early afternoon.- Set up camp with rented/bought tents and sleeping gear...

<|The End of Assistant B's Conversation with User|>

Think step-by-step, compare the first turn responses from the two assistants, and finish your response with "Overall, Response X is better." where X is either A or B.'

assistant (machine generated; used to derive the prediction)

Based on the detailed analysis provided for the first turn responses from Assistant A and Assistant B, it is clear that Assistant A's response is better. Assistant A not only accurately describes the landscape and climate of the location shown in the image but also provides a thorough explanation of the key elements depicted in the desert-like landscape. The response from Assistant A delves into the presence of desert vegetation, the condition of the road, the surrounding mountains, and the clear sky, painting a vivid picture of a hot and dry climate in a remote, isolated environment.
Therefore, for the first turn instruction, **Overall, Response A is better.

Figure 17: The prompt used for evaluating perception turn in a pairwise grading method, accompanied by a sample completion from ChatGPT, is provided. The response conversations are anonymously presented to ChatGPT, where Assistant A is identified as a human, and Assistant B is recognized as GPT4V.

## system prompt (human authored)

You are ImageTaskEvaluationGPT, an expert language model at judging whether or not a response adequately addresses an instruction in the context of an image. More specifically, you will be given the following:

1. An image context: This will describe the contents of an image with sufficient detail to address the instruction.
2. Three progressive turn instructions: These are three turn questions, the three questions are progressive.
3. Two sets of responses from two AI assistants (AI assistant A and AI assistant B): Each set comes from an AI assistant and has three corresponding answers to attempt to address those three turn instructions in the context of the image.

Your job is to judge whether the second turn response from Assistant A or the second turn response from Assitant B better. A and B are randomly ordered.

Some things to remember:

- Even though you are just a language model, the image description will be sufficiently detailed so that your judgements can be accurate.
- You should choose the assistant the follows the user's second instruction and answers the user's second question better.
- You are capable of judging response quality, accounting for important factors like correctness, relevance, fluency, specificity, etc.
- Avoid any position biases and ensure that the order in which the responses were presented does not influence your decision.
- Do not allow the length of the responses to influence your evaluation.
- Do not favor certain names of the assistants. Be as objective as possible.
- You think step-by-step, but ultimately respond with "Response A" or "Response B.

## user (human authored)

I will describe an image to you, and provide three progressive instructions. Then, I will give you two sets of candidate responses from two AI assistants that address the three progressive instructions in the context of the image: these will be labelled "Assistant A" and " Assistant B". Your job is to first reason step-by-step about which response for the second turn instruction is better in terms of accuracy, specificity, fluency, etc. After reasoning step-by-step and comparing the pros/cons of corresponding responses for the second turn instruction, in the end, respond with "Overall, Response X is better." where X is either A or B.

## assistant (human authored)

Sure, please provide the image context, the three progressive instructions, and the two sets of candidate responses. Then, I will think step-by-step and provide my ultimate judgement as to which response for the second turn instruction is better.

## user (from human authored template)

OK. Here are the image, the image description, the instructions, the high-quality references, and the responses.
Image context: There is portrait of a Native American named Geronimo. He is wearing a blue coat and a red shirt. He is also wearing a yellow tie. There is a old man in the painting. He has short ear-length hair. He is a native American. His skin is winked. He wears a red shirt inside and a blue coat outside. He is wearing a yellow scarf. "GERONIMO APACHE" and "COPYRIGHT 1898 F.A. RINEHART OMAHA" written on the painting.

<|The Start of Assistant A's Conversation with User|>
### The first turn question from user:
Who is this man shown in the image?

### The first turn response from Assistant A:
The man shown in this image represents Geronimo, who was a prominent leader and medicine man from the Apache tribe.

### The second turn question from user:
Who did this fearless man stand up to? How do we remember him today?

### The second turn response from Assistant A:
Geronimo, a fearless Native American leader,\_ stood up against both Mexican and United States forces that encroached on the lands and rights of his Apache people. He fought to protect his people's way of life, culture, and land. Today, we remember him as a symbol of resistance and a defender of indigenous culture. He is remembered today as a symbol of Native American resistance and a defender of indigenous rights...

### The third question from user:
Write an essay for an assessment of this historical figure.

### The third turn response from Assistant A:
Title: The Dialectical Evaluation of Geronimo: Resistance and Reflection in Historical Context
Introduction:
In the annals of history, few figures have embodied th spirit of resistance and the fight for cultural survival as Geronimo, the renowned Apache leader...
Thesis Statement:
Through the dialectic process, one may gain a profound understanding of Geronimo''s actions, motives, and legacy within the larger historical narrative...
Affirmation:
Geronimo''s legacy is often celebrated for his unyielding determination to resist the encroachment of foreign forces on Apache lands. He led a life committed to the...
<|The End of Assistant A's Conversation with User|>

<|The Start of Assistant B's Conversation with User|>
### The first turn question from user:
Who is this man shown in the image?

### The first turn response from Assistant B:
Sorry, I can't provide the identity of real people in images.

### The second turn question from user:
Who did this fearless man stand up to? How do we remember him today?

### The second turn response from Assistant B:
While I can''t identify the person in the image, I can provide generic historical information that might apply to an individual resembling the description of a "fearless man" from Native American history. There were numerous Native American leaders who stood up to various adversaries, including European-American settlers and the United States government during the 19th and early 20th centuries...

###The third turn question from user:
Write an essay for an assessment of this historical figure.

### The third turn response from Assistant B:
With a budget of $2,000 for a three-day, two-night desert camping trip for five people, here's a suggested plan:
Title: A Legacy of Resistance: Assessing the Impact of a Native American Leader\n\nIntroduction:
The American frontier of the 19th century was a battleground of cultures, ideologies, and survival. Among the most notable figures of this era was a legendary Apache leader\u2014a man whose name became synonymous with Native American resistance against U.S...

<|The End of Assistant B's Conversation with User|>

Think step-by-step, compare the second turn responses from the two assistants, and finish your response with "Overall, Response X is better." where X is either A or B.'

## assistant (machine generated; used to derive the prediction)

Assistant A''s response for the second turn question provides a detailed and specific overview of who Geronimo stood up against and how he is remembered today...
On the other hand, Assistant B''s response for the second turn question takes a more generic approach, not confirming the identity of the individual in the image and instead providing historical information about Native American leaders in general...
Comparing the second turn responses, Assistant A''s response is more accurate, specific, and directly relevant to the image context and user''s question. Therefore, in terms of accuracy, specificity, and relevance to the given image description, Assistant A''s response for the second turn instruction is better.

Overall, Response A is better.

Figure 18: The prompt used for evaluating reasoning turn in a pairwise grading method, accompanied by a sample completion from ChatGPT, is provided. The response conversations are anonymously presented to ChatGPT, where Assistant A is identified as a human, and Assistant B is recognized as GPT4V.

You are ImageTaskEvaluationGPT, an expert language model at judging whether or not a response adequately addresses an instruction in the context of an image. More specifically, you will be given the following:

1. An image context: This will describe the contents of an image with sufficient detail to address the instruction.
2. Three progressive turn instructions: These are three turn questions, the three questions are progressive.
3. Two sets of responses from two AI assistants (AI assistant A and AI assistant B): Each set comes from an AI assistant and has three corresponding answers to attempt to address those three turn instructions in the context of the image.
4. Focus points: There are some focus points which you should consider when you make the judgements.

Your job is to judge whether the third turn response from Assistant A or the third turn response from Assitant B better. A and B are randomly ordered.

Some things to remember:

- Even though you are just a language model, the image description will be sufficiently detailed so that your judgements can be accurate.
- You should choose the assistant the follows the user's third instruction and answers the user's third question better.
- You are capable of judging response quality, accounting for important factors like correctness, relevance, fluency, specificity, etc.
- Avoid any position biases and ensure that the order in which the responses were presented does not influence your decision.
- Do not allow the length of the responses to influence your evaluation.
- Do not favor certain names of the assistants. Be as objective as possible.
- You think step-by-step, but ultimately respond with "Response A" or "Response B- You think step-by-step, but ultimately respond with "Response A" or "Response B.

user (human authored)

I will describe an image to you, and provide three progressive instructions. Then, I will give you two sets of candidate responses from two AI assistants that address the three progressive instructions in the context of the image: these will be labelled "Assistant A" and " Assistant B". Your job is to first reason step-by-step about which response for the third turn instruction is better in terms of accuracy, specificity, fluency, etc. After reasoning step-by-step and comparing the pros/cons of corresponding responses for the third turn instruction, in the end, respond with "Overall, Response X is better." where X is either A or B.

assistant (human authored)

Sure, please provide the image context, the three progressive instructions, and the two sets of candidate responses. Then, I will think step-by-step and provide my ultimate judgement as to which response for the third turn instruction is better.

user (from human authored template)

OK. Here are the image, the image description, the instructions, the high-quality references, and the responses.
Image context: In this photo, the sun is setting which casts an orange huge against the black tree-line. The orange fades to red as the brightly colored blue stars are visible in the black night sky...

<|The Start of Assistant A's Conversation with User|>
### The first turn question from user:
What are the bright dots? What is the red hue?

### The first turn response from Assistant A:
The bright dots in the image are stars scattered across the night sky. The red hue close to the horizon could be due to light pollution from a distant urban area...

### The second turn question from user:
Why does it look like that, what causes the bright dots, what causes the red hue?

### The second turn response from Assistant A:
The bright dots in the image are caused by the light from distant stars reaching the camera sensor or film...

### The third question from user:
Write a romantic poem for explaining the previous causes.

### The third turn response from Assistant A:
Beneath the celestial tapestry so wide and deep,
Stars whispersecrets that the night does keep.
Each a tiny heartbeat in the cosmos' breast,
Light-years filled with love's soft cadence, unrest...
<|The End of Assistant A's Conversation with User|>

<|The Start of Assistant B's Conversation with User|>
### The first turn question from user:
What are the bright dots? What is the red hue?

### The first turn response from Assistant B:
The bright dots are stars. The red hue is the afterglow of the setting sun. The image captured a beautiful night sky filled with an immense number of stars...

### The second turn question from user:
Why does it look like that, what causes the bright dots, what causes the red hue?

### The second turn response from Assistant B:
The image looks like that due to the scattering of sunlight by the Earth's atmosphere during sunset, which causes the orange and red hues...

###The third turn question from user:
Write a romantic poem for explaining the previous causes.

### The third turn response from Assistant B:
In the tender clutch of evening''s soft descent,
Where the sun kisses the horizon, and day is spent,
The sky, a canvas vast, begins its nightly hue,
A ballet of light where orange fades to red anew...
<|The End of Assistant B's Conversation with User|>

There are some concerns which you should focus when make your judgements for the response:
1. Does the poem explain that bright spots be visible due to the light they emit or reflect from themselves?
2. Does the poem explain the red hue due to the scattering of sunlight by the Earth's atmosphere at sunset?
3. Is this poem romantic?'

Think step-by-step, compare the third turn responses from the two assistants, and finish your response with "Overall, Response X is better." where X is either A or B.'

assistant (machine generated; used to derive the prediction)

When comparing the third turn responses from Assistant A and Assistant B, let's consider the following points:
1. **Accuracy and Relevance**:
 - Assistant A's response beautifully integrates the concepts of stars, their distant nature...
Assistant B's response also includes romantic elements and metaphorical language to describe the stars...
2. **Fluency and Poetic Quality**:
- Assistant A's poem flows well, utilizing poetic language effectively...
- Assistant B's poem also maintains a poetic tone..
3. **Tie Back to Previous Explanations**:
- Both responses successfully tie the romantic poem back to the scientific explanations provided earlier about stars, light pollution, and the red hue, offering a creative and artistic interpretation of those concepts.\n\nBased on the above considerations,
**Overall, Response A is better**. Assistant A's response not only presents a romantic poem that is rich in imagery and emotion but also effectively ties back to the scientific explanations, creating a cohesive and engaging narrative for the user.

Figure 19: The prompt used for evaluating creation turn in a pairwise grading method, accompanied by a sample completion from ChatGPT, is provided. The response conversations are anonymously presented to ChatGPT, where Assistant A is identified as GPT4V, and Assistant B is recognized as a human.

Figure 20: The prompt used for evaluating overall conversation in a pairwise grading method, accompanied by a sample completion from ChatGPT, is provided. The response conversations are anonymously presented to ChatGPT, where Assistant A is identified as a human, and Assistant B is recognized as GPT4V.

Figure 21: The prompt used for evaluating perception in a direct grading method, accompanied by a sample completion from ChatGPT, is provided.

Figure 22: The prompt used for evaluating reasoning in a direct grading method, accompanied by a sample completion from ChatGPT, is provided.

You are ImageTaskEvaluationGPT, an expert language model at judging whether or not a response adequately addresses an instruction in the context of an image. More specifically, you will be given the following:

1. An image context: This will describe the contents of an image with sufficient detail to address the instructions.
2. Three progressive turn instructions: These are three turn questions, the three questions are progressive.
3. Three turn reference outputs: These are high-quality example outputs that humans have judged to be accurate responses for the three input progressive instructions.
4. Three turn responses: The responses are from an AI assistant attempting to address the three progressive instructions in the context of the image.
5. Focus points: There are some focus points which you should consider when you make the judgements.

Your job is to rate the third turn response from the AI assistant for the composition on a scale of 1 to 10. Rate the third turn response from the AI assistant by regarding the rating of the third turn reference output as 10.

Some things to remember:

- Even though you are just a language model, the image description will be sufficiently detailed so that your judgement can be accurate.
- Regard the ratings of the high-quality references as 10. Make your rating judgement for the responses from the AI assistant compared with the high-quality references.
- You are capable of judging responses quality. Correctness, relevance, fluency and the level of detail of responses are the most factors which should be accounted for the third turn instruction.
- You think step-by-step and be as objective as possible, after providing your explanation, you must rate the third turn response for the composition on a scale of 1 to 10 by strictly following this format:"Rating:{rating}", for example: "Rating:{5}".

## user (human authored)

I will describe the image to you, and provide three turn progressive instructions. Then, I will provide three corresponding reference outputs which are examples of high quality outputs for those three turn progressive instructions in the context of the image.
Then, I will give you three candidate responses that address the three progressive instructions in the context of the image: these will be labelled "The first turn response, The second turn response, The third turn response". Your job is to first reason step-by-step about the procs/cons of the third turn candidate response in terms of accuracy, relevance, creativity, fluency, the level of detail of responses etc. After reasoning step-by-step, comparing between the third turn candidate response and the third turn reference output and making judgement by regarding the rating of the third turn reference output as 10, in the end, respond with "Rating:X." where X is a scale of 1 to 10.

## assistant (human authored)

Sure, please provide the image context, the three instructions, the three reference outputs, and the three candidate responses, the focus points. Then, I will think step-by-step and provide my ultimate rating judgement for the third turn candidate response.

## user (from human authored template)

OK. Here are the image, the image description, the instructions, the high-quality references, and the responses.
Image context: The image shows a model of a molecule represented as functional groups with sulfur compounds sulfone sulfonic acid sulfonamide. There are three parts in the molecule image. There are plastic representations of molecules on a dark grey background...
<The Start of Reference Answer|>
### The first turn question from user:
What is the compound shown in the image?

### The first turn high quality reference:
The image provided shows models of molecules typically used for educational purposes to demonstrate molecular structure. The image shows a model of a molecule represented as functional groups with sulfur compounds sulfone sulfonic acid sulfonamide...

### The second turn question from user:
What is the connection between this compound and flatulence?

### The second turn high quality reference:
The molecules depicted in the image are indeed models of sulfur-containing compounds, which can be related to flatulence due to the production of sulfur-bearing gases during digestion. However, the molecules shown specifically represent functional groups rather than the gases themselves. Here is a clarification of the connection:

### The third question from user:
What other side effect does the compound which produces flatulence have. Pretend yourself to be a medical student.

### The third turn high quality reference:
Sulfonamides, also known as sulfa drugs, are a class of antimicrobial agents that are widely used in the treatment of bacterial infections. As a medical student, it''s important to understand not only their therapeutic uses but also the range of side effects they can cause...
<|The End of Reference Answer|>

<|The Start of Assistant A's Conversation with User|>
### The first turn question from user:
What is the compound shown in the image?

### The first turn response from Assistant A:
The image shows molecular models of water ($H_2O$) and ammonia ($NH_3$). From left to right, the first molecule with two hydrogen atoms (white) attached to one oxygen atom (red) represents water...

### The second turn question from user:
What is the connection between this compound and flatulence?

### The second turn response from Assistant A:
The compound ammonia ($NH_3$), as represented by two of the molecular models in the image (with nitrogen in blue and hydrogen in white), is relevant to flatulence because it is one of the compounds that can be produced during the digestion process. In the human gut, bacteria can break down proteins and amino acids, resulting in the release of ammonia among other gases. ..

### The third question from user:
What other side effect does the compound which produces flatulence have. Pretend yourself to be a medical student.

### The third turn response from Assistant A:
As a medical student, you would understand that ammonia ($NH_3$), which is produced in the gut during protein digestion, not only contributes to flatulence but also to other physiological processes. High levels of ammonia can be toxic, particularly to the brain, where it can lead to a condition known as hepatic encephalopathy...
<|The End of Assistant A's Conversation with User|>

There are some concerns which you should focus when make your judgements for the response
1. Whether allergic reactions are mentioned in the answer?
2. Whether renal effects are mentioned in the answer?
3. Whether central nervous system effects are mentioned in the answer?:
Think step-by-step, rate the third turn response from Assistant A, and finish your response with "Rating:X." where X is all a scale of 1 to 10.

## assistant (machine generated; used to derive the prediction)

In analyzing the third turn response from Assistant A, let''s break down the key points:
1. **Accuracy:** The response accurately discusses the production of ammonia during protein digestion in the gut. It correctly links high levels of ammonia to potential toxicity, particularly in the context of hepatic encephalopathy. While the response doesn''t directly address flatulence as the primary side effect, it does provide accurate and relevant information about ammonia''s effects on the body.
2. **Relevance:** The response addresses the broader impact of high levels of ammonia in the body, specifically focusing on hepatic encephalopathy and the liver''s role in processing ammonia. While it doesn''t directly
3. **Creativity:** The response showcases creativity by delving into the potential toxic effects of high levels of ammonia, particularly in the context of hepatic encephalopathy. It demonstrates a deeper understanding of the
4. **Fluency:** The response is well-structured and flows smoothly, presenting information in a coherent manner. The use of technical terms like hepatic encephalopathy and ammonia metabolism ..
5. **Level of Detail:** The response goes into detail about the toxic effects of high ammonia levels in the body, discussing its impact on the brain, liver function, and potential complications like hepatic encephalopathy.
Comparing the third turn response from Assistant A with the high-quality reference output, Assistant A provides accurate and detailed information about the effects of high levels of ammonia in the body. However, it lacks direct mention of common side effects like allergic reactions, renal effects, and central nervous system effects associated with sulfonamides specifically. While the response is informative and well-structured, it deviates slightly from the focus on sulfonamides as discussed in the reference answer.
Considering these points, I would rate the third turn response from Assistant A as follows:
Rating: 7

Figure 23: The prompt used for evaluating creation in a direct grading method, accompanied by a sample completion from ChatGPT, is provided.

You are ImageTaskEvaluationGPT, an expert language model at judging the multi-turn conversation instruction-following ability of an AI assistant. More specifically, you will be given the following:

1. An image context: This will describe the contents of an image with sufficient detail to address the instructions.
2. Three progressive turn instruction: These are three turn questions, the three questions are progressive.
3. Three reference outputs: These are high-quality example outputs that humans have judged to be accurate responses for these input progressive instructions.
4. Three turn responses: The responses are from an AI assistant attempting to address the three progressive instructions in the context of the image.
5. Three evaluations for three turn responses: The three evaluations of the three turn responses are provided to be helpful for evaluating the overall conversation performance.

Your job is to rate the overall conversation on a scale of 1 to 10. Rate the overall conversation by regarding the ratings of the reference outputs as 10.

Some things to remember:

- Even though you are just a language model, the image description will be sufficiently detailed so that your judgements can be accurate.
- Regard the ratings of the high-quality references as 10. Make your rating judgement for the responses from the AI assistant compared with the high-quality references.
- Take the rating of each turn into account when making your judgements for the overall conversation.
- You are capable of judging overall conversation quality, accounting for the multi-turn conversation and instruction-following ability. Correctness, relevance, fluency and the level of detail of responses are the most factors which should be considered.
- As for the open-ended instructions, creativity, helpfulness, specificity and level of detail of responses should also be considered.
- You think step-by-step and be as objective as possible, after providing your explanation, you must rate the overall conversation on a scale of 1 to 10 by strictly following this format:"Rating: {rating}", for example: "Rating: 3.".

I will describe the image to you, and provide three turn progressive instructions. Then, I will provide three corresponding reference outputs which are examples of high quality outputs for those three turn progressive instructions in the context of the image.
Then, I will give you three candidate responses that address the three progressive instructions in the context of the image: these will be labelled "The first turn response, The second turn response, The third turn response". Your job is to first reason step-by-step about the procs/cons of the overall conversation in terms of accuracy, relevance, creativity, fluency, the level of detail of responses etc. After reasoning step-by-step, comparing between the each turn candidate response and the each turn reference output and making judgement by regarding the rating of the each turn reference output as 10, in the end, respond with "Rating:X" where X is a scale of 1 to 10.

Sure, please provide the image context, the three instructions, the reference outputs, the candidate responses and the three evaluations for the three turn responses. Then, I will think step-by-step and provide my ultimate rating judgement for the overall conversation.

OK. Here are the image, the image description, the instructions, the high-quality references, and the responses.
Image context: There is are two shelves with a slightly reflective surface, which are each in a separate area of the cabinet. On the left of the image, there are three columns of round white medicine bottles...

<The Start of Reference Answer|>
### The first turn question from user:
Describe the image in detail focusing on the words.

### The first turn high quality reference:
There is are two shelves with a slightly reflective surface, which are each in a separate area of the cabinet. On the left of the image, there are three columns of round white medicinal bottles. Each bottle has the label 'stress killers 200pcs' written on the front in black text...

### The second turn question from user:
What do these two medical products do?

### The second turn high quality reference:
Based on the labels, the products are suggested to serve the following purposes: the bottles labeled "stress killers" imply that they are intended to eliminate or reduce stress, and the cans labeled "inner calmness" suggest that they are meant to promote or enhance a sense of calm and tranquility within an individual...

### The third question from user:
Come up with a short, catchy ad title for the two products displayed in the cabinet, including the efficacy of these two medicines.

### The third turn high quality reference:
Conquer Stress, Embrace Serenity: Unleash the power of 'Stress Killers' and 'Inner Calmness' for a peaceful you!\_ Your Ultimate Wellness Duo!\"...
<|The End of Reference Answer|>

<|The Start of Assistant A's Conversation with User|>
### The first turn question from user:
Describe the image in detail focusing on the words.

### The first turn response from Assistant A:
The image displays two shelves containing products with labels that suggest they are related to stress management and well-being. On the left side of the image, there are white bottles with labels that read "stress killers." ...

### The second turn question from user:
What do these two medical products do?

### The second turn response from Assistant A:
Based on the image, the products are labeled in a way that suggests they may be designed to alleviate stress. The white bottles labeled "stress killers" imply that they might be used to help reduce or eliminate stress, while the silver cans labeled "inner calmness" suggest they could be intended to promote a feeling of calm or well-being...

### The third question from user:
Come up with a short, catchy ad title for the two products displayed in the cabinet, including the efficacy of these two medicines.

### The third turn response from Assistant A:
Conquer Stress and Embrace Serenity: Find Your Inner Calmness and Slay Stress with Every Dose!...
<|The End of Assistant A's Conversation with User|>

The first turn evaluation: The first turn response from Assistant A provides a thorough and detailed description of the image, focusing on the words featured on the products. The assistant accurately captures the details of the image, mentioning the labels "stress killers" and "inner calmness," along with the quantity and weight specifications on the labels...
The second turn evaluation: Firstly, let''s evaluate the second turn response from Assistant A in terms of accuracy, relevance, fluency, and level of detail compared to the high-quality second turn reference output...
The third turn evaluation: Upon examining the third turn response from Assistant A, here is a step-by-step analysis:
1. **Relevance to the instruction:** - The response does come up with a catchy ad title for the two products, which alignswith the instruction given. It emphasizes stress relief and serenity, as requested. It is relevant in addressing the task at hand...
Think step-by-step, rate the overall conversation from Assistant A, and finish your response with "Rating:X." where X is all a scale of 1 to 10.

Overall, Assistant A demonstrated a strong understanding of the image content and responded well to the three progressive turn instructions. The responses were accurate, relevant, and provided a good level of detail throughout the conversation.
In the first turn, Assistant A accurately described the image, focusing on the words present on the products, which was similar to the high-quality reference output. The response was detailed and effectively conveyed the information in the image.
In the second turn, Assistant A correctly interpreted the potential functions of the medical products based on their labels. The response was relevant, cautious in its interpretation, and maintained fluency in explaining the implications of the labels.
For the third turn, Assistant A came up with a catchy ad title that highlighted the efficacy of the two medicines in stress relief and serenity. While the response was creative, it lacked specificity in mentioning the product names, which could have enhanced the impact of the advertisement.
Overall, Assistant A's responses were consistently accurate and relevant, with good fluency and detail. The conversation flowed logically and maintained a focus on the image content and instructions provided.
Rating: 8

Figure 24: The prompt used for evaluating overall conversation in a direct grading method, accompanied by a sample completion from ChatGPT, is provided.

- Keep your answers short: only output "Final Answer: X" where X is A or B
- If the explanation doesn't mention a final choice explicitly, you can output "Unknown"

**user (human authored)**

Step-by-step reasoning:

- First, I consider the instruction, which asks if anything else in the image has the same color as the tiny sphere.
- Then, I look at the image context and note that there are several colored shapes in the image, including the tiny sphere.
- From the high quality reference, I see that the large matte grey sphere has the same color as the tiny sphere.
- Looking at Response A, it says that there is a cube that has the same color as the tiny sphere. But the high quality reference only mentions the large matte grey sphere having the same color, not the cube. Therefore, Response A is not accurate.
- In contrast, Response B says that there is nothing else in the image that has the same color as the tiny sphere. The high quality reference contradicts this response, stating that the large matte grey sphere shares the same color with the tiny sphere.
- Therefore, Response B is not accurate or correct.
- Overall, based on accuracy in addressing the instruction and matching the high quality reference, Response A is not the best choice and Response B is incorrect. Therefore, the better response is Response B: "No, there is nothing else in the image that has the same color as the tiny sphere.
Please extract the final answer from the above text.

**assistant (human authored)**

Final Answer: Response B

**user (human authored)**

Let's compare the two responses step-by-step:

Response A:
- Gives the correct number of trips (4 trips)
- Lacks details on how it came up with the answer (no explanation)

Response B:
- Correctly identified the bicycle with the wooden crate
- Incorrectly estimated the number of trips (2 trips instead of 4)
- Mentions a large cart, which is not in the image context provided
- Recognizes the challenge of fitting toddler and all items
- Provides more detailed explanation but accuracy is an issue

Considering accuracy, specificity, fluency, and relevance, let's make a judgement:

Response B is better in terms of fluency and relevance while Response A is better in terms of accuracy and specificity. Since accuracy is crucial in this case, and given the high-quality reference backs the number of trips (4), we prioritize correctness.

Overall, I choose A.
Please extract the final answer from the above text.

**assistant (human authored)**

Final Answer: Response A

**user (from human authored)**

Upon analyzing the image description and the instruction, it is clear that the instruction is asking for the implied meaning of a comment received by the creator of the burnt pizza in the image. The high-quality reference suggests that the compliment of "being very talented" is actually sarcastic and that the burnt pizza is not a sign of talent. Looking at Response A, it seems like the response is describing the pizza and the plate without really addressing the implication of the comment. Response B, on the other hand, does directly address the instruction and implies that the praise is sincere even though the pizza is burnt. Therefore, overall, Response B is better.
Please extract the final answer from the above text.

**assistant (human authored)**

Final Answer: Response B

Figure 25: The prompt used for extracting answers from ill-formatted ChatGPT responses in a pairwise grading method.

Figure 26: The prompt used for extracting answers from ill-formatted ChatGPT responses in a direct grading method.

# Q  Datasheet

**Motivation**

**For what purpose was the dataset created?** Was there a specific task in mind? Was there a specific gap that needed to be filled? Please provide a description.

The dataset was created to enable research on multi-turn visual conversation evaluation for large vision-language models (LVLMs)-i.e., given an image and three-turn instructions, evaluate whether the responses from LVLms reach human preferences. The benchmark organization can also provide hierarchical ablation evaluation for error analysis. ConvBench progressively examines the LVLMs' perception, reasoning, and creativity capabilities in each conversation and can decouple these capabilities in evaluations and thus perform reliable error attribution.

**Who created this dataset (e.g., which team, research group) and on behalf of which entity (e.g., company, institution, organization)?**

The dataset was created by Shuo Liu, Wenqi Shao, and Kaipeng Zhang at Shanghai AI Lab.

**Who funded the creation of the dataset?** If there is an associated grant, please provide the name of the grantor and the grant name and number.

This dataset is partially supported by the National Key R&D Program of China No.2022ZD0161000 and the General Research Fund of Hong Kong No.17200622 and 17209324.

**Any other comments?**

None.

**Composition**

**What do the instances that comprise the dataset represent (e.g., documents, photos, people, countries)?** Are there multiple types of instances (e.g., movies, users, and ratings; people and interactions between them; nodes and edges)? Please provide a description.

An instance in ConvBench is an image, three sequential instructions, three task categories, an instruction-conditioned caption, three-turn human-verified reference answers, and focus demands for the creation instruction. An example instance is shown in Figure 27.

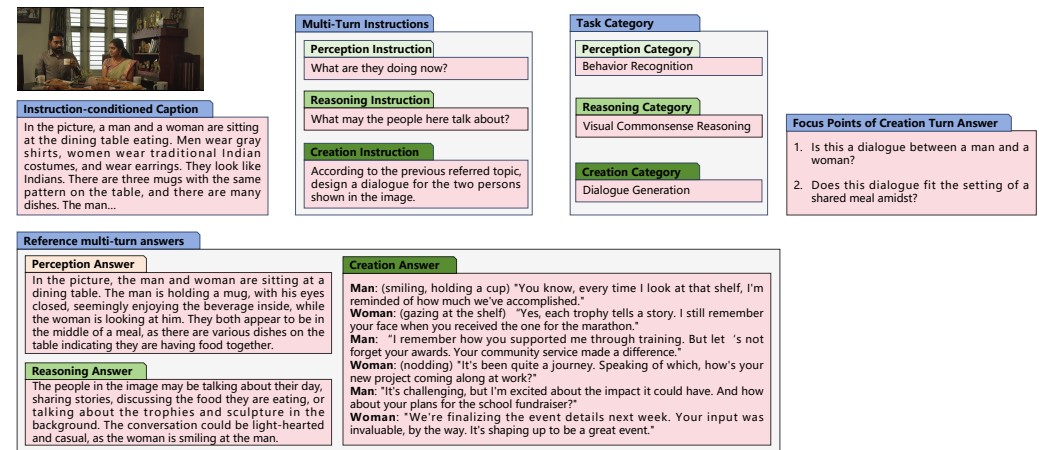

Figure 27: An example of ConvBench.

**How many instances are there in total (of each type, if appropriate)?**

There are 577 samples in total in ConvBench. Each sample has an image, three sequential instructions, three task categories, an instruction-conditioned caption, three-turn human-verified reference answers, and focus demands for the creation instruction. An example instance is shown in Figure 27.

**Does the dataset contain all possible instances or is it a sample (not necessarily random) of instances from a larger set?** If the dataset is a sample, then what is the larger set? Is the sample representative of the larger set (e.g., geographic coverage)? If so, please describe how this representativeness was validated/verified. If it is not representative of the larger set, please describe why not (e.g., to cover a more diverse range of instances, because instances were withheld or unavailable).

VisIT-Bench [7] is a single-turn visual question answering (VQA) dataset collected from users' "wish list". ConvBench uses its images and extends its questions in our multi-turn hierarchical structure. Therefore, these questions are representative of real-world use. No tests were run to determine representativeness.

**What data does each instance consist of? "Raw" data (e.g., unprocessed text or images) or features?** In either case, please provide a description.

Each instance consists of an image, three sequential instructions, three task categories, an instruction-conditioned caption, three-turn human-verified reference answers, and focus demands for the creation instruction. Each caption and each answer is verified by human annotators to abolish bias and errors.

**Is there a label or target associated with each instance?** If so, please provide a description.

The labels are human-verified reference answers for instructions derived from human annotators, as described above.

**Is any information missing from individual instances?** If so, please provide a description, explaining why this information is missing (e.g., because it was unavailable). This does not include intentionally removed information, but might include, e.g., redacted text.

Everything is included. No data is missing.

**Are relationships between individual instances made explicit (e.g., users' movie ratings, social network links)?** If so, please describe how these relationships are made explicit.

There is no explicit relationship between individual instances.

**Are there recommended data splits (e.g., training, development/validation, testing)?** If so, please provide a description of these splits, explaining the rationale behind them.

The instances all belong to the testing split.

**Are there any errors, sources of noise, or redundancies in the dataset?** If so, please provide a description.

No, we have carefully checked and annotated to avoid any errors. However, due to the large volume of data, there may be a very small number of errors.

**Is the dataset self-contained, or does it link to or otherwise rely on external resources (e.g., websites, tweets, other datasets)?** If it links to or relies on external resources, a) are there guarantees that they will exist, and remain constant, over time; b) are there official archival versions of the complete dataset (i.e., including the external resources as they existed at the time the dataset was created); c) are there any restrictions (e.g., licenses, fees) associated with any of the external resources that might apply to a future user? Please provide descriptions of all external resources and any restrictions associated with them, as well as links or other access points, as appropriate.

The benchmark is built upon the VisIT-Bench [7] the seed sample. There are no additional external resources required. The dataset is entirely self-contained.

**Does the dataset contain data that might be considered confidential (e.g., data that is protected by legal privilege or by doctor-patient confidentiality, data that includes the content of individuals non-public communications)?** If so, please provide a description.

No, the dataset does not contain data that might be considered confidential.

**Does the dataset contain data that, if viewed directly, might be offensive, insulting, threatening, or might otherwise cause anxiety?** If so, please describe why.

No, the dataset does not contain data which might be offensive, insulting, threatening, or might otherwise cause anxiety.

**Does the dataset relate to people?** If not, you may skip the remaining questions in this section.

Yes, part of the image of the dataset relates to people, however, they are not the primary emphasis of the dataset.

**Does the dataset identify any subpopulations (e.g., by age, gender)?** If so, please describe how these subpopulations are identified and provide a description of their respective distributions within the dataset.

No, the dataset does not identify any subpopulations.

**Is it possible to identify individuals (i.e., one or more natural persons), either directly or indirectly (i.e., in combination with other data) from the dataset?** If so, please describe how.

No, it is not possible to identify individuals. There is no personal information included in the dataset.

**Does the dataset contain data that might be considered sensitive in any way (e.g., data that reveals racial or ethnic origins, sexual orientations, religious beliefs, political opinions or union memberships, or locations; financial or health data; biometric or genetic data; forms of government identification, such as social security numbers; criminal history)?** If so, please provide a description.

No, the dataset does not contain data that might be considered sensitive in any way.

**Any other comments?**

None.

---

| Collection Process |
| :---: |

**How was the data associated with each instance acquired?** Was the data directly observable (e.g., raw text, movie ratings), reported by subjects (e.g., survey responses), or indirectly inferred/derived from other data (e.g., part-of-speech tags, model-based guesses for age or language)? If data was reported by subjects or indirectly inferred/derived from other data, was the data validated/verified? If so, please describe how.

As shown in Figure 3, the image and seed instruction is from the VisIT-Bench [7]. Our benchmark is constructed based on the VisIT-Bench [7]. We extend the seed instruction to three-turn instructions.

The annotations by human are applied for acquiring the instruction-conditioned caption, human-verified reference answers, focus demands and so on.

**What mechanisms or procedures were used to collect the data (e.g., hardware apparatus or sensor, manual human curation, software program, software API)?** How were these mechanisms or procedures validated?

As shown in Figure 3, To construct the multi-turn conversation and establish the capability hierarchy, each instance in ConvBench is composed of an input image, three hierarchical instructions, three

human-verified references, and an instruction-conditioned caption verified by humans (see the Figure 3). Specifically, the annotators start by extending an instruction from the VisIT-Bench [7] into three hierarchical instructions in a multi-turn manner. The above annotation process is generalized that according to an instruction in VisIT-Bench [7], we curate the perception instruction for the first turn, followed by the reasoning and creativity instructions, which are generated in response to the instructions and answers at preceding turns. These instructions are annotated by humans reflecting the real-world needs of human beings. Similar to VisIT-Bench [7], we then annotate the instruction-conditioned captions and gather human-validated reference answers. In these two processes, GPT4V plays the role of an auxiliary tool. The annotators have eliminated errors and biases by providing human-preference responses for all the requests.

**If the dataset is a sample from a larger set, what was the sampling strategy (e.g., deterministic, probabilistic with specific sampling probabilities)?**

The dataset is not a sample from a larger set. We use all the provided seed examples from VisIT-Bench [7].

**Who was involved in the data collection process (e.g., students, crowdworkers, contractors) and how were they compensated (e.g., how much were crowdworkers paid)?**

The authors were involved in the data collection process for no payment. They are working in the Shanghai AI Lab.

**Over what timeframe was the data collected? Does this timeframe match the creation timeframe of the data associated with the instances (e.g., recent crawl of old news articles)?** If not, please describe the timeframe in which the data associated with the instances was created.

The data set is annotated from September 2023 to February 2024.

**Were any ethical review processes conducted (e.g., by an institutional review board)?** If so, please provide a description of these review processes, including the outcomes, as well as a link or other access point to any supporting documentation.

No, there were not any ethical review processes conducted. Our benchmark is meticulously crafted upon the foundation of VisIT-Bench [7], a dataset that has been subjected to rigorous ethical scrutiny and content filtering. This meticulous process ensures that our benchmark not only meets but exceeds the current ethical standards, reflecting our commitment to upholding the highest levels of integrity and responsibility in the development and application of our technology.

**Does the dataset relate to people?** If not, you may skip the remaining questions in this section.

Yes, part of the image of the dataset relates to people, however, they are not the primary emphasis of the dataset.

**Did you collect the data from the individuals in question directly, or obtain it via third parties or other sources (e.g., websites)?**

We construct the data based on the VisIT-Bench [7], which is obtained via https://github.com/mlfoundations/VisIT-Bench/.

**Were the individuals in question notified about the data collection?** If so, please describe (or show with screenshots or other information) how notice was provided, and provide a link or other access point to, or otherwise reproduce, the exact language of the notification itself.

No. The data was from the public dataset, and the authors presumably knew that their image would be public.

**Did the individuals in question consent to the collection and use of their data?** If so, please describe (or show with screenshots or other information) how consent was requested

and provided, and provide a link or other access point to, or otherwise reproduce, the exact language to which the individuals consented.

No. The data was from the public dataset, and the authors presumably knew that their image would be public.

**If consent was obtained, were the consenting individuals provided with a mechanism to revoke their consent in the future or for certain uses?** If so, please provide a description, as well as a link or other access point to the mechanism (if appropriate).

Users can contact us to remove any annotation in our proposed benchmark.

**Has an analysis of the potential impact of the dataset and its use on data subjects (e.g., a data protection impact analysis) been conducted?** If so, please provide a description of this analysis, including the outcomes, as well as a link or other access point to any supporting documentation.

No.

**Any other comments?**

None.

---

**Preprocessing/cleaning/labeling**

---

**Was any preprocessing/cleaning/labeling of the data done (e.g., discretization or bucketing, tokenization, part-of-speech tagging, SIFT feature extraction, removal of instances, processing of missing values)?** If so, please provide a description. If not, you may skip the remainder of the questions in this section.

As shown in Figure 3, the image and the instructions are provided for the annotators to generate a caption. We first prompt GPT4V with "Describe this image in detail." We then polish the responses by humans according to the instructions to obtain the final instruction-conditioned caption. For each sample, we feed GPT4V with the instruction-conditioned caption, the image, multi-turn instructions, and our well-designed prompt in a multi-turn conversation fashion to generate each instruction's response. We meticulously refine these responses as reference answers, removing their biases and enhancing their quality and relevance. The preprocessing and cleaning of annotations can maintain the quality of the proposed benchmark.

**Was the "raw" data saved in addition to the preprocessed/cleaned/labeled data (e.g., to support unanticipated future uses)?** If so, please provide a link or other access point to the "raw" data.

The "raw" data is the VisIT-Bench, which can be downloaded via https://github.com/mlfoundations/VisIT-Bench/.

**Is the software used to preprocess/clean/label the instances available?** If so, please provide a link or other access point.

Yes, we use GPT4V to help with annotation (https://chat.openai.com/?model=gpt-4).

**Any other comments?**

None.

---

**Uses**

---

**Has the dataset been used for any tasks already?** If so, please provide a description.

The dataset has been only used for multi-turn visual conversation evaluation task and hierarchical ablation evaluation task in this paper.

**Is there a repository that links to any or all papers or systems that use the dataset?** If so, please provide a link or other access point.

No, there is only the original paper using this dataset now.

**What (other) tasks could the dataset be used for?**

The dataset could be used for anything related to researching multi-turn visual conversation. But, our dataset should only be used for non-commercial academic researc

**Is there anything about the composition of the dataset or the way it was collected and preprocessed/cleaned/labeled that might impact future uses?** For example, is there anything that a future user might need to know to avoid uses that could result in unfair treatment of individuals or groups (e.g., stereotyping, quality of service issues) or other undesirable harms (e.g., financial harms, legal risks) If so, please provide a description. Is there anything a future user could do to mitigate these undesirable harms?

There is minimal risk for harm.

**Are there tasks for which the dataset should not be used?** If so, please provide a description.

This data is collected for LVLMs to evaluate the ability of multi-turn visual conversation and help analyze the error attribution. Our dataset should only be used for non-commercial academic researc

**Any other comments?**

None.

| Distribution |
|:---:|

**Will the dataset be distributed to third parties outside of the entity (e.g., company, institution, organization) on behalf of which the dataset was created?** If so, please provide a description.

Yes, the benchmark will be open-sourced.

**How will the dataset will be distributed (e.g., tarball on website, API, GitHub)** Does the dataset have a digital object identifier (DOI)?

The data will be available through GitHub.

**When will the dataset be distributed?**

Code and benchmark are released at https://github.com/shirlyliu64/ConvBench

**Will the dataset be distributed under a copyright or other intellectual property (IP) license, and/or under applicable terms of use (ToU)?** If so, please describe this license and/or ToU, and provide a link or other access point to, or otherwise reproduce, any relevant licensing terms or ToU, as well as any fees associated with these restrictions.

The ConvBench dataset is licensed under the Creative Commons Attribution 4.0 International License (CC BY 4.0). For images, the original licensing terms are respected and remain applicable (VisIT-Bench [7]).

**Have any third parties imposed IP-based or other restrictions on the data associated with the instances?** If so, please describe these restrictions, and provide a link or other access point to, or otherwise reproduce, any relevant licensing terms, as well as any fees associated with these restrictions.

No, the benchmark owns the metadata and release as CC-BY-4.0 and we do not own the copyright of the images.

**Do any export controls or other regulatory restrictions apply to the dataset or to individual instances?** If so, please describe these restrictions, and provide a link or other access point to, or otherwise reproduce, any supporting documentation.

No.

**Any other comments?**

None.

| Maintenance |
| :---: |

**Who will be supporting/hosting/maintaining the dataset?**

OpenGVLab of Shanghai AI Laboratory will maintain the samples distributed. Huggingface will support hosting of the metadata.

**How can the owner/curator/manager of the dataset be contacted (e.g., email address)?**

The email addresses can be contacted are liushuo@pjlab.org.cn, zhangkaipeng@pjlab.org.cn and shaowenqi@pjlab.org.cn.

**Is there an erratum?** If so, please provide a link or other access point.

There is not an explicit erratum. We plan to maintain it through GitHub issues.

**Will the dataset be updated (e.g., to correct labeling errors, add new instances, delete instances)?** If so, please describe how often, by whom, and how updates will be communicated to users (e.g., mailing list, GitHub)?

No. However, specific samples can be removed on request.

**If the dataset relates to people, are there applicable limits on the retention of the data associated with the instances (e.g., were individuals in question told that their data would be retained for a fixed period of time and then deleted)?** If so, please describe these limits and explain how they will be enforced.

People may contact us to add specific samples to a blacklist.

**Will older versions of the dataset continue to be supported/hosted/maintained?** If so, please describe how. If not, please describe how its obsolescence will be communicated to users.

We will only support and maintain the latest version at all times and a new version release will automatically deprecate its previous version.

**If others want to extend/augment/build on/contribute to the dataset, is there a mechanism for them to do so?** If so, please provide a description. Will these contributions be validated/verified? If so, please describe how. If not, why not? Is there a process for communicating/distributing these contributions to other users? If so, please provide a description.

We warmly embrace contributions to ConvBench and will keep the community informed about any dataset expansions through our GitHub repository. However, it is imperative that contributors provide evidence of the high quality and non-harmful nature of the proposed data annotations. Submissions that do not meet these stringent criteria will not be integrated into our benchmark. Our commitment to maintaining a standard of excellence and safety in our dataset is unwavering, ensuring that ConvBench remains a reliable and ethical resource for the research community.

**Any other comments?**

None.

# R License and Intended Use

The ConvBench dataset is licensed under the Creative Commons Attribution 4.0 International License (CC BY 4.0). For images, the original licensing terms are respected and remain applicable (VisIT-Bench [7]). The images and associated annotations remain readily available for direct download (https://github.com/shirlyliu64/ConvBench). The metadata is also provided at https://github.com/shirlyliu64/ConvBench/metadata.json.

We release the benchmark under the CC-BY license and Terms of Use, requiring disclosure when used for model evaluation. This license supplements, but does not replace, the original licenses of source materials; compliance with these and any applicable rights of data subjects is necessary. This statement clarifies the responsibilities and liabilities associated with using this benchmark. While we've made every effort to ensure the samples' accuracy and legality, we cannot guarantee their absolute completeness or correctness. We assume no liability for any rights violations, whether legal or otherwise, that may occur through the use of this benchmark, including but not limited to copyright infringement, privacy violations, or misuse of sensitive information. By accessing, downloading, or using this benchmark, you implicitly accept this statement and agree to adhere to the terms and conditions of the CC-BY license. If you do not agree with these terms or the CC-BY license, you are not authorized to use this benchmark. The benchmark will be hosted and maintained on GitHub and the Hugging Face Hub platform.

