are three rows of kettlebells, with six kettlebells in each row. From up to down, the colors of the kettlebells from left to right on the first row is pale pink...

<|The Start of Assistant A's Conversation with User|>
### The first turn question from user:
How much do each color of kettlebells weigh?

### The first turn response from Assistant A:
Based on the given image and caption: A pale pink kettlebell weighs 8 kg. A light blue kettlebell weighs 10 kg. An orange kettlebell weighs 16 kg...

### The second turn question from user:
What are the colors on the second row but backwards?

### The second turn response from Assistant A:
The colors on the second row of kettlebells from right to left (backwards from the standard left-to-right reading order) are dark blue, dark blue, orange, orange, light blue, and dark brown.

### The third question from user:
What is the total weight of the kettlebells in the second row  but backwards in the even-numbered positions?

### The third turn response from Assistant A:
The 2nd kettlebell from the right in the second row is dark blue and weighs 20 kg. The 4th kettlebell from the right in the second row is orange and weighs 16 kg. The 6th kettlebell from the right in the second row is dark brown and weighs 14 kg. The total weight of the kettlebells in the second row, but backwards in the even-numbered positions, is 50 kg.

<|The End of Assistant A's Conversation with User|>

<|The Start of Assistant B's Conversation with User|>
### The first turn question from user:
How much do each color of kettlebells weigh?

### The first turn response from Assistant B:
The kettlebells in this image have their weights marked in kilograms. Starting from the top shelf and moving left to right, they are: Pink Kettlebell: 8 kg, Light Blue Kettlebell: 10 kg...

### The second turn question from user:
What are the colors on the second row but backwards?

### The second turn response from Assistant B:
The colors on the second row (middle shelf) from right to left (backwards from the original left to right order) are: Purple, Orange, Blue, Dark Brown.

###The third turn question from user:
What is the total weight of the kettlebells in the second row  but backwards in the even-numbered positions?

### The third turn response from Assistant B:
2nd position (from the right): Orange kettlebell - 32 kg 4th position: Dark Brown kettlebell - 24 kg. Adding these weights together: 32 kg (Orange) + 24 kg (Dark Brown) = 56 kg. So the total weight of the kettlebells in the even-numbered positions on the second row, when considered backwards from the usual left-to-right order, is 56 kilograms.
<|The End of Assistant B's Conversation with User|>

The first turn evaluation: Assistant A accurately provides the weights of each color of kettlebells as requested by  the user,...based on the accuracy and relevance to the user's question in the first turn: Overall, Response A is better.1. Does the poem explain that bright spots are visible due to the light they emit or reflect from themselves?

Let's analyze the second turn responses of both Assistant A and Assistant B: Assistant A accurately identifies the colors on the second row and provides the answer in reverse order as requested.  Assistant B provides the colors on the second row in reverse order as asked, but it includes a non-existent purple kettlebell, which does not align with the image information. Overall, Response A is better.

The third turn evaluation: In evaluating the third turn responses provided by Assistant A and Assistant B: Assistant A correctly identifies the kettlebells in the second row .... Assistant B also correctly identifies the kettlebells in the second row... Therefore, in this case, Assistant A provides the more accurate calculation .. Overall, Response A is better.
Think step-by-step, compare the overall conversation responses from the two assistants, and finish your response with "Overall, Response X is better." where X is either A or B.

assistant (machine generated; used to derive the prediction)

Let''s analyze the conversations:
### Assistant A:
In the first turn, Assistant A correctly provides the weights of each color of kettlebell as asked by the user. In the second turn, Assistant A accurately identifies the colors on the second row but provides the answer in reverse order as requested. In the third turn, Assistant A calculates the total weight of the kettlebells in the second row in the even-numbered positions, doing so accurately by summing the weights as instructed.
### Assistant B:
In the first turn, Assistant B confuses the colors and weights of the kettlebells, mentioning a purple kettlebell that does not exist in the image and misidentifying the weights of the blue kettlebells. In the second turn, Assistant B provides the colors on the second row in reverse order as asked, but it includes a non-existent purple kettlebell, which does not align with the image information. In the third turn, Assistant B correctly identifies the total weight of the kettlebells in the even-numbered positions on the second row, but only after making a mistake in the first turn, which could confuse the user.
### Comparison and Conclusion:
Assistant A follows the instructions accurately, provides correct information, and maintains consistency throughout the conversation, while Assistant B makes mistakes in the first turn that could lead to confusion and inconsistency. Assistant A''s responses are more precise, clear, and follow the user''s instructions without errors.
Overall, Response A is better.'

Figure 16: The prompt used for evaluating overall conversation in a pairwise grading method, accompanied by a sample completion from ChatGPT, is provided. The response conversations are anonymously presented to ChatGPT, where Assistant A is identified as a human, and Assistant B is recognized as GPT4V.

You are ImageTaskEvaluationGPT, an expert language model at judging whether or not a response adequately addresses an instruction in the context of an image. More specifically, you will be given the following:

1. An image context: This will describe the contents of an image with sufficient detail to address the instructions.
2. Three progressive turn instructions: These are three turn questions, the three questions are progressive.
3. Three turn reference outputs: These are high-quality example outputs that humans have judged to be accurate responses for the three input progressive instructions.
4. Three turn responses: The responses are from an AI assistant attempting to address the three progressive instructions in the context of the image.

Your job is to rate the first turn response from the AI assistant for the visual perception performance on a scale of 1 to 10. Rate the first turn response from the AI assistant by regarding the rating of the first turn reference output as 10.

Some things to remember:

- Even though you are just a language model, the image description will be sufficiently detailed so that your judgement can be accurate.
- Regard the ratings of the high-quality references as 10. Make your rating judgement for the responses from the AI assistant compared with the high-quality references.
- You are capable of judging responses quality. The first turn instruction is visual perception perspective. Correctness, relevance, fluency and the level of detail of responses are the most important factors which should be accounted for the first turn response.
- You think step-by-step and be as objective as possible, after providing your explanation, you must rate the first turn response for the visual perception performance on a scale of 1 to 10 by strictly following this format:"Rating:{rating}", for example: "Rating:{5}".

user (human authored)

I will describe the image to you, and provide three turn progressive instructions. Then, I will provide three corresponding reference outputs which are examples of high quality outputs for those three turn progressive instructions in the context of the image.
Then, I will give you three candidate responses that address the three progressive instructions in the context of the image: these will be labelled "The first turn response, The second turn response, The third turn response". Your job is to first reason step-by-step about the procs/cons of the first turn candidate response in terms of accuracy, relevance, fluency, the level of detail of responses etc. After reasoning step-by-step, comparing between the first turn candidate response and the first turn reference output and making the judgement by regarding the rating of the first turn reference output as 10, in the end, respond with "Rating:X." where X is a scale of 1 to 10.

assistant (human authored)

Sure, please provide the image context, the three instructions, the three reference outputs, and the three candidate responses. Then, I will think step-by-step and provide my ultimate rating judgement for the first turn candidate response.

user (from human authored template)

OK. Here are the image, the image description, the instructions, the high-quality references, and the responses.
Image context: A woman is playing a game of miniature golf. She is holding a mini golf putt. The image portrays an outdoor setting, most likely a mini-golf course or putting green, given the context...

<The Start of Reference Answer|>
### The first turn question from user:
What game is the person playing?

### The first turn high quality reference:
A woman is playing a game of miniature golf.

### The second turn question from user:
Is this person playing at a professional level? Give me your reasoning for your answer.

### The second turn high quality reference:
It does not appear that the person is playing mini-golf at a professional level. Here are some reasons for this determination:1. Attire: The individual is dressed in casual clothing\u2014a blue T-shirt...

### The third question from user:
Can you develop a weekly exercise plan to improve the sport level of playing the game for this person?

### The third turn high quality reference:
Improving at miniature golf largely involves enhancing hand-eye coordination, precision, and understanding the physics of how the ball moves on different surfaces. Here's a weekly exercise plan that focuses on these skills, which are crucial for improving performance in miniature golf:
Week 1: Fundamentals and Putting Technique
Monday - Understanding Basics: 30 minutes of putting on a flat surface. Focus on stance, grip, and swing mechanics.
Tuesday - Control Practice:Practice putting different distances (3 ft, 6 ft, 9 ft). 20 putts from each distance, noting consistency and accuracy. Regular practice on different types of putting surfaces and inclines to develop a consistent putting stroke and better judgment of distances and slopes.
Wednesday - Rest Day.
Thursday - Directional Control: Place obstacles on the putting surface and practice putting around them. 30 minutes of practice aiming for precision rather than power.
<|The End of Reference Answer|>

<|The Start of Assistant A' s Conversation with User|>
### The first turn question from user:
What game is the person playing?

### The first turn response from Assistant A:
The person in the image is playing miniature golf, also known as mini-golf or putt-putt golf.

### The second turn question from user:
Is this person playing at a professional level? Give me your reasoning for your answer.

### The second turn response from Assistant A:
1. The setting: The course looks like a typical recreational mini-golf course rather than a professional putting green used in tournaments. 2. Attire: The person is dressed in casual clothing, including sandals and a straw hat, which is more indicative of a casual player rather than professional golf attire...

### The third question from user:
Can you develop a weekly exercise plan to improve the sport level of playing the game for this person?

### The third turn response from Assistant A:
Creating a weekly exercise plan to improve performance in mini-golf should focus on enhancing coordination, balance, and concentration, as well as practising putting skills. However, it''s important to note that any new exercise plan should be started after consulting with a health professional, especially if the individual has any pre-existing health conditions or concerns.
Here''s a sample plan tailored for someone playing mini-golf:
 **Monday:**
 - Warm-up: 10 minutes of brisk walking.
 - Core strengthening exercises: 3 sets of plank holds for 30 seconds each.
 - Balance exercises: Standing on one foot for 30 seconds, then switch feet (3 sets each side).
 - Putting practice: 30 minutes focusing on stance and stroke consistency.
 **Tuesday:**
 - Rest day or light activity such as walking to allow the body to recover...
<|The End of Assistant A' s Conversation with User|>

Think step-by-step, rate the first turn response from Assistant A, and finish your response with "Rating:X." where X is all a scale of 1 to 10.

assistant (machine generated; used to derive the prediction)

The first turn response from Assistant A accurately identifies the game being played in the image as miniature golf, which aligns perfectly with the image context and the first turn reference output. The response is relevant, clear, and fluent in its delivery. It explicitly states the specific game being played and uses alternative terms (mini-golf, putt-putt golf) that demonstrate a comprehensive understanding of the activity depicted in the image. Furthermore, the response is adequately detailed by providing specific terminology related to the game, showing attention to contextual accuracy.
Compared to the high-quality reference output for the first turn instruction, which is succinct and directly answers the question, Assistant A''s response mirrors the precision and correctness expected, hence deserving a high rating. Rating: 9

Figure 17: The prompt used for evaluating perception in a direct grading method, accompanied by a sample completion from ChatGPT, is provided.

**system prompt (human authored)**

You are ImageTaskEvaluationGPT, an expert language model at judging whether or not a response adequately addresses an instruction in the context of an image. More specifically, you will be given the following:

1. An image context: This will describe the contents of an image with sufficient detail to address the instructions.
2. Three progressive turn instructions: These are three turn questions, the three questions are progressive.
3. Three turn reference outputs: These are high-quality example outputs that humans have judged to be accurate responses for the three input progressive instructions.
4. Three turn responses: The responses are from an AI assistant attempting to address the three progressive instructions in the context of the image.

Your job is to rate the second turn response from the AI assistant for the visual reasoning performance on a scale of 1 to 10. Rate the second turn response from the AI assistant by regarding the rating of the second turn reference output as 10.

Some things to remember:

- Even though you are just a language model, the image description will be sufficiently detailed so that your judgement can be accurate.
- Regard the ratings of the high-quality references as 10. Make your rating judgement for the responses from the AI assistant compared with the high-quality references.
- You are capable of judging responses quality. The second turn instruction is visual reasoning perspective. Correctness, relevance, fluency and the level of detail of responses are the most factors which should be accounted for the second instruction.
- You think step-by-step and be as objective as possible, after providing your explanation, you must rate the second turn response for the visual reasoning performance on a scale of 1 to 10 by strictly following this format:"Rating:{rating}", for example: "Rating:{5}".

**user (human authored)**

I will describe the image to you, and provide three turn progressive instructions. Then, I will provide three corresponding reference outputs which are examples of high quality outputs for those three turn progressive instructions in the context of the image.
Then, I will give you three candidate responses that address the three progressive instructions in the context of the image: these will be labelled "The first turn response, The second turn response, The third turn response". Your job is to first reason step-by-step about the procs/cons of the second turn candidate response in terms of accuracy, relevance, fluency, the level of detail of responses etc. After reasoning step-by-step, comparing between the second turn candidate response and the second turn reference output and making the judgement by regarding the rating of the second turn reference output as 10, in the end, respond with "Rating:X." where X is a scale of 1 to 10.

**assistant (human authored)**

Sure, please provide the image context, the three instructions, the three reference outputs, and the three candidate responses. Then, I will think step-by-step and provide my ultimate rating judgement for the second turn candidate response.

**user (from human authored template)**

OK. Here are the image, the image description, the instructions, the high-quality references, and the responses.
Image context: The image portrays two individuals, both dressed in what appears to be period attire. Here's a detailed description: Setting: The background is relatively dark, perhaps indicating an indoor or theatrical setting. The floor appears to be wooden...

<|The Start of Reference Answer|>
### The first turn question from user:
Can you describe the people's attire shown in the image?

### The first turn high quality reference:
A woman is playing a game of miniature golf.

### The second turn question from user:
What religion do these people belong to? What time period and location are they from?

### The second turn high quality reference:
Based on their attire, which includes modest and functional clothing with white bonnets, large white collars, brown skirts, white aprons, and black shoes, the individuals in the image seem to be representing Puritans...

### The third question from user:
Outline a powerpoint what the slides could be like for a history teacher to introduce the relevant history based on the previous responses and image.

### The third turn high quality reference:
I can outline what the slides could look like for a histor teacher''s presentation about Puritans in 17th-century Massachusetts based on the attire and context provided:
  **Slide 1: The Puritans of 17th-Century Massachusetts**
  - Image: (Insert image from above)
  - Subtitle: "An Exploration of Colonial Life and Beliefs"
<|The End of Reference Answer|>

<|The Start of Assistant A's Conversation with User|>
### The first turn question from user:
Can you describe the people's attire shown in the image?

### The first turn response from Assistant A:
The individuals in the image are wearing attire that resembles clothing from a historical period, perhaps suggestive of peasant or working-class dress from several centuries ago. Their costumes include the following elements:...

### The second turn question from user:
What religion do these people belong to? What time period and location are they from?

### The second turn response from Assistant A:
Based on the image alone, it's not possible to accurately determine the religion, exact time period, or the location of the people depicted...

### The third question from user:
Outline a powerpoint what the slides could be like for a history teacher to introduce the relevant history based on the previous responses and image.

### The third turn response from Assistant A:
Title: Life in the Past: Exploring Historical Clothing and Society
  Slide 1: Introduction
  - Title: "Understanding the Past through Fashion"
  - Image of the people with historical attire
  - Brief introductory text: "Discover how clothing can reveal information about historical
  periods, social status, and daily life."
<|The End of Assistant A's Conversation with User|>

Think step-by-step, rate the second turn response from Assistant A, and finish your response with "Rating:X." where X is all a scale of 1 to 10.

**assistant (machine generated; used to derive the prediction)**

The second turn candidate response from Assistant A provides a detailed analysis of the attire shown in the image but falls short in accurately determining the religion, time period, and location of the individuals depicted. The response correctly describes the clothing as resembling attire from past centuries, such as the Medieval period to the 17th or 18th century, and identifies the lack of clear religious symbols. However, it fails to definitively establish the individuals" affiliation with Puritanism, their specific location (Massachusetts), or the time period (around 1620). The reference output, in contrast, accurately ties the attire to Puritans in the early 17th century in Massachusetts, emphasizing the religious and historical context with more precision.

In terms of relevance, the response does make valid points about the limitations of determining religious affiliation solely based on clothing and acknowledges the need for additional context. The fluency of the response is relatively smooth, maintaining a logical flow in discussing the historical significance of the attire. However, the level of detail could be improved by incorporating more historical context and specifics regarding Puritan clothing styles to align better with the reference.

Overall, the second turn response from Assistant A demonstrates a moderate understanding of the image but lacks the precision and depth found in the high-quality reference output. Therefore, I would rate the second turn response