# OpenReview forum: "ConvBench: A Multi-Turn Conversation Evaluation Benchmark with Hierarchical Ablation Capability for Large Vision-Language Models"
_NeurIPS.cc/2024/Datasets_and_Benchmarks_Track — NeurIPS 2024 Track Datasets and Benchmarks Spotlight_

### Official Review · Reviewer_q8kb · 2024-07-27
**Review for #1478**

**Rating:** 8
**Confidence:** 5
**Correctness:** Yes, the dataset and the experimental…
**Clarity:** This paper is well written.

**Review:**

Strengths:

1. The proposed benchmark has been meticulously constructed, featuring three levels of abilities: perception, reasoning, and creation. It includes 577 multi-turn conversations spread across 215 tasks, offering a fine-grained and comprehensive evaluation framework.

2. This paper is well-written and easy to follow.

3. The experimental findings underscore the benchmark's challenges for current MLLMs and yield valuable insights for future research. Notably, the critical role of perception in facilitating reasoning and creation is highlighted, along with the inherent challenges posed by multi-turn conversations.


Weaknesses:

1. The benchmark relies on ChatGPT-3.5 to serve as judges. Why not employ more advanced models, such as GPT-4V? Additionally, how much are the associated costs of conducting a comprehensive evaluation?

2. There are some typos, for example, the authors use different format versions for GPT4v, GPT-4V, and GPT4-V.

3. The authors should discuss more related works on multi-turn dialogue evaluation, I've found some very relevant works such as [1] introducing MT-Bench-101, a benchmark that proposes a hierarchical, fine-grained approach to multi-turn dialogue assessment.

[1] Ge Bai, Jie Liu, Xingyuan Bu, Yancheng He, Jiaheng Liu, Zhanhui Zhou, Zhuoran Lin, Wenbo Su, Tiezheng Ge, Bo Zheng, Wanli Ouyang. “MT-Bench-101: A Fine-Grained Benchmark for Evaluating Large Language Models in Multi-Turn Dialogues.”. ACL 2024.

**Strengths:**

1. The proposed benchmark has been meticulously constructed, featuring three levels of abilities: perception, reasoning, and creation. It includes 577 multi-turn conversations spread across 215 tasks, offering a fine-grained and comprehensive evaluation framework.

2. This paper is well-written and easy to follow.

3. The experimental findings underscore the benchmark's challenges for current MLLMs and yield valuable insights for future research. Notably, the critical role of perception in facilitating reasoning and creation is highlighted, along with the inherent challenges posed by multi-turn conversations.

**Additional Feedback:**

N/A

**Documentation:**

Yes

**Ethics:**

No ethical concerns.

**Opportunities For Improvement:**

1. The benchmark relies on ChatGPT-3.5 to serve as judges. Why not employ more advanced models, such as GPT-4V? Additionally, how much are the associated costs of conducting a comprehensive evaluation?

2. There are some typos, for example, the authors use different format versions for GPT4v, GPT-4V, and GPT4-V.

3. The authors should discuss more related works on multi-turn dialogue evaluation, I've found some very relevant works such as [1] introducing MT-Bench-101, a benchmark that proposes a hierarchical, fine-grained approach to multi-turn dialogue assessment.

[1] Ge Bai, Jie Liu, Xingyuan Bu, Yancheng He, Jiaheng Liu, Zhanhui Zhou, Zhuoran Lin, Wenbo Su, Tiezheng Ge, Bo Zheng, Wanli Ouyang. “MT-Bench-101: A Fine-Grained Benchmark for Evaluating Large Language Models in Multi-Turn Dialogues.”. ACL 2024.

**Relation To Prior Work:**

The authors should discuss more related works on multi-turn dialogue evaluation, I've found some very relevant works such as [1] introducing MT-Bench-101, a benchmark that proposes a hierarchical, fine-grained approach to multi-turn dialogue assessment.

[1] Ge Bai, Jie Liu, Xingyuan Bu, Yancheng He, Jiaheng Liu, Zhanhui Zhou, Zhuoran Lin, Wenbo Su, Tiezheng Ge, Bo Zheng, Wanli Ouyang. “MT-Bench-101: A Fine-Grained Benchmark for Evaluating Large Language Models in Multi-Turn Dialogues.”. ACL 2024.

**Summary And Contributions:**

This paper introduces a new benchmark named LVLMs, designed to evaluate the multi-turn conversation capabilities of MLLMs. The benchmark encompasses 577 multi-turn conversations across 215 tasks, organized into three levels of ability: perception, reasoning, and creation. The experimental results indicate that even the most advanced models, such as GPT-4V, face significant challenges with this benchmark.

---

> ### Author Rebuttal · Authors · 2024-08-17
>
> Thank you for your thoughtful and comprehensive review of our paper. We are excited to see your positive comments and acknowledgment of our writing, our meticulously constructed multi-turn multi-modal conversation benchmark for fine-grained evaluation of hierarchical ablation capability, and our insightful experimental findings yield valuable insights for future research of MLLMs. We will address your minor concerns as follows. Due to the limited space, we place the second half of the response in the official comment.
>
> > **Cost Estimate for Evaluation. (Weakness #1)**
>
> According to the the process of ConvBench. As shown in Table 1, a comprehensive evaluation for a model need 9 scores including $S_{1}$, $S_{2}$, $S_{3}$, $S_{O}$, $\hat{S_{2}}$, $\hat{S_{3}}$, $\hat{S_{O}}$, $\tilde{S_{3}}$, $\tilde{S_{O}}$. The instruction-conditioned-caption is dense and the reference answers are annotated in detail. A comprehensive experiment need to consume about 42-52 billion tokens. In total, at current ChatGPT-3.5 prices (0.5 dollars per 1M tokens), the multi-turn comparison evaluations required to assess a new model costs about 21-26 dollars. When we conduct our experiments, ChatGPT-3.5 is the most cost-effective. However, after the price reduction, GPT-4o mini is cheaper and more accurate than ChatGPT 3.5 (6.3-7.8 dollars with regular API, 1.6-3.9 dollars with Batch API). In summary, the cost is satisfied, and thus, our evaluation is easily accessible to the community.
>
> > **Concerns on ChatGPT-3.5 Evaluation. (Weakness #1)**
>
> To save on costs and make our evaluation framework more accessible for the community, we selected ChatGPT-3.5 for evaluation in the early stages of our research. We also found that the limitation of ChatGPT-3.5. In order to strike a balance between evaluation effectiveness and evaluation cost, we write high-quality instruciton-conditioned caption includes fine-grained descriptions of the image and correct answers, which largely improves the performance of ChatGPT-3.5 evaluation. In section C Human Evaluation in Supplementary File, we have assessed our automatic evaluation's agreement with human evaluation to have reached 81.83%, with the results being essentially indistinguishable from those of GPT-4V (85.56%).
>
> > **Typographical Errors. (Weakness #2)**
>
> We apologize for these typos about different format versions for GPT4v, GPT-4V, and GPT4-V. We will represent it in the same format in the camera-ready version.
>
> > **A Discussion on More Related Works on Multi-Turn Dialogue Evaluation. (Weakness #3)**
>
> Thank you for helping us supplement related works on multi-turn dialogue evaluation. We will add more related works about this and related citations in the final camera-ready version.
>
> - **Benchmarks for Multi-turn LLMs:** For Large Language Models (LLMs), there have been some classic work. MT-Bench [1] is the first multi-turn conversation benchmark, which focuses on two-turn follow-up dialogues across eight topics ("Writing", "Knowledge", "Math" and so on). It only focuses on the most basic and important multi-turn abilities of Context Memory and Anaphora Resolution. MT-Bench-101 [2] is the first dataset to specifically focus on 13 fine-grained multi-turn dialogue abilities, such as, "Separate Input", "Topic Shift", "Content Confusion", "Content Rephrasing" and so on. Otherwise, there are other benchmarks: MT Bench++ [3] is an eight-turn multi-turn conversation benchmark to qualitatively evaluate multi-turn instruction following ability. It is built on expanding MT-Bench by manually annotating six additional follow-up questions. MINT [4] is a  benchmark that evaluates LLMs’ ability to solve challenging tasks with multi-turn interactions by (1) using tools and (2) leveraging natural language feedback.
>
> - **Benchmarks for Multi-turn LVLMs:** For Large Vision-Language Models (LVLMs), to the best of our knowledge, ConvBench is the first multi-turn visual conversation benchmark, which can also be considered as a multi-modal version of MT-Bench, but with the innovative addition of hierarchical ablation testing. ConvBench depends on manual annotation for obtaining the three-turn interactive instructions to allow evaluating multi-turn instruction-following ability for visual dialogue. ConvBench also mostly focuses on the most basic and important multi-turn visual abilities of Context Memory and Anaphora Resolution. We will explore more fine-grained multi-turn visual dialogue abilities in ConvBench 2.0, which will be represented as a multi-modal version of MT-Bench-101 [2].
>
> The similarities and differences between ConvBench and MT-Bench-101 can summarized as follows:
>
> - **Design Purpose and Evaluation Emphasis:** ConvBench is the first multi-turn visual conversation benchmark for **LVLMs**, which mainly focuses on multi-turn visual abilities of Context Memory and Anaphora Resolution, but with the innovative addition of hierarchical ablation testing. MT-Bench-101 is the first comprehensive multi-turn dialogues with 13 distinct abilities for **LLMs**, such as "Topic Shift", "Content Confusion", "Content Rephrasing" and so on.
>
> - **Meaning of "Hierarchy" Varies:** "Hierarchy" in ConvBench is about decoupling the capability requirements of LVLMs onto instructions for three progressive turns, where the first turn is perception of instructions, the second turn is reasoning based on perception, and the third turn is creation based on both perception and reasoning. "Hierarchy" in MT-Bench-101 is represented as a three-tier hierarchical ability taxonomy, as shown in Figure 2 of MT-Bench-101. The first layer outlines three progressive overarching abilities: Perceptivity, Adaptability, and Interactivity. **However, it does not create relationships among these abilities like what ConvBench did.** The second tier specifies seven detailed abilities based on the first layer. The third tier further decomposes these abilities in the second tier into 13 distinct tasks.

---

> > ### Author Response · Authors · 2024-08-17
> > **Response (2/2)**
> >
> > - **Some Similar Conclusions Drawn:**  Because both of these works focus on the multi-turn dialogue capabilities of large models, they have reached some similar conclusions regarding multi-turn dialogue capabilities based on their respective experimental results: (a) A decline between the first turn and subsequent turns, which suggests that LVLMs tend to generate comprehension biases as the multi-turn dialogue progresses or forget the content of previous turns. (b) The high-quality dialogue history, which plays an important role in in-context learning examples, provides effective guidance for the LVLMs' responses. (c) We find that the trend of increasing language model size is related  to an improvement in model's performance in multi-turn conversation.
> >
> > [1] Lianmin Zheng, Wei-Lin Chiang, Ying Sheng, Siyuan Zhuang, Zhanghao Wu, Yonghao Zhuang,  Zi Lin, Zhuohan Li, Dacheng Li, Eric P. Xing, Haotong Zhang, Joseph Gonzalez, and Ion Stoica. Judging llm-as-a-judge with mt-bench and chatbot arena. ArXiv, abs/2306.05685, 2023.
> >
> > [2] Ge Bai, Jie Liu, Xingyuan Bu, Yancheng He, Jiaheng Liu, Zhanhui Zhou, Zhuoran Lin, Wenbo Su, Tiezheng Ge, Bo Zheng, Wanli Ouyang. “MT-Bench-101: A Fine-Grained Benchmark for Evaluating Large Language Models in Multi-Turn Dialogues.”. ACL 2024.
> >
> > [3] Wang, Xingyao et al. “MINT: Evaluating LLMs in Multi-turn Interaction with Tools and Language Feedback.” ArXiv abs/2309.10691 (2023): n. pag.
> >
> > [4] Xingyao Wang, Zihan Wang, Jiateng Liu, Yangyi  Chen, Lifan Yuan, Hao Peng, and Heng Ji. 2023b.  Mint: Evaluating llms in multi-turn interaction  with tools and language feedback. arXiv preprint  arXiv:2309.10691.

---

> > > ### Comment · Reviewer_q8kb · 2024-08-20
> > > **Response to Authors**
> > >
> > > I would like to thank the authors for the detailed response which solved my concerns. I will raise my scores from 7 to 8.

---

> > > > ### Author Response · Authors · 2024-08-20
> > > >
> > > > Dear Reviewer q8kb,
> > > >
> > > > Thank you once again for your comprehensive review and insightful feedbacks on our ConvBench paper. Your thoughtful review can substantially improve our paper.
> > > >
> > > > Best regards,
> > > >
> > > > The ConvBench Authors.

---

### Official Review · Reviewer_qGUt · 2024-07-27
**A well structured valuable contribution to understanding limitations in Large Vision-Language Models**

**Rating:** 8
**Confidence:** 3
**Correctness:** As far as I can tell, the claims made…
**Clarity:** The paper is well written.

**Review:**

The paper explains the aim and process of providing this benchmark data set well. It has sufficient clarity and originality to significantly contribute to the track.

**Strengths:**

The idea and realization of the hierarchical division of tasks related to images allow an innovative and more rigorous evaluation of large vision-language models that were not possible before.

**Additional Feedback:**

It would be perfect if the data set were released soon. Furthermore, a more detailed overview of the data set's details would be helpful.

**Documentation:**

Unfortunately the code and data is not released yet. However, the procedure of data creation is described in detail.

**Ethics:**

The data set is built on top of another data set that was already released, so there are implications for the proposed data set. Ethical considerations are not detailed in the paper.

**Limitations:**

The authors addressed the issue of the data set being not too large. However, one could also address the relevance of the tasks, balance of categories, types of images, etc.

**Opportunities For Improvement:**

It seems that Figure 3 displays the wrong image (it does not fit the description).
In section 4 (i) and following, the formulas use the index i (outside and inside) in more than one way. Please use another index like j or k here.

**Relation To Prior Work:**

The discussions of related work show the limitations in possible evaluations that are now enhanced by the data set the paper proposes.

**Summary And Contributions:**

The paper introduces a benchmark data set to evaluate large vision-language models. In contrast to many benchmarks, it divides the task associated with an image into three steps: perception, reasoning, and (where applicable) creation. This fosters the understanding of specific abilities and challenges different models might have. The structure of the paper allows the reader to follow the ideas and results of the authors.

---

> ### Author Rebuttal · Authors · 2024-08-17
>
> We appreciate your careful evaluation and acknowledgment of the significance of our work, which made a well-structured valuable contribution to understanding limitations in Large Vision-Language Models. We are also excited that you find our work allows an innovative and more rigorous evaluation of large vision-language models that were not possible before and provides sufficient clarity and originality to contribute significantly to the track. We will address your minor concerns as follows.
>
> > **Image Errors and Ambiguous Expressions. (Opportunities For Improvement)**
>
> Thanks for your detailed reviews. We apologize for the wrong image in Figure 3, which will be corrected in the final version. Also, we apologize for the confusion of index $i$, which will be replaced with an index like j or k in the final version.
>
> > **Limitations of ConvBench. (Limitations)**
>
> Thank you very much for your thoughtful review. More discussion about the limitations of ConvBench will be added in the final version, including the relevance of tasks, balance of categories, and types of images.
>
> > **Ethics Considerations. (Ethics)**
>
> We apologize for not detailing the Ethics Considerations in the paper. A detailed discussion about this will be added in the final version. ConvBench is built on the VisIT-Bench [1], whose images  all carry a public license specification. Our contributions such as multi-turn instructions, reference outputs, instruction-conditioned captions, and focus points, is licensed under the Creative Commons Attribution 4.0 International License (CC BY 4.0). There are no or only very minor ethical concerns.
>
> [1] Yonatan Bitton, Hritik Bansal, Jack Hessel, Rulin Shao, Wanrong Zhu, Anas Awadalla, Josh Gardner, Rohan Taori, and Ludwig Schimdt. Visit-bench: A benchmark for vision-language instruction following inspired by real-world use. ArXiv, abs/2308.06595, 2023.
>
> > **Benchmark Release. (Documentation & Additional Feedback)**
>
> Thank you very much again for your affirmation. The dataset and code are partly distributed on the GitHub webpage at: https://github.com/shirlyliu64/ConvBench . We will add this url to the final version. The complete dataset will be released as soon as possible. We have already written a datasheet in section **N Datasheet** in the Supplementary File . A more detailed overview of the dataset's details would be added to the final version as follows:
>
> > **Dataset Name:**
> > ConvBench
> >
> > **Purpose:**
> > ConvBench is designed to evaluate the hierarchical capabilities of LVLMs in multi-turn visual conversations, including perception, reasoning, and creativity.
> >
> > **Composition:**
> > - **Size:** The dataset comprises 577 curated multi-turn conversations.
> > - **Tasks:** It encompasses 215 distinct tasks, which are broad and open-ended, resembling real-world user behaviors.
> > - **Types of Tasks:** The tasks are categorized into 71 perception tasks, 65 reasoning tasks, and 79 creation tasks.
> >
> > **Structure:**
> > - **Hierarchical Questions:** ConvBench structures questions in a hierarchical manner, progressively examining the model's capabilities in perception, reasoning, and creativity within each conversation.
> > - **Decoupling Capabilities:** The dataset allows for the decoupling of different capabilities in evaluations, enabling more accurate error attribution.
> >
> > **Source of Data:**
> > - **Images and Questions:** Collected from real-world user behavior, using images from datasets like VisIT-Bench and extending its questions into a multi-turn hierarchical structure.
> >
> > **Unique Features:**
> > - **Open-Ended Questions:** Unlike traditional benchmarks, ConvBench allows for open-ended responses, mimicking real-world user behavior.
> > - **Automatic Evaluation Framework:** An efficient and reliable automatic evaluation method, ConvBenchEval, is introduced to assess open-ended visual questions.
> >
> > **Evaluation Methodology:**
> > - **Reference Answers:** Each question is annotated with reference answers.
> > - **Image Captions:** Each image is provided with a question-aware caption.
> > - **Assessment Focus Points:** For creation questions, assessment focus points are annotated for more precise evaluation.
> > - **Evaluation Tool:** ChatGPT is utilized to judge if the LVLM's response is superior to the reference answer, achieving a high agreement rate with human evaluations.
> >
> > **Findings from Experiments:**
> > - **Performance of LVLMs:** Even advanced models like GPT4v score only 39.51%, indicating the challenge ConvBench presents.
> > - **Weaknesses Identified:** The dataset reveals weaknesses in "OCR", "Fine-grained", and "Spatial" perception, as well as reasoning capabilities requiring "Professional Knowledge", "Emotional Intelligence", "Imagination", and "Sense of Space".
> >
> > **Significance:**
> > - **Benchmark for AGI:** ConvBench is aligned with the features expected of artificial general intelligence (AGI), focusing on open-ended multi-turn visual conversations.
> > - **Insights into LVLM Development:** Performance across different tasks and models suggests synchronous development in the field, with improvements correlating to the size of the language model.
> >
> > **Contributions:**
> > - ConvBench is the first of its kind, offering a challenging, real-world benchmark for evaluating LVLMs in open-ended multi-turn visual conversations.
> > - It introduces a novel hierarchical ablation evaluation method, providing insights into the strengths and weaknesses of current LVLMs.
> > - The automatic evaluation method presented achieves high reliability, setting a new standard for evaluating such complex conversational AI capabilities.
> >
> > This dataset represents a significant advancement in the evaluation of AI models, pushing the boundaries of what is expected from LVLMs in terms of understanding and generating human-like responses in visual conversation contexts.

---

### Official Review · Reviewer_HBs1 · 2024-08-10
**ConvBench: A Multi-Turn Conversation Evaluation  Benchmark with Hierarchical Ablation Capability for  Large Vision-Language Models**

**Rating:** 7
**Confidence:** 4

**Review:**

The paper presents an innovative approach by transforming single-turn instructions into multi-turn sequences. This method allows for the identification of errors in both perception and reasoning phases, offering valuable insights for benchmarking the distinct capabilities of language models. The approach is well-supported by carefully chosen examples that effectively demonstrate its practical applicability. However, the evaluation using GPT-3.5 is limited. In my experience, GPT-3.5 often delivers inconsistent scores, even with references, especially when dealing with out-of-distribution data. The method may also struggle to detect subtle differences between generated samples without detailed prompts. Additionally, the heavy reliance on human annotations at every step makes this benchmark difficult to scale.

**Strengths:**

1. The innovative approach of converting single-turn instructions into multi-turn sequences is a significant contribution, as it allows for the identification of errors in both the perception and reasoning phases. This methodology offers valuable insights for benchmarking the distinct capabilities of language models.

2. The idea is effectively supported by well-chosen examples, which illustrate its potential and practical applicability.

**Additional Feedback:**

None.

**Clarity:**

The paper is well-organized and somehow clearly written, making it easy to follow the authors' arguments. However, the experiment part is hard to read.

**Correctness:**

The methods for data collection and labeling appear sound and are supported by comprehensive documentation.

**Documentation:**

The documentation is comprehensive, with detailed information on data collection processes, intended uses, and ethical considerations.

**Ethics:**

No significant ethical concerns were identified, although further clarification on data consent processes would be beneficial.

**Limitations:**

The author states the limitations of this paper and adds ethical considerations.

**Opportunities For Improvement:**

1. The evaluation using GPT-3.5 is limited. I have attempted a similar approach, using few-shot examples (references) to guide GPT-3.5 in scoring generated results. From my experience, GPT-3.5 often delivers inconsistent scores when dealing with out-of-distribution data. Did you create few-shot examples (references) for every category to ensure consistency? Additionally, GPT-3.5 struggles to detect subtle differences between generated samples. For instance, if one response incorrectly states that g5 can capture f7, while another correctly states that f3 can capture e5, GPT-3.5 might not differentiate these scores accurately without a detailed prompt listing all the correct answers.


2. The heavy reliance on human annotations in this benchmark makes it challenging to scale. The processes of Multi-Turn Instruction Formation and Task Category Induction are particularly dependent on manual effort. Many of these tasks could be automated, such as using a language model for Task Category Induction. There should be a discussion on the most effective ways to extend single-turn to multi-turn instructions (e.g., what types of multi-turn interactions best enhance a model’s perception and reasoning). Additionally, it would be valuable to explore how to automate these processes to facilitate scaling.

**Relation To Prior Work:**

The authors provide a thorough comparison with existing datasets and clearly articulate the unique contributions of their work.

**Summary And Contributions:**

This paper introduces ConvBench, a comprehensive benchmark for multi-turn conversation evaluation in large vision-language models. The key contributions of this work are:

1. A novel annotation method that transforms single-turn instructions into multi-turn sequences, enabling the identification of where errors occurring.

2. A high-quality, human-annotated dataset.

3. A hierarchical ablation evaluation method that assesses models across three dimensions: perception, reasoning, and creation, providing a nuanced understanding of model performance.

---

> ### Author Rebuttal · Authors · 2024-08-17
>
> We thank you for your careful evaluation and appreciate your acknowledgment of our contribution. We are encouraged to see you find our work (a) makes a significant contribution by introducing an innovative approach to convert single-turn instructions into multi-turn sequences, allowing for the identification of errors in both the perception and reasoning phases. (b) offers valuable insights for benchmarking the distinct capabilities of language models. (c) is well-supported by carefully chosen examples that effectively demonstrate its practical applicability. We will address your concerns as follows. Due to the limited space, we place some responses in the official comment.
>
> >  **Concerns on ChatGPT-3.5 Evaluation. (Opportunities For Improvement #1)**
>
> Thank you for highlighting this concern. We selected ChatGPT-3.5 for evaluation in the early stages of our research to save on costs and make our evaluation framework more accessible to the community. We also found that the limitation of ChatGPT-3.5. Therefore, we aim to ensure the effectiveness of the evaluation. We write high-quality instruciton-conditioned captions that include fine-grained descriptions of the image and information about the correct answers, which largely improves the performance of ChatGPT-3.5 evaluation. In section C Human Evaluation in Supplementary File, we have assessed our automatic evaluation's agreement with human evaluation to have reached 81.83%, with the results being essentially indistinguishable from those of GPT-4V (85.56%).
>
> Compared with your mention of creating few-shot examples (references) for every category to ensure consistency, ConvBenchEval is implemented with greater refinement by fine-grained caption generation.  Every sample's instruction-conditioned caption not only includes fine-grained descriptions of the image but also incorporates the information about the correct answers to the instructions into the caption to ensure ChatGPT-3.5 has enough information to make a convincing judgment. Although ChatGPT-3.5 may have difficulty detecting subtle differences between generated samples, **ChatGPT-3.5 combined with detailed captions** acts as a more impartial and effective judge for scoring.  We apologize for this important detail not being reflected in the current version, and it will be included in the final version.
>
> According to the example you referred to, with the help of the detailed caption, ChatGPT-3.5 can also detect subtle differences between generated samples. We use the prompt via ChatGPT-3.5 for evaluating reasoning turn in pairwise grading method using the prompt in Figure 14 in the Supplementary File.
>
> If we only use **the image caption which is not involved with the information of correct answers to instructions**, the judge's result will be inaccurate.
>
> > **Caption used in this setting:**  There is a chessboard in the image. The chessboard is arranged in a total of 64 squares. In order to facilitate describing the chess game, the chessboard is marked horizontally with the letters "abcdefgh" and vertically with the numbers "12345678". For example, the lower left corner is "a1". The white King is on "e1", the white Queen is on "d1", the white Rooks are on "a1" and "h1", the white Bishops are on "c1" and "a4", the white Knights are on "b1" and "f3", the white Pawns are on "a2", "b2", "c2","e4","f2","g2", and "h2". The black King is on "e8", the black Queen is on "d8", the black Rooks are "a8" and "h8", the black Bishops are "c8" and "f8", the black Knights are "c6" and "g8", the black Pawns are "a6", "b7", "c7", "d7", "e5", "f7", "g7" and "h7".
>
> The judge will wrongly think Assistant A and Assistant B are both incorrect. The detailed judge results are as follows, where Assistant A is a LVLM, and Assistant B is human.
>
> > Let's break down the second-turn responses from Assistant A and Assistant B to determine which one is better.
> >
> > ### User's Second Turn Question:
> > "I am the player with the white chess pieces. Is there a move I can take on this turn to capture one of my opponent’s pieces?"
> >
> > ### Response from Assistant A:
> > "Yes, as the player with the white pieces, you have a capturing move available. Your pawn on e5 can capture the black pawn on d6. This is written in algebraic notation as e5xd6."
> >
> > ### Response from Assistant B:
> > "Yes, you can capture your opponent's piece by moving your white knight from f3 to e5, capturing the black pawn."
> >
> >### Analysis:
>  >1. **Accuracy:**
>    >- **Assistant A:** Incorrectly states that the white pawn on e5 can capture the black pawn on d6. This is incorrect based on the image context provided, as the white pawn on e5 is not in a position to capture a black pawn on d6.
>   > - **Assistant B:** Incorrectly states that the white knight on f3 can move to e5 to capture a black pawn. However, based on the image context, there is no black pawn on e5; the black pawn is actually on e5, so this move is not possible either.
> > 2. **Specificity:**
>    >- **Assistant A:** Provided a specific move in algebraic notation (e5xd6) but based on an inaccurate assessment of the board.
>   > - **Assistant B:** Also provided a specific move, mentioning the knight move to e5, but again, this is not possible as described.
> > 3. **Fluency:**
>    > - Both assistants provided fluent and understandable responses. However, fluency does not compensate for the lack of accuracy in both responses.
>  >
> >### Conclusion:
> >Both Assistant A and Assistant B made errors in their responses by suggesting moves that are not possible based on the positions of the pieces described in the image context. However, Assistant B's mistake is somewhat less significant since it suggests a knight move that could be plausible with different board configurations, whereas Assistant A's move is completely off due to misidentifying the potential capture.
>  >
> >Overall, **Response B is better** despite both being incorrect, as it involves a lesser error in the interpretation of the board state.

---

> > ### Author Response · Authors · 2024-08-17
> > **Response (2/5)**
> >
> > Otherwise, if we use **the image caption which is involved with the information of correct answers to instructions**, the judge's result will be correct.
> >
> >   > **Caption used in this setting:**  There is a chessboard in the image. The chessboard is arranged in a total of 64 squares. In order to facilitate describing the chess game, the chessboard is marked horizontally with the letters "abcdefgh" and vertically with the numbers "12345678". For example, the lower left corner is "a1". The white King is on "e1", the white Queen is on "d1", the white Rooks are on "a1" and "h1", the white Bishops are on "c1" and "a4", the white Knights are on "b1" and "f3", the white Pawns are on "a2", "b2", "c2","e4","f2","g2", and "h2". The black King is on "e8", the black Queen is on "d8", the black Rooks are "a8" and "h8", the black Bishops are "c8" and "f8", the black Knights are "c6" and "g8", the black Pawns are "a6", "b7", "c7", "d7", "e5", "f7", "g7" and "h7". **The player with white chess pieces can capture the opponent's piece by moving your white knight from f3 to e5, capturing the black pawn.**
> >
> >   The judge results are as follows, where Assistant A is a LVLM, and Assistant B is human.
> >
> >   > To evaluate which second turn response better addresses the user's instruction, let's analyze both responses step by step.
> >   >
> >   > ### The user's second instruction:
> >   > "I am the player with the white chess pieces. Is there a move I can take on this turn to capture one of my opponent’s pieces?"
> >   >
> >   > ### Response from Assistant A:
> >   > **Content:**
> >      > - Assistant A suggests that the white pawn on e5 can capture the black pawn on d6, written as e5xd6.
> >   >
> >   > **Accuracy:**
> >      > - This response is inaccurate based on the image context. The white pawn is not on e5; it is on e4, and there is no black pawn on d6 to capture.
> >   >
> >   > **Relevance:**
> >      > - The response addresses the user's question directly but provides incorrect information.
> >   >
> >   > **Specificity:**
> >      > - The move suggestion is clear but incorrect, which significantly reduces the response's value.
> >   >
> >   > ### Response from Assistant B:
> >   > **Content:**
> >      > - Assistant B correctly identifies that the white knight on f3 can capture the black pawn on e5 by moving to e5.
> >   >
> >   > **Accuracy:**
> >      > - This response is accurate according to the image context. The knight on f3 can indeed capture the black pawn on e5.
> >   >
> >   > **Relevance:**
> >      > - The response is directly relevant to the user's question and provides the correct move.
> >   >
> >   > **Specificity:**
> >      > - The move suggestion is both clear and accurate, making it a strong response.
> >   >
> >   > ### Conclusion:
> >   > - **Response A** provides incorrect information and suggests an impossible move.
> >   > - **Response B** correctly identifies a valid capture move for the white pieces.
> >   >
> >   >Overall, **Response B is better**.

---

> > ### Author Response · Authors · 2024-08-17
> > **Response (3/5)**
> >
> > Thanks for your suggestions. We will add a discussion about Automation in the final version. As a pioneering effort in multi-turn multi-modal conversation evaluation with hierarchical ablation capability, we believe this work holds great significance for the community. However, we also believe there is still room for improvement in the automation of data production. After many adjustments and refinements to our annotation framework, we utilized GPT to reduce human labor in steps 3 and 4 (refer to Figure 3) for annotating captions and answers. For other steps, we have found that GPT works not good enough, and we will continue to explore automation in data production.
> >
> > > **A Discussion on Automation for Multi-Turn Instruction Formation. (Opportunities For Improvement #2)**
> >
> > It would surely be valuable to explore how to automate these processes to facilitate scaling. However, ensuring automatic annotation results of the same high quality as those of humans is undoubtedly very challenging. Our experiment results reveal LVLMs (like GPT4V) have weak performance for solving diverse and never-before-seen real-world tasks, and not to mention break down these kinds of tasks into sub-tasks (the first turn is perception, the second turn is reasoning and the third turn is creation.) Moreover, the pronouns (e.g it,they) are also needed for formatting the follow-up multi-turn instructions.
> >
> > Facing these challenges, there are some strategies for automating these processes to facilitate scaling: (a) GPT4V can be prompted to extend single-turn to multi-turn instructions, and human annotators rigorously filter these multi-turn instructions to form the final dataset. (b) A set of hierarchical instructions for a task (e.g board game playing) can be reused for multiple similar images. The samples in ConvBench can be used as seeds to facilitate scaling. （c)  If ConvBench's method of decomposing instructions becomes a paradigm in the benchmarking field for multi-modal large models, with the power of the community, the hierarchical multi-turn instructions that are practical for different domains will be accumulated and scaled up. (d) Depending on the above collection methods of hierarchical multi-turn instructions, a high-quality pretrained dataset for extending single-turn to multi-turn instructions can also be organized. The dataset can be applied for Continue PreTraining and Supervised Finetuning on LLMs or LVLMs to obtain a large model, which can extend single-turn to multi-turn instructions automatically and effectively.
> >
> > > **A Discussion on Automation for Task Category Induction. (Opportunities For Improvement #2)**
> >
> > Thanks for your brilliant suggestions. A discussion about this will also be added in the final version. Using a large language model for Task Category Induction can be a very feasible solution. However, there are still some hidden problems: (a) the results of induction via LLMs are not correct. (b) A task is summarized into two forms of expression (e.g. Travel Plan Writing and Formulating a Travel Plan), which will be represented as two kinds of tasks. Therefore, necessary manual annotations are still required to ensure the accuracy of Task Category Induction so far, and we will continue research in future work.

---

> > ### Author Response · Authors · 2024-08-17
> > **Response (4/5)**
> >
> > > **Types of Multi-Turn Interactions for Best Enhancing Perception and Reasoning. (Opportunities For Improvement #2)**
> >
> > Every sample in ConvBench carefully designs the three-turn progressive hierarchical instructions. These instructions have also undergone additional manual review to ensure their progressive and tiered nature. In other words, the correct answers from the previous turn are, in theory, designed to assist with the subsequent turn. According to hierarchical evaluation shown in Table 1, we can summarize the types of multi-turn interactions, which can best enhance reasoning and creation as follows:
> >
> > (a) **OCR, Fine-Grained Visual Recognition, and Visual Spatial Perception possess the strongest enhancing capabilities:** Under conditions of perfect perception, we see significant improvements in reasoning and creation abilities, as indicated by the data in the $\hat{S_{2}}$ comparing with $S_{2}$ and $\hat{S_{3}}$ comparing with $S_{3}$, as shown in Table 1.
> >
> > By analyzing these enhanced reasoning or creation cases with perfect perception information, we can summarize the perception tasks which can best enhance the reasoning and creation tasks:
> >
> > (1) **OCR Perception Task** enhances the performances of In-context Visual Scene Understanding, Meme Reasoning, and Chart Reasoning.
> >
> > (2) **Fine-Grained Perception Tasks** like Location Recognition perception task enhances the performances of Location Understanding and Travel Plan Writing tasks, Celebrity Recognition perception task may influence the performances of Celebrity Understanding, PowerPoint Production and Cultural Knowledge Reasoning tasks.
> >
> > (3) **Spatial Perception Tasks** like Board Chess Position Description enhances the performances of Board Game Reasoning and Diagram Generation tasks.
> >
> > (b) **Professional Knowledge, Emotional Intelligence, Imagination, and Sense of Space possess the strongest enhancing capabilities:** Under conditions of perfect perception, we see significant improvements in reasoning and creation abilities, as indicated by the data in and $\hat{S_{3}}$ comparing with $\tilde{S_{3}}$, as shown in Table 1.
> >
> > By analyzing these enhanced creation cases with perfect reasoning information, we can summarize the reasoning tasks which can best enhance creation tasks:
> >
> > (1) **Professional Knowledge Reasoning** can enhance the next turn creation tasks. For example, Physical Knowledge Reasoning improves the accuracy of Physical Problem Computing, Flower Reasoning enhances the performance of Plant Growing Plan.
> >
> > (2) **Emotional Intelligence Reasoning** can improve the relative creation tasks. For example, Human Emotion Reasoning improves the performance of Blog Writing, Humanity Discussion, and How Visual Content Arouses Emotions creation tasks.
> >
> > (3) **Imagination Reasoning** can boost the performance of relative creation tasks. For example, Tangram Speculation improves the corresponding creative task (Tangram Segmentation).
> >
> > (4) **Spatial Reasoning** can increase the performance of corresponding creation tasks. For example, Relative Position Reasoning can enhance performance for Navigation tasks.

---

> > > ### Author Response · Authors · 2024-08-17
> > > **Response (5/5)**
> > >
> > > > **Hard to Read for the Experiment Part. (Clarity)**
> > >
> > > We apologize for the lack of clarity in the experimental section. We will provide a concise and clear introduction to the experimental section. **If there are any specific areas that are still unclear to you, we would be very grateful if you could point them out, and we will go to great lengths to provide you with answers.**
> > >
> > > As for section 4, we explain how the evaluation for ConvBench work. ChatGPT3.5 with detailed caption will be used for judging LVLMs' performance on four dimensions: perception, reasoning, creation, and overall conversation.
> > >
> > > This evaluation process needs to be conducted three times by different settings:
> > >
> > > (a) Evaluation Without Any Ground-Truth, $S_{1}$, $S_{2}$, $S_{3}$, and $S_{O}$ in Table 1 are the performance results evaluated in this setting.
> > >
> > > (b) Evaluation With Perfect Perception Condition (the responses of reasoning and creation turns are obtained based on ground-truth visual perception, like the most right sub-figure of Figure 1(c)). $\hat{S_{2}}$, $\hat{S_{3}}$, and $\hat{S_{O}}$ in Table 1 are the performance results evaluated in this setting.
> > >
> > > (c) Evaluation With Perfect Perception and Reasoning Conditions (the responses of creation turns are obtained based on ground-truth visual perception and reasoning). $\tilde{S_{3}}$, and $\tilde{S_{O}}$ in Table 1 are the performance results evaluated in this setting.
> > >
> > > As for section 5, we have given some insightful findings by analyzing the experimental results:
> > >
> > > (a) ConvBench is a significant challenge for current LVLMs, even for GPT4V, which achieves only a 39.51% score.
> > >
> > > (b) By comparing $S_{2}$ and $\hat{S_{2}}$ in Table 1, we can find that weak perception undermines LVLMs' reasoning performance, especially for "OCR Perception Task", "Fine-grained Perception Task" and "Spatial Perception Task".
> > >
> > > (c) By comparing $\hat{S_{3}}$ and $\hat{S_{3}}$ in Table 1, we can find limited reasoning impacts LVLMs' creation abilities, especially for "Professional Knowledge Reasoning", "Emotional Intelligence Reasoning", and "Imagination Reasoning".
> > >
> > > (d) As shown in Figure 2 in the Supplementary File, we can see ConvBench aids in highlighting the strengths and weaknesses of the instruction-following LVLMs along various real-world use cases. It can also be seen that the development of current LVLMs are synchronous.
> > >
> > > (e) According to $S_{O}$ in Table 1, ConvBench presents substantial challenges in multi-turn visual conversations with LVLMs and plays a role in the LVLM field, such as MT-Bench [1] in the LLM field.
> > >
> > > [1] Lianmin Zheng, Wei-Lin Chiang, Ying Sheng, Siyuan Zhuang, Zhanghao Wu, Yonghao Zhuang, Zi Lin, Zhuohan Li, Dacheng Li, Eric P. Xing, Haotong Zhang, Joseph Gonzalez, and Ion Stoica. Judging llm-as-a-judge with mt-bench and chatbot arena. ArXiv, abs/2306.05685, 2023.
> > >
> > > (f) A decline between the first turn and subsequent turns, which suggests that LVLMs tend to generate
> > > comprehension biases as the multi-turn dialogue progresses or forget the content of previous turns.
> > >
> > > (g) The high-quality dialogue history, which plays an important role in in-context learning examples, provides effective guidance for the LVLMs' responses.
> > >
> > > (h) We find that the trend of increasing language model size is related  to an improvement in LVLMs' performance on multi-turn conversation by comparing the InterVL-Chat-v1-2, ShareGPT4-13B, and LLaVA-V1.5-13B with the InterVL-Chat-v1-5, ShareGPT4-7B, and LLaVA-V1.5-7B.

---

> > > > ### Author Response · Authors · 2024-08-26
> > > >
> > > > Dear Reviewer HBs1,
> > > >
> > > > Thank you once again for your comprehensive review and constructive comments on our paper. In response to your feedback, we have proved the effectiveness of the ConvBenchEval, have discussed the possible automation strategies for the annotation pipeline and the enhanced types of multi-turn interactions, have made a clarification for the experiment part.
> > > >
> > > > As the rebuttal period concludes the discussion deadline, we would like to inquire if there are any additional questions or concerns about our work. We would be happy to provide any additional clarifications if requested.
> > > >
> > > >
> > > > Thanks again for the time and effort in our work!
> > > >
> > > > Best wishes,
> > > >
> > > > ConvBench Authors.

---

> > > > > ### Comment · Reviewer_HBs1 · 2024-08-30
> > > > > **Response**
> > > > >
> > > > > Dear Authors:
> > > > >
> > > > > Thank you for your thorough and detailed response to my concerns. I appreciate the clarifications provided, particularly regarding the enhancements made to the GPT-3.5 evaluation process and the ongoing efforts to explore automation. Your explanations and the additional insights into how the detailed captions improve the evaluation accuracy have addressed many of my initial reservations.
> > > > >
> > > > > Given these considerations, I will raise my score.

---

> > > > > > ### Author Response · Authors · 2024-08-31
> > > > > >
> > > > > > Dear Reviewer HBs1,
> > > > > > ﻿
> > > > > >
> > > > > > We would like to extend our sincere gratitude for the thorough and insightful review you have provided for our ConvBench paper. Your constructive feedback is invaluable and will undoubtedly lead to significant enhancements in the quality of our work. We are particularly appreciative of your final decision to adjust your score upwards. Thank you once again for the time and effort you've invested in our work!
> > > > > > ﻿
> > > > > >
> > > > > > Best regards,
> > > > > > ﻿
> > > > > >
> > > > > > The ConvBench Authors.

---

### Decision · Program_Chairs · 2024-09-26

**Decision:**

Accept (Spotlight)

**Comment:**

This paper presents ConvBench, a multi-turn conversation benchmark that contains 577 curated multi-turn conversations.  While the number is limited, the tasks contained in these conversations are broad and open-ended.  They are good at testing the capability of LVLMs.  The paper also includes an automatic framework for evaluation.   The paper experiments the leading LLMs on this dataset and shows a significant challenge for current LVLMs including GPT-4v.   The authors successfully addressed a few concerns from the reviewers. Two reviewers increased their scores.  Please revise the paper according to these concerns.